# Provably Accelerated Imaging with Restarted Inertia and Score-based Image Priors

**Marien Renaud**[1]*, **Julien Hermant**[1]*, **Deliang Wei**[2]*, **Yu Sun**[2]

[1]University of Bordeaux, Bordeaux INP, CNRS, IMB, UMR 5251, F-33400 Talence, France
[2]Johns Hopkins University, Baltimore MD, USA
`{deliang, ysun214}@jh.edu`
`{marien.renaud, julien.hermant}@math.u-bordeaux.fr`

## Abstract

Fast convergence and high-quality image recovery are two essential features of algorithms for solving ill-posed imaging inverse problems. Existing methods, such as *regularization by denoising (RED)*, often focus on designing sophisticated image priors to improve reconstruction quality, while leaving convergence acceleration to heuristics. To bridge the gap, we propose *Restarted Inertia with Score-based Priors (RISP)* as a principled extension of RED. RISP incorporates a restarting inertia for fast convergence, while still allowing score-based image priors for high-quality reconstruction. We prove that RISP attains a faster stationary-point convergence rate than RED, without requiring the convexity of the image prior. We further derive and analyze the associated continuous-time dynamical system, offering insight into the connection between RISP and the *heavy-ball ordinary differential equation (ODE)*. Experiments across a range of imaging inverse problems demonstrate that RISP enables fast convergence while achieving high-quality reconstructions.

## 1 Introduction

Imaging inverse problems aim to recover an unknown image from its undersampled, noisy measurements. These problems arise in many domains, ranging from low-level computer vision to biomedical imaging. Since the measurements often cannot losslessly specify the underlying image, the problem is inherently *ill-posed*. A standard remedy is to incorporate image priors to regularize reconstruction and promote high-quality solutions (Rudin et al., 1992; Mallat, 1999; Elad & Aharon, 2006).

There has been considerable interest in using image denoisers as priors in iterative reconstruction algorithms (Zhang et al., 2017b). This interest continues to grow with the recognition of the conceptual link between denoisers and the score function used in generative diffusion models (Efron, 2011; Song & Ermon, 2019; Song et al., 2021). *Regularization by denoising (RED)* (Romano et al., 2017) is such a framework that uses an off-the-shelf denoiser to approximate the *score (i.e., the gradient of the log-density)* of the implicit image prior encoded by that denoiser. When equipped with deep image denoisers, RED methods have been shown to achieve excellent imaging performance in various imaging problems (Metzler et al., 2018; Wu et al., 2020; Iskender et al., 2023; Guennec et al., 2025). Similar results have also been achieved by a closely related class of algorithms known as *plug-and-play priors* (Venkatakrishnan et al., 2013; Sreehari et al., 2016).

However, RED methods are inherently iterative optimization procedures and can need a large number of iterations; this can result in substantial runtime on problems that require real-time processing or involve large-scale data. While considerable interest has been devoted to design sophisticated denoising priors to improve reconstruction quality (Renaud et al., 2024; Wei et al., 2024; Martin et al., 2024), fewer attention has been given to convergence acceleration and is often handled through heuristics (Sun et al., 2019a; Tang & Davies, 2020). A key challenge is that incorporating learned priors typically renders the problem nonconvex, limiting the applicability of classical acceleration techniques developed for convex settings.

---

*Equal contribution.

**Contributions.** In this work, we address the gap by proposing a novel *Restarted Inertia with Score-based Priors (RISP)* method. RISP significantly extends RED by integrating a *restarted inertia* technique to provably accelerate convergence, while still allowing the use of advanced score-based priors for high-quality image reconstruction. We present two algorithmic instantiations of RISP, termed *RISP-GM* and *RISP-Prox*, based on gradient and proximal formulations, respectively. Under practical assumptions, we establish for both algorithms stationary-point convergence at the rate of $\mathcal{O}(n^{-4/7})$, surpassing the $\mathcal{O}(n^{-1/2})$ rate of RED. Notably, our analysis does not require score-based priors to be convex, thus accommodating those parameterized by deep neural networks. We further derive and analyze the continuous dynamical system underlying RISP, and show that RISP-GM and RISP-Prox correspond to alternative discretizations of the second-order *heavy-ball ordinary differential equation (ODE)*. We experimentally validate RISP across a range of linear and nonlinear inverse imaging problems. In particular, our results show that RISP can enable substantial acceleration (up to $24\times$) for large-scale image reconstruction.

## 2 BACKGROUND

**Inverse Problems.** We consider the general inverse problem $\boldsymbol{y} = \boldsymbol{A}(\boldsymbol{x}) + \boldsymbol{n}$, in which the goal is to recover $\boldsymbol{x} \in \mathbb{R}^d$ given the measurements $\boldsymbol{y} \in \mathbb{R}^m$. The forward operator $\boldsymbol{A} : \mathbb{R}^d \to \mathbb{R}^m$ models the response of the imaging system, and $\boldsymbol{n} \in \mathbb{R}^m$ represents the measurement noise, which is often assumed to be additive white Gaussian noise (AWGN). One popular inference framework for solving inverse problems is based on the *maximum a posteriori (MAP)* estimation

$$\hat{\boldsymbol{x}} \in \arg\max_{\boldsymbol{x}\in\mathbb{R}^d}\{p(\boldsymbol{y}|\boldsymbol{x})p(\boldsymbol{x})\} = \arg\min_{\boldsymbol{x}\in\mathbb{R}^d}\big\{F(\boldsymbol{x}) := f(\boldsymbol{x}) + g(\boldsymbol{x})\big\}, \tag{1}$$

where $f = -\log p(\boldsymbol{y}|\cdot)$ is often known as the data-fidelity term and $g = -\log p$ as the regularizer. Commonly, the data-fidelity is set to the least-square loss $f(\boldsymbol{x}) = (\lambda/2)\|\boldsymbol{y} - \boldsymbol{A}(\boldsymbol{x})\|^2$, which corresponds to the Gaussian likelihood $\boldsymbol{y}|\boldsymbol{x} \sim \mathcal{N}(\boldsymbol{A}(\boldsymbol{x}), (1/\lambda)\boldsymbol{I})$.

**Regularization by denoising.** RED is a widely used framework that employs an image denoiser $\mathsf{D}_\sigma : \mathbb{R}^d \to \mathbb{R}^d$ to model prior, where $\sigma > 0$ controls the denoising strength. In RED, the noise residual $\boldsymbol{x} - \mathsf{D}_\sigma(\boldsymbol{x})$ is used as an surrogate for the gradient of an implicit regularizer. Two popular RED algorithms are the *RED gradient method (RED-GM)*

$$\boldsymbol{x}^+ = \boldsymbol{x} - \eta\big(\nabla f(\boldsymbol{x}) + \tau(\boldsymbol{x} - \mathsf{D}_\sigma(\boldsymbol{x}))\big) \tag{2}$$

and the *RED proximal method (RED-Prox)*

$$\boldsymbol{x}^+ = \mathsf{prox}_{\eta f}\big(\boldsymbol{x} - \eta \cdot \tau(\boldsymbol{x} - \mathsf{D}_\sigma(\boldsymbol{x}))\big), \tag{3}$$

where $\tau > 0$ is the weighting parameter and $\eta > 0$ the stepsize. The two algorithms differ in their treatment to $f$, where the former computes the gradient $\nabla f$, while the latter evaluates the *proximal map* $\mathsf{prox}_{\eta f}(\boldsymbol{z}) := \arg\min_{\boldsymbol{x}\in\mathbb{R}^d}\big\{\frac{1}{2}\|\boldsymbol{x} - \boldsymbol{z}\| + \eta f(\boldsymbol{x})\big\}$. The detailed formulations of both algorithms are summarized in Appendix G. By Tweedie's formula (Efron, 2011), when $\mathsf{D}_\sigma$ is the *minimum mean squared error (MMSE)* denoiser, the noise residual equals to the scaled version of the prior score

$$\mathsf{S}(\boldsymbol{x}) = -\big(\boldsymbol{x} - \mathsf{D}_\sigma(\boldsymbol{x})\big)/\sigma^2,$$

where $\mathsf{S} : \mathbb{R}^d \to \mathbb{R}^d$ denotes the score induced by the denoiser. When we fix $\tau = 1/\sigma^2$, RED can be interpreted as a score-based method (Reehorst & Schniter, 2019; Laumont et al., 2022; Sun et al., 2024). The empirical success has motivated theoretical work on the convergence properties of RED algorithms. Recent works have analyzed their fixed-point convergence (Cohen et al., 2021; Sun et al., 2021), interpreted RED as a MAP estimator (Hurault et al., 2022a; Laumont et al., 2023), or established convergence guarantees for stochastic RED (Laumont et al., 2023; Renaud et al., 2024). Our work complements these efforts by providing a formal analysis of accelerated convergence under restarted inertia.

**Plug-and-play priors (PnP).** PnP is an alternative framework that exploits denoisers as priors, inspired by the mathematical connection between proximal operator and MAP denoiser (Venkatakrishnan et al., 2013). Its key idea is to replace the proximal operator in an iterative algorithm directly with $\mathsf{D}_\sigma$ to impose the prior information. The applications and theory of PnP have been widely studied in (Zhang et al., 2017a; Ahmad et al., 2020; Wei et al., 2020; Zhang et al., 2022; Wei et al.,

| **Algorithm 1** RISP-GM | **Algorithm 2** RISP-Prox |
|---|---|
| **Require:** $\boldsymbol{x}^{-1} = \boldsymbol{x}^0 \in \mathbb{R}^d$, $K \in \mathbb{N}$, $\eta > 0$, $\theta \in$ (0, 1], and $B \geq 0$ | **Require:** $\boldsymbol{x}^{-1} = \boldsymbol{x}^0 \in \mathbb{R}^d$, $K \in \mathbb{N}$, $\eta > 0$, $\theta \in$ (0, 1], and $B \geq 0$ |
| 1: **while** $0 \leq k < K$ **do** | 1: **while** $0 \leq k < K$ **do** |
| 2: $\quad \boldsymbol{z}^k = \boldsymbol{x}^k + (1-\theta)(\boldsymbol{x}^k - \boldsymbol{x}^{k-1})$ | 2: $\quad \boldsymbol{z}^k = \boldsymbol{x}^k + (1-\theta)(\boldsymbol{x}^k - \boldsymbol{x}^{k-1})$ �è Inertia |
| 3: $\quad \boldsymbol{x}^{k+1} = \boldsymbol{z}^k - \eta\big(\nabla f(\boldsymbol{z}^k) - \mathsf{S}(\boldsymbol{z}^k)\big)$ | 3: $\quad \boldsymbol{x}^{k+1} = \mathsf{prox}_{\eta f}\big(\boldsymbol{z}^k + \eta\,\mathsf{S}(\boldsymbol{z}^k)\big)$ |
| 4: $\quad k = k + 1$ | 4: $\quad k = k + 1$ |
| 5: $\quad$ **if** $k \sum_{t=0}^{k-1} \|\boldsymbol{x}^{t+1} - \boldsymbol{x}^t\|^2 > B^2$ **then** | 5: $\quad$ **if** $k \sum_{t=0}^{k-1} \|\boldsymbol{x}^{t+1} - \boldsymbol{x}^t\|^2 > B^2$ **then** |
| 6: $\quad\quad \boldsymbol{x}^{-1} = \boldsymbol{x}^0 = \boldsymbol{x}^k, k = 0$ | 6: $\quad\quad \boldsymbol{x}^{-1} = \boldsymbol{x}^0 = \boldsymbol{x}^k, k = 0$ �è Restart |
| 7: $\quad$ **end if** | 7: $\quad$ **end if** |
| 8: **end while** | 8: **end while** |
| 9: $K_0 = \arg\min_{\lfloor \frac{K}{2} \rfloor \leq k \leq K-1} \|\boldsymbol{x}^{k+1} - \boldsymbol{x}^k\|$ | 9: $K_0 = \arg\min_{\lfloor \frac{K}{2} \rfloor \leq k \leq K-1} \|\boldsymbol{x}^{k+1} - \boldsymbol{x}^k\|$ |
| 10: **return** $\hat{\boldsymbol{z}} = \frac{1}{K_0+1} \sum_{k=0}^{K_0} \boldsymbol{z}^k$ | 10: **return** $\hat{\boldsymbol{z}} = \frac{1}{K_0+1} \sum_{k=0}^{K_0} \boldsymbol{z}^k$ |

2022)and (Chan et al., 2016; Buzzard et al., 2018; Ryu et al., 2019; Sun et al., 2019b; Sun* et al., 2021), respectively; see also a review in (Kamilov et al., 2023). Recently, (Park et al., 2025) showed that the denoiser in PnP can also be formulated as a score function, rendering PnP as a score-based method. While we develop RISP through the extension of RED, our framework and analysis can potentially be applied to PnP as well.

**Inertial acceleration for RED/PnP.** Several recent works have investigated the use of inertia (*i.e.*, momentum) in RED/PnP (Kamilov et al., 2018; He et al., 2019; Tan et al., 2024; Wu et al., 2024; Chow et al., 2024). These adaptations include an inertia update before applying the gradient and denoiser; detailed discussion on these works are provided in Appendix A. While these works demonstrate the practical effectiveness of inertia, convergence analysis with an explicit accelerated rate remains missing. Our work fills the gap by establishing a provable accelerated convergence rate.

Proving such acceleration is challenging due to the nonconvex nature of the score-based priors. It has been shown that the use of inertia does not improve the worst-case convergence rate under general Lipschitiz-continuous gradients (Carmon et al., 2020). Our analysis is motivated by recent advances in nonconvex optimization, which rely on the *Lipschitz-continuous Hessian* condition (Carmon et al., 2017b; Jin et al., 2018; Li & Lin, 2023); we discuss in Section 5 that such property is satisfied in many imaging applications. We note that the continuous-time limit of inertial acceleration offers complementary insights (Su et al., 2016; Siegel, 2019; Aujol et al., 2023; 2024b; Attouch et al., 2022; Shi et al., 2021; Li et al., 2024; Hermant et al., 2024; Gupta & Wojtowytsch, 2025). Our analysis further investigate this aspect by bridging the discrete RISP algorithms with the continuous-time heavy-ball ODE.

**Relationship to diffusion models (DMs).** DMs (Ho et al., 2020; Song et al., 2021) have recently emerged as a powerful tool for solving imaging inverse problems. Unlike PnP/RED, which seeks a MAP estimate via optimization, DM-based solvers aim to sample from the posterior distribution $p(\boldsymbol{x}|\boldsymbol{y})$. Recent works have investigated various strategies for incorporating the likelihood term $p(\boldsymbol{y}|\boldsymbol{x})$ into the diffusion process (Chung et al., 2022a;b; Zhu et al., 2023; Song et al., 2023; Mardani et al., 2023). Despite progress in linear inverse problems (Dou & Song, 2024), provable posterior sampling under general settings remains open. A key element shared by both DM and RED is Tweedie's formula, which connects the score function with a denoiser. This implies that score networks from pre-trained diffusion models can be used within RISP.

## 3 RESTARTED INERTIA WITH SCORE-BASED PRIORS

RISP addresses slow convergence by leveraging an inertia update paired with a restarting mechanism. We begin by discussing this central component and then present two algorithmic instantiations of RISP.

The first mechanism in RISP is an inertial step. Consider RED-GM as an example. A natural approach to accelerate convergence is given by

$$\begin{aligned}
\boldsymbol{z} &= \boldsymbol{x} + (1 - \theta)(\boldsymbol{x} - \boldsymbol{x}^-) \\
\boldsymbol{x}^+ &= \boldsymbol{z} - \eta\big(\nabla f(\boldsymbol{z}) - \mathsf{S}(\boldsymbol{z})\big),
\end{aligned} \tag{4}$$

where $\boldsymbol{z}$ accounts the inertia from the previous update $\boldsymbol{x}^-$, and parameter $\theta \in (0, 1]$ controls the inertial contribution. When $\theta = 1$, (4) reduces to normal RED-GM. Note that the corresponding RED-Prox variant follows analogously.

Nevertheless, accumulated inertia can cause instability and undesirable overshooting, especially in nonconvex cases (O'donoghue & Candes, 2015); this behavior causes the theoretical limitation in establishing faster convergence rates. The top panel of Figure 1 presents an intuitive illustration. To address the problem, RISP further employs a *restarting mechanism* (Li & Lin, 2023) that resets the inertia whenever the accumulated relative error since the last restart exceeds a threshold

$$\textbf{if} \quad k \sum_{t=0}^{k-1} \|\boldsymbol{x}^{t+1} - \boldsymbol{x}^t\|^2 > B^2, \tag{5}$$
$$\textbf{then} \quad \boldsymbol{x}^{-1} = \boldsymbol{x}^0 = \boldsymbol{x}^k, k = 0$$

where $B > 0$ is a user-defined constant. Once triggered, (5) clears accumulated inertia and forces RISP to rely on the local gradient, which helps prevent overshooting and deviation from stationary points. The bottom panel of Figure 1 conceptually illustrates this behavior.

We derive RISP-GM and RISP-Prox as two algorithmic instantiations of RIPS, extending RED-GM and RED-Prox, respectively. The algorithmic details of both algorithms are summarized in Algorithms 1 and 2, where the inertia and restarting steps are highlighted in color. Unlike the original RED algorithms in (2) and (3), RIPS directly employs a pre-trained score function as the negative gradient of a regularizer, which streamlines both the theoretical analysis and the empirical implementation. In the next section, our analysis shows that the restarting mechanism enables provable accelerated convergence under the nonconvex score-based priors.

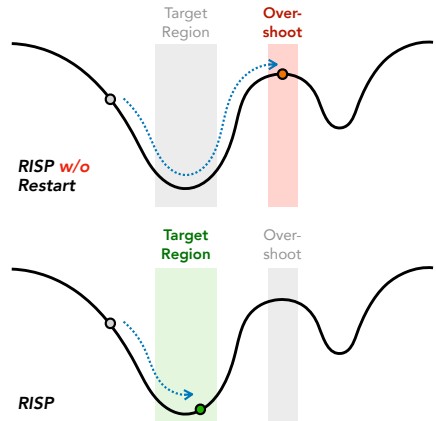

Figure 1: Conceptual illustration of how restarting stabilizes inertial algorithms. Without restarting, accumulated inertia can cause overshooting and escape from stationary points (*top*). Restarting clears inertia and enforces local gradient updates to prevent overshooting (*bottom*).

## 4 THEORETICAL ANALYSIS

In this section, we analyze the convergence behavior of RISP. We first derive explicit convergence rates for RISP-GM and RISP-Prox, showing they reach stationary points faster than their RED counterparts. Next, we analyze the underlying continuous-time dynamics of RISP to provide complementary insights into its behavior.

### 4.1 CONVERGENCE RATES OF RISP-GM & RISP-PROX

**Assumption 1.** *The score function* $\mathsf{S} : \mathbb{R}^d \to \mathbb{R}^d$ *is a gradient field, i.e., there exists a regularizer* $g$ *such that* $\mathsf{S} = -\nabla g$.

The assumption formally connects the score function with a regularizer. In practice, one can implement such a deep score function using the *gradient step denoiser* technique proposed in (Hurault et al., 2022a).

**Assumption 2.** *(i) The data-fidelity term* $f$ *has an* $L_f$*-Lipschitz continuous gradient, i.e., for all* $\boldsymbol{x}, \boldsymbol{y} \in \mathbb{R}^d$, $\|\nabla f(\boldsymbol{x}) - \nabla f(\boldsymbol{y})\| \le L_f \|\boldsymbol{x} - \boldsymbol{y}\|$. *(ii) The score function* $\mathsf{S}$ *is Lipschitz continuous with constant* $L_g$; *namely, regularizer* $g$ *has a* $L_g$*-Lipschitz continuous gradient. Throughout the paper, we set* $L := L_f + L_g$.

Assumption 2(i) is a standard condition that can be satisfied in most imaging applications. Assumption 2(ii) is also practical because it only assumes the existence of the Lipschitz constant. One can enforce this condition by employing *Lipschitz activation functions*. We refer to Apendix G.1 for a detailed discussion.

To facilitate discussion, we present the convergence rate of RED as baseline. The following proposition states the result for RED-GM; the result for RED-Prox can be found in Appendix C.

**Proposition 1.** *Let Assumptions 1-2 hold and $\eta = 1/L$. Then, with at most $n$ iterations, RED-GM outputs a point $\hat{x}$ such that*

$$\|\nabla F(\hat{x})\| \leq \frac{A_0}{\sqrt{n}} = \mathcal{O}(n^{-1/2}),$$

*where $A_0 = \sqrt{2L\Delta_F}$, and $\Delta_F := F(x^0) - \min F$. We recall that $x^0$ is the initialization.*

The detailed proof is provided in the appendix. The proposition shows that RED-GM approximates a stationary point at the rate of $\mathcal{O}(n^{-1/2})$, which is consistent with classic nonconvex results for gradient descent. The same convergence rate also holds for RED-Prox.

We now establish faster convergence rates for the RISP algorithms. Our analysis requires an additional assumption on the second-order properties of the data-fidelity term and of the score function.

**Assumption 3.** *(i) The data-fidelity term $f$ has an $\rho_f$-Lipschitz continuous Hessian, i.e., for all $x, y \in \mathbb{R}^d$, $\|\nabla^2 f(x) - \nabla^2 f(y)\| \leq \rho_f \|x - y\|$. (ii) The score function $\mathsf{S}$ is Lipschitz Jacobian continuous with constant $\rho_g$; namely, regularizer $g$ has a $\rho_g$-Lipschitz continuous Hessian. Throughout the paper, we set $\rho := \rho_f + \rho_g$.*

This assumption is standard in analyses of accelerated methods for nonconvex problems (Carmon et al., 2017a; Agarwal et al., 2017; Li & Lin, 2023; Marumo & Takeda, 2024). We note that Assumption 3(i) holds for all linear inverse problems with AWGN. Even when the forward model is nonlinear, the assumption could also be satisfied; one example is the Rician denoising problem presented in Section 5. While Assumption 3(ii) may seem strong, we stress that the condition can be satisfied in practice by combining gradient-step denoisers and Lipschitz activation functions. In Appendix G, we present a proposition that formally establishes the existence of $\rho_g$ for such score functions. Importantly, both Assumption 2(ii) and 3(ii) do not assume the convexity of score-based priors.

**Theorem 1.** *Let Assumptions 1-3 hold. Let $\eta = 1/(4L)$, $B = \sqrt{\varepsilon/\rho}$, $\theta = 4(\varepsilon\rho\eta^2)^{1/4} \in (0, 1)$, and $K = \theta^{-1}$. Then, with at most $n$ iterations, RISP-GM outputs a point $\hat{z}$ such that*

$$\|\nabla F(\hat{z})\| \leq 82\varepsilon = \mathcal{O}(n^{-4/7}),$$

*where $\varepsilon = 2^{4/7}\Delta_F^{4/7}L^{2/7}\rho^{1/7}n^{-4/7} + L^2\rho^{-1}n^{-4}$.*

The proof of the theorem is provided in Appendix D, which extends that of (Li & Lin, 2023, Theorem 1) to the RISP setting. Theorem 1 establishes that RISP-GM approximates a stationary point at the rate of $\mathcal{O}(n^{-4/7})$, which is clearly faster than the $\mathcal{O}(n^{-1/2})$ rate attained by RED.

The second theorem analyzes RISP-Prox. The proximal step introduces additional analytical challenges. Specifically, it requires a uniform lower bound on the curvature of the objective to control the proximal mapping. To address this issue, we introduce the following assumption.

**Assumption 4.** *The data-fidelity term $f$ is convex, and the regularizer $g$ is $\nu$-weakly convex, i.e., $g + \frac{\nu}{2}\|\cdot\|^2$ is convex.*

Note that $f$ is convex for all linear inverse problems with AWGN. As $g$ is already assumed to be $L_g$-Lipschitz, it follows that $g$ is weakly convex with $\nu \leq L_g$ (Zhou, 2018). Assumption 4 defines a specific lower bound on the eigenvalues of the Hessian matrix of $g$.

**Theorem 2.** *Let Assumptions 1-4 hold. Let $\nu \leq 8(\varepsilon\rho)^{1/4}\sqrt{L}$, $\eta = 1/(8L)$, $B = \sqrt{\varepsilon/(4\rho)}$, $\theta = 4(\varepsilon\rho\eta^2)^{1/4} \in (0, 1)$, and $K = \theta^{-1}$. Then, with at most $n$ iterations, RISP-Prox outputs a point $\hat{z}$ such that*

$$\|\nabla F(\hat{z})\| \leq 45\varepsilon = \mathcal{O}(n^{-4/7}),$$

*where $\varepsilon = \Delta_F^{4/7}(2L)^{2/7}\rho^{1/7}n^{-4/7} + 4L^2\rho^{-1}n^{-4}$.*

The proof of the theorem is provided in Appendix E. Theorem 1 establishes a comparable acceleration guarantee for RISP-Prox to that given for RISP-GM, yielding the same $\mathcal{O}(n^{-4/7})$ convergence. Together, the two theorems rigorously show that both RISP-GM and RISP-Prox achieve improved convergence rates relative to their RED counterparts.

## 4.2 CONTINUOUS-TIME ANALYSIS

We further study the continuous-time system associated with RISP. We begin by first establishing the connection between RISP and the heavy ball ODE. Under Assumption 1, we can show that *inertial part* of RISP algorithms (lines 2-3 in Algorithms 1 & 2) is governed by

$$\ddot{\boldsymbol{x}}_t + \alpha\dot{\boldsymbol{x}}_t + \nabla F(\boldsymbol{x}_t) = 0, \tag{6}$$

where $\alpha := \lim_{\eta\to 0}(\theta(\eta)/\sqrt{\eta})$, and we recall that $F = f + g$ where $g$ is defined by the score $\mathsf{S}$. A detailed derivation can be found in Appendix F. Equation (6) defines the *heavy ball* equation (Polyak, 1964), whose solution $(\boldsymbol{x}_t)_{t\geq 0}$ can be thought as a rolling object on the landscape of $F$, subject to some friction parameterized by $\alpha$.

However, the heavy ball ODE does not account for the restarting mechanism used in RISP. To address this, we introduce a restarted variant of (6), which is summarized in Algorithm 3. The system follows the heavy-ball dynamics until a restart criterion is met, at which point the inertia (*i.e.*, the velocity term) is reset to zero. This continuous restarted ODE generalizes the discrete RISP-GM and RISP-Prox algorithms and thus bridges the continuous and discrete perspectives. In particular, RISP-GM can be viewed as the Euler discretization of the continuous RISP dynamics, while RISP-Prox corresponds to the forward-backward discretization where the proximal step computes the backward gradient. We note that continuous RISP serves as a general formulation and can inspire the development of other discrete acceleration algorithms.

---

**Algorithm 3** Continuous RISP

Initialize $\boldsymbol{x}_{0,0} \in \mathbb{R}^d$, $\dot{\boldsymbol{x}}_{0,0} = 0$, $k = 0$.
**while** $t < T_{\mathsf{max}}$ **do**
    **while** $t\int_0^t \|\dot{\boldsymbol{x}}_t\|^2\,ds \leq B^2$ **do**
        $\{\boldsymbol{x}_{t,k}\}_{t\in\mathbb{R}_+}$ follows
        $\ddot{\boldsymbol{x}}_{t,k} + \alpha\dot{\boldsymbol{x}}_{t,k} + \nabla F(\boldsymbol{x}_{t,k}) = 0$
    **end while**
    $k = k+1$, $t = 0$
    $\boldsymbol{x}_{0,k+1} = \boldsymbol{x}_{t,k}$, $\dot{\boldsymbol{x}}_{0,k+1} = 0$
**end while**

---

The following theorem establishes the convergence guarantee for continuous RISP.

**Theorem 3.** *Let Assumptions 1 and 3 hold. Consider the $(\boldsymbol{x}_t^c)_{t\geq 0}$ running for a total execution time $T \in \mathbb{R}_+$. Define the output $\hat{\boldsymbol{x}}$ as the average $\hat{\boldsymbol{x}} := \frac{1}{K_0}\int_0^{K_0} \boldsymbol{x}_{t+\sum_{i=1}^K T_i}^c\,dt$, where $K_0 = \arg\min_{t\in\left[\frac{T_{\mathsf{max}}}{2}, T_{\mathsf{max}}\right]} \left\|\dot{\boldsymbol{x}}_{t+\sum_{i=1}^K T_i}^c\right\|$. Set the parameters as $\alpha = (\varepsilon\rho)^{1/4}$, $T_{\mathsf{max}} = (\varepsilon\rho)^{-1/4}$, and $B = \sqrt{\varepsilon/\rho}$. Then, under these conditions, the gradient norm satisfies*

$$\|\nabla F(\hat{\boldsymbol{x}})\| \leq 5\varepsilon = \mathcal{O}(T^{-4/7}),$$

*with $\varepsilon = 2^{4/7}\rho^{1/7}\Delta_F^{4/7}T^{-4/7} + 2^4\rho^{-1}T^{-4}$.*

The complete proof is provided in Appendix F.3. Theorem 3 states that continuous RISP approximate a stationary point at the rate of $\mathcal{O}(T^{-4/7})$, which is consistent with the $\mathcal{O}(n^{-4/7})$ convergence achieved by discrete RISP algorithms. Note that the Lipschitz continuity of the gradient is not required in the theorem. In fact, this condition is often introduced in discrete analyses for selecting a stepsize that preserves the property of the continuous dynamics. However, the Lipschitz-Hessian assumption is essential here, as it ensures that when the process stops, the final point has a small gradient norm.

## 5 EXPERIMENTS

We now present our experimental results, which are organized around three goals. First, we validate the accelerated convergence of RISP on linear inverse problems that satisfy our assumptions, providing empirical support for the theoretical guarantees. Second, we apply RISP to a nonlinear inverse problem where some assumptions do not hold, demonstrating the method's robustness

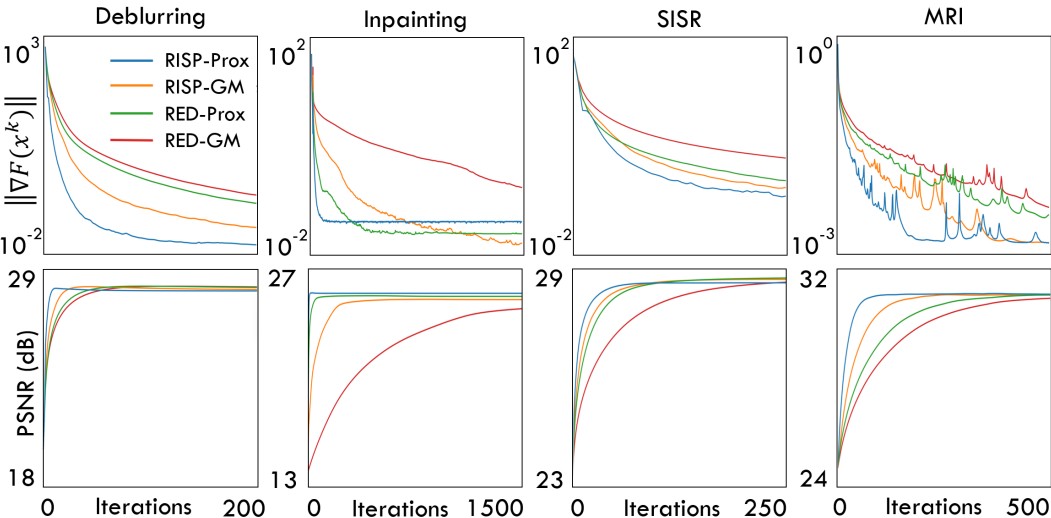

Figure 2: Visualizations of the gradient-norm (*top row*) and PSNR (*bottom row*) curves for RISP and baseline methods on four linear inverse problems. All curves are averaged over the test dataset, and the $x$-axis shows the iteration number. By employing the restarted inertia, RISP achieves faster convergence without compromising the performance.

beyond the ideal setting. Third, we evaluate RISP on a large-scale image reconstruction task to highlight its practical efficiency. All hyperparameters are manually tuned to maximize the *peak signal-to-noise ratio (PSNR)*; additional experimental results, hyperparameter robustness studies, and implementation details are provided in Appendix G. The source code is publicly available at `https://github.com/Hopkins-CIG/RISP`.

## 5.1 LINEAR INVERSE PROBLEMS

We validate our theorems on four linear inverse problems including *image deblurring*, *inpainting*, *single image super-resolution (SISR)*, and *magnetic resonance imaging (MRI)*. In these tasks, the data-fidelity terms satisfy all the assumptions under AWGN. We refer to Appendix G.2 for the technical details and additional results.

**Experimental setup.** We set the inertia parameter to $\theta = 0.2$ for RISP algorithms. We note that RISP is robust to this choice as the restart mechanism enhances stability; see Figure 16 in appendix. We employ a score function defined as $\mathsf{S} = -\nabla g_\sigma$. The associated regularizer is given by $g_\sigma(\boldsymbol{x}) = \sigma^{-2}/2\|\boldsymbol{x} - \mathsf{N}_\sigma(\boldsymbol{x})\|^2$, where $\mathsf{N}_\sigma$ is the DRUNet (Zhang et al., 2021). Note that this implementation of $\mathsf{S}$ fulfills all required assumptions; see Proposition 2 in appendix for a formal proof. We use the pretrained model provided in (Hurault et al., 2022a) in our experiments.

**Results.** Theorems 1-2 establish the convergence of RISP to a stationary point with accelerated rate compared with RED. This is illustrated in Figure 2 for the considered linear inverse problems. In the first row, the averaged gradient norm is plotted against the iteration number. It is clearly shown that RISP achieves a faster decay of gradient norm than RED in iteration. Specifically, in the deblurring task, the gradient norm by RISP has decreased by five orders of magnitude within 200 iterations, whereas RED decreases by only three orders.

RISP achieves accelerated convergence without compromising reconstruction quality. The second row of Figure 2 plots PSNR values against iterations for RISP and the baseline methods. These plots shows that RISP achieve comparable PSNR values as RED but in much fewer iterations. For example, RISP reaches the same PSNR roughly five times faster than RED in the MRI experiment. Overall, these results support the theoretical guarantees on acceleration and demonstrate the practical effectivness of RISP across diverse tasks. We additionally provide visual comparisons of the reconstructed images in Appendix G.2.

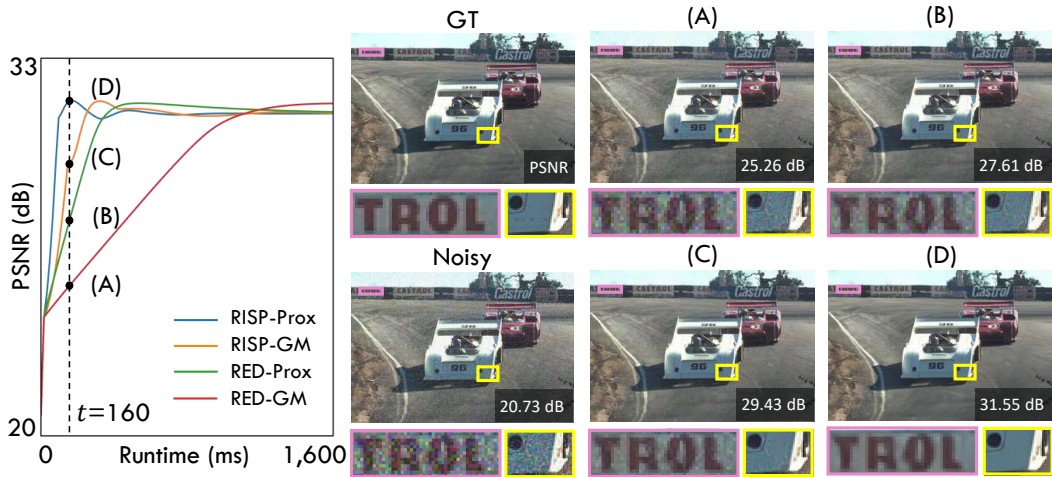

Figure 3: *Left:* Visualization of PSNR curves for RISP and baseline methods on the Rician denoising task with noise level $25.5/255$. The x-axis shows runtime in milliseconds. RISP-Prox reaches 31.55 dB in 160 ms, whereas RED-GM requires roughly ten times longer to achieve comparable performance. *Right:* Visual comparison of reconstructions produced by each algorithm after 160 milliseconds. Zoomed-in regions highlight residual noise and artifacts. Note how RISP-Prox yields a substantially cleaner, nearly noise-free reconstruction within this time budget.

## 5.2 NONLINEAR INVERSE PROBLEM

We further investigate the robustness of RISP on the problem of *Rician noise removal*. Note that the problem is nonlinear, hence leading to a nonconvex data-fidelity term $f$ that violates Assumption 4 required for RISP-Prox. On the other hand, we can show that $f$ still has a Lipschitz continuous gradient and Hessian, namely, satisfy Assumptions 2-3. We refer to Appendix G.3 for formal propositions and additional technical details.

**Experimental setup.** We employ the same score-based prior construction as in the linear problems, thereby satisfying Assumptions 1–3 and validating Theorem 1. We evaluate the methods on the CBSD10 dataset with a Rician noise level of $25.5/255$.

**Results.** Figure 3 compares the reconstruction performance for RISP and the baseline methods. The left panel plots the PSNR curves against runtime in milliseconds (ms), while the right panel visualizes the reconstructions obtained at 160 ms. First, the PSNR-runtime plot demonstrates that RISP-Prox converges even though the data-fidelity term is nonconvex, indicating its robustness beyond ideal settings. Second, both RISP variants achieve high PSNR values substantially faster than RED-GM and RED-Prox. For example, RISP-Prox reaches 31.55 dB in 160 ms, whereas RED-GM requires roughly ten times longer to achieve comparable performance.

Visual comparisons at the 160 ms timestep further corroborate the results. Reconstructions by RED-GM and RED-Prox remain visibly noisy, while RISP-GM produces cleaner images with fewer artifacts, and RISP-Prox yields sharp denoising with well-preserved edges. The zoomed-in regions highlight these visual differences. Overall, these results demonstrate the superior reconstruction quality of RISP under tight time constraints. In the context of Rician denoising, our method nearly reaches the level of real-time processing (the persistence of human vision is around 100 ms).

## 5.3 TIME-EFFICIENCY FOR LARGE-SCALE IMAGING

We finally highlight the acceleration benefit enabled by RISP for reconstructing large-scale images. Here, we consider the *inverse scattering* task that arises in various imaging applications such as tomographic microscopy (Choi et al., 2007), digital holography (Brady et al., 2009; Pellizzari et al., 2020), and radar imaging (Liu et al., 2016). The task aims to reconstruct the permittivity contrast distribution of an object from measurements of its scattered field captured by an array of receivers. We refer the readers to Appendix G.4 for additional technical details on the problem.

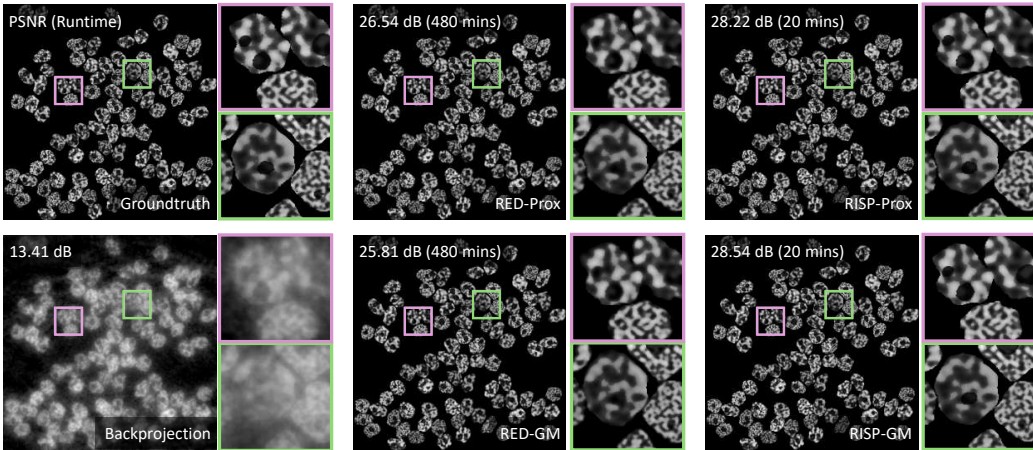

Figure 4: Visual comparison of the reconstructions by RISP and baselines for the inverse scattering task, where the underlying image has a size of $1024 \times 1024$ pixels. All algorithms are ran until convergence or after reaching the maximum runtime (480 minutes). With only 20 minutes, RISP algorithms can restore clear structures and fine details; on the other hand, RED algorithms still cannot provide a comparable result after 480 minutes. Note the substantial difference in the PSNR values and visual differences highlighted in the zoomed-in regions.

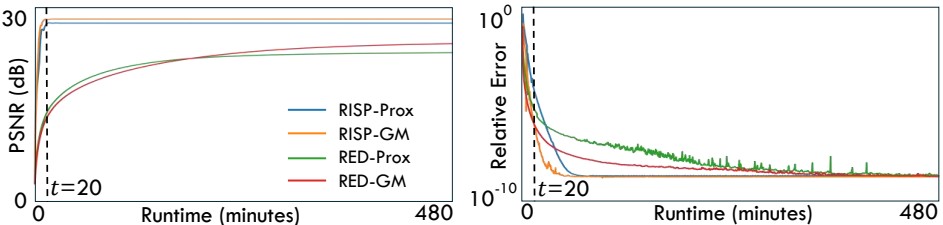

Figure 5: Visualization of the PSNR and relative error curves achieved by RISP and baselines. The relative error is computed by $\|x^{k+1} - x^k\|/\|x^0\|$ and is plotted in the log-scale. The $x$-axis shows runtime in minutes. Note the fast convergence of RISP algorithms to high-quality reconstructions.

**Experimental setup.** The dataset is generated using the CytoPacq Web Service (Wiesner et al., 2019), comprising cell images with a digital resolution of $1024 \times 1024$ pixels. The score prior is implemented using DRUNet (Zhang et al., 2021), trained on 500 cell images of size $384 \times 384$ pixels; we handle the resolution mismatch via patch-based processing. We use 360 receivers and 240 transmitters, and add AWGN with standard deviation $10^{-4}$.

**Results.** Figure 4 visually compares the reconstruction quality obtained by RISP and baseline methods. Since the compressing ratio is $m/d = 360 \times 240/1024^2 \approx 8.2\%$, the problem is severely ill-posed. This is evidenced by the poor reconstruction obtained by the conventional back-projection method. As shown in the zoomed-in regions, The RED algorithms still reconstruct a blurred nuclear envelope despite 480 minutes of runtime. In contrast, both RISP-GM and RISP-Prox recover fine cell-wall structures and detailed features near the nuclei in only 20 minutes. The visual results clearly show that RISP methods can significantly accelerate the reconstruction while maintaining high imaging quality.

Figure 5 demonstrates the fast convergence of RISP by plotting curves of PSNR values and relative errors. As shown, both RISP variants show rapid, stable convergence as well as a steady decrease in relative error. This empirical trend matches the theoretical acceleration established in Theorems 1–2. Furthermore, we observe that the acceleration becomes more pronounced on this large-scale reconstruction task. In particular, RISP-GM and RISP-Prox converge at least twenty-four times faster than RED-GM and RED-Prox. We attribute this speedup to higher per-iteration costs. When each iteration is more expensive, reducing the number of iterations could produce a larger gain in runtime.

## 6 CONCLUSION

In this work, we propose RISP as an extension of the popular RED framework for imaging inverse problems. RISP achieves provably faster convergence while remaining compatible with score-based image priors for high-quality reconstruction. The central component of RISP is the restarted inertia mechanism, which employs inertial updates for acceleration but uses the restarting strategy to prevent instability and overshooting. We theoretically analyze the convergence of RISP, showing that it achieves an accelerated rate of $\mathcal{O}(n^{-4/7})$ over RED. We also provide a continuous-time interpretation that connects RISP with the heavy-ball dynamics and offers complementary insights into the speed-up. Extensive experiments demonstrate that RISP algorithms substantially reduce reconstruction time while preserving competitive image quality, highlighting the theoretical and practical benefits of integrating restarted inertia with score priors.

**Limitations.** Our acceleration guarantees rely on Lipschitz regularity conditions, which excludes some problems such as Poisson denoising. In addition, RISP introduces hyperparameters (*e.g.*, inertial weights $\theta$ and restarting threshold $B$); while our experiments show robustness across wide ranges, some manual tuning may be needed in practice.

## ACKNOWLEDGMENT

This work was supported in part by the ANR project PEPR PDE-AI and the French Direction Générale de l'Armement (DGA). Experiments reported in this paper were carried out using computational resources from the PlaFRIM experimental testbed (https://www.plafrim.fr ), supported by Inria, CNRS (LABRI and IMB), Université de Bordeaux, Bordeaux INP, and the Conseil régional d'Aquitaine, and from the Advanced Research Computing at Hopkins (ARCH) core facility (rockfish.jhu.edu), supported by the National Science Foundation (NSF) under grant OAC-1920103.

## REPRODUCIBILITY STATEMENT

All theoretical proofs are provided in the appendix. Additional technical details on the experiments are also provided in the appendix.

## ETHICS

We do not notice any ethical concerns in this work. We used public data in all experiments. Large language models (LLMs) were used for grammar checking and polishing.

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

# Appendix: Proofs and Additional Experiments

# Contents

## A EXTENDED RELATED WORKS

**Accelerating gradient descent with inertia.** In convex optimization, it is well known that adding an inertial mechanism to gradient descent can accelerate the optimization process (Polyak, 1964; Nesterov, 1983; 2013; Fercoq & Qu, 2019; Su et al., 2016). As many applications involve the minimization of a non-convex function, the study of inertial acceleration in non-convex optimization has attracted significant interest (Apidopoulos et al., 2022; Yue et al., 2023; Danilova et al., 2020; Hinder et al., 2020; Guminov et al., 2023; Nesterov et al., 2021; Wang & Wibisono, 2023; Hermant et al., 2024; Gupta & Wojtowytsch, 2025). Inertial algorithms are often studied on a popular class of non-convex functions, namely functions with Lipschitz continuous gradients. In this case, gradient descent finds a point $\boldsymbol{x} \in \mathbb{R}^d$ such that $\|\nabla f(\boldsymbol{x})\| \leq \varepsilon$ in at most $\mathcal{O}(\varepsilon^{-2})$ iteration. Importantly, only under the gradient Lipschitz assumption, this bound cannot be improved using inertia (Carmon et al., 2020). Going one step further, *i.e.* assuming the Hessian of the function is Lipschitz, acceleration using inertia has been established (Carmon et al., 2017b; Jin et al., 2018; Li & Lin, 2023; Marumo & Takeda, 2024; Okamura et al., 2024). This opens the door to demonstrating the benefits of momentum in applications involving challenging non-convex landscapes.

**Previous works on accelerated RED methods.** Since RED methods are known to be slower than end-to-end networks for image reconstruction, a lot of efforts have been made to make RED methods more efficient. (Sun et al., 2019a) have focused on reducing the computational cost of regularization by implementing a block-coordinate gradient descent. Alternatively, (Tang & Davies, 2020) apply stochastic gradient descent to the data-fidelity term. (Hong et al., 2019) modifies the RED algorithm by incorporating vector extrapolation to accelerate it.

**Inertia in RED methods.** Previous works have proposed to incorporate inertia into the RED framework, not only for acceleration but also for improving restoration quality. (Wu et al., 2024) add inertia to enhance restoration quality in deblurring and super-resolution tasks without analyzing the convergence rate. In a similar line of work, (Chow et al., 2024) focus on improving the quality for Rician noise removal and phase retrieval. (He et al., 2019) study Poisson noise removal and observe an empirical acceleration with inertial methods. However, no theoretical justification is provided for this acceleration.

**Iterative schemes.** Our work proposes to adapt the algorithm in (Li & Lin, 2023) into the RED framework. An extensive line of research has adapted various optimization schemes to solve problem (1). This includes alternating direction method of multipliers (ADMM) (Venkatakrishnan et al., 2013; Ryu et al., 2019), proximal gradient descent (PGD) (Terris et al., 2020; Hurault et al., 2022b), half-quadratic splitting (HQS) (Zhang et al., 2021; Hurault et al., 2022a), stochastic gradient descent (SGD) (Renaud et al., 2024) or Langevin algorithms (Laumont et al., 2022; Sun et al., 2024; Renaud et al., 2025).

**Restarting methods.** To our knowledge, restarted inertia algorithms are first introduced by (O'donoghue & Candes, 2015) in the convex optimization setting. It considers both a fixed restarting criterion, namely restarting every $N$ iterations, and adaptive ones. Specifically, denoting the iterations by $\{x^k\}_{k \geq 0}$, we restart if

$$F(x^{k+1}) > F(x^k) \qquad \text{(Function criterion)}$$

$$\left\langle \nabla F(z^{k-1}), x^k - x^{k-1} \right\rangle > 0. \qquad \text{(Gradient criterion)}$$

The authors observed that these restarting techniques exhibited a clear numerical advantage over vanilla inertial algorithms, but lacked strong theoretical guarantees. Using a continuous version of the Gradient criterion, (Su et al., 2016, Section 5) shows that the restarted ODE associated with the following ODE

$$\ddot{x}_t + \frac{\alpha}{t} \dot{x}_t + \nabla F(x_t) = 0, \qquad (7)$$

converges linearly when $F$ is $\mu$-strongly convex. Without restart, the best achievable convergence rate in this setting is polynomial (Aujol et al., 2019, Proposition 7). This gives a theoretical guarantee of the benefit of restart in the continuous setting. This result of (Su et al., 2016) is extended to more general dynamical systems, restart criteria and geometric assumptions (Maulén & Peypouquet, 2023; Guo et al., 2024; Maulén et al., 2025). In the discrete setting, still for $\mu$-strongly convex functions (or slightly weaker assumptions), provable fast convergence of restarted inertia relies on fixed criteria. This criterion involves the knowledge of $\mu$ (Necoara et al., 2019; Fercoq & Qu, 2019), which can be estimated by algorithms (Alamo et al., 2019; Aujol et al., 2024a; 2025). To our knowledge, it remains to demonstrate that adaptive criteria, such as function or gradient criterion, allows for such convergence speed of the algorithm. Finally, closer to our work, in the context of functions with Lipschitz gradient and Hessian, we note (Li & Lin, 2023; Marumo & Takeda, 2024). In particular, (Li & Lin, 2023) introduced a new restart criterion related to the length of the trajectory, see Section 3. This restart mechanism allowed to recover the convergence result of algorithms that mixed the inertial mechanism with an alternative step that exploits the negative curvature of the function, thus showing that this procedure can be discarded. We generalize their algorithm for composite optimization, where the objective function is the sum of two non-convex function, with RISP-GM (Algorithm 1) and RISP-Prox (Algorithm 2). The latter uses the proximal operator, which provides better experimental results. Finally, note that we focus on the Nesterov inertial mechanism and not on the heavy ball because the heavy-ball method requires a non-trivial averaging step during restart.

## B  PRELIMINARIES

The following lemmas are classical results. We provide their proof for the sake of completeness.

**Lemma 1.** *Let $u : \mathbb{R}^d \to \mathbb{R}$. If $u$ has $L_u$-Lipschitz gradient and has $\rho_u$-Lipschitz Hessian, then for any $\boldsymbol{x}, \boldsymbol{y} \in \mathbb{R}^d$ we have*

*(i)* $u(\boldsymbol{y}) \leq u(\boldsymbol{x}) + \langle \nabla u(\boldsymbol{x}), \boldsymbol{y} - \boldsymbol{x} \rangle + \frac{L_u}{2} \|\boldsymbol{x} - \boldsymbol{y}\|^2,$

*(ii)* $u(\boldsymbol{y}) \leq u(\boldsymbol{x}) + \langle \nabla u(\boldsymbol{x}), \boldsymbol{y} - \boldsymbol{x} \rangle + \frac{1}{2} \langle \nabla u^2(\boldsymbol{x})(\boldsymbol{y} - \boldsymbol{x}), \boldsymbol{y} - \boldsymbol{x} \rangle + \frac{\rho_u}{6} \|\boldsymbol{x} - \boldsymbol{y}\|^3.$

*Proof.* **(i)** We define $\varphi(t) = u(\boldsymbol{x} + t(\boldsymbol{y} - \boldsymbol{x}))$ for $t \in \mathbb{R}$. By the fundamental theorem of calculus, one has

$$u(\boldsymbol{y}) - u(\boldsymbol{x}) = \varphi(1) - \varphi(0) = \int_0^1 \varphi'(s)ds = \int_0^1 \langle \nabla u(\boldsymbol{x} + s(\boldsymbol{y} - \boldsymbol{x})), \boldsymbol{y} - \boldsymbol{x} \rangle\, ds.$$

We subtract on each side $\langle \nabla u(\boldsymbol{x}), \boldsymbol{y} - \boldsymbol{x} \rangle$, such that

$$u(\boldsymbol{y}) - u(\boldsymbol{x}) - \langle \nabla u(\boldsymbol{x}), \boldsymbol{y} - \boldsymbol{x} \rangle \leq \int_0^1 \langle \nabla u(\boldsymbol{x} + s(\boldsymbol{y} - \boldsymbol{x})) - \nabla u(\boldsymbol{x}), \boldsymbol{y} - \boldsymbol{x} \rangle\, ds$$

$$\leq \int_0^1 \|\nabla u(\boldsymbol{x} + s(\boldsymbol{y} - \boldsymbol{x})) - \nabla u(\boldsymbol{x})\| \, \|\boldsymbol{y} - \boldsymbol{x}\|\, ds$$

$$\leq L \int_0^1 \|\boldsymbol{x} + s(\boldsymbol{y} - \boldsymbol{x}) - \boldsymbol{x}\| \, \|\boldsymbol{y} - \boldsymbol{x}\|\, ds$$

$$= L_u \|\boldsymbol{x} - \boldsymbol{y}\|^2 \int_0^1 s\, ds = \frac{L}{2} \|\boldsymbol{x} - \boldsymbol{y}\|^2,$$

where we used Cauchy Schwarz and the $L_u$-Lipschitz property of $\nabla u$.

**(ii)** We define $\varphi(t) = u(\boldsymbol{x} + t(\boldsymbol{y} - \boldsymbol{x}))$ for $t \in \mathbb{R}$. The proof is similar to point (i), except that we use the second order Taylor formula.

$$\varphi(1) = \varphi(0) + \varphi'(0) + \frac{1}{2} \int_0^1 \varphi''(s)ds,$$

which also writes

$$u(\boldsymbol{y}) - u(\boldsymbol{x}) - \langle \nabla u(\boldsymbol{x}), \boldsymbol{y} - \boldsymbol{x} \rangle = \frac{1}{2} \int_0^1 \langle \nabla^2 f(\boldsymbol{x} + s(\boldsymbol{y} - \boldsymbol{x}))(\boldsymbol{y} - \boldsymbol{x}), \boldsymbol{y} - \boldsymbol{x} \rangle\, ds.$$

We then subtract $\frac{1}{2} \langle \nabla^2 f(\boldsymbol{x})(\boldsymbol{y} - \boldsymbol{x}), \boldsymbol{y} - \boldsymbol{x} \rangle$ on each side, use Cauchy-Swcharz, the property of the operator norm and the $\rho_u$ Hessian Lipschitz property to get

$$u(\boldsymbol{y}) - u(\boldsymbol{x}) - \langle \nabla u(\boldsymbol{x}), \boldsymbol{y} - \boldsymbol{x} \rangle - \frac{1}{2} \langle \nabla^2 f(\boldsymbol{x})(\boldsymbol{y} - \boldsymbol{x}), \boldsymbol{y} - \boldsymbol{x} \rangle$$

$$\leq \frac{1}{2} \int_0^1 \langle (\nabla^2 f(\boldsymbol{x} + s(\boldsymbol{y} - \boldsymbol{x})) - \nabla^2 f(\boldsymbol{x}))(\boldsymbol{y} - \boldsymbol{x}), \boldsymbol{y} - \boldsymbol{x} \rangle\, ds$$

$$\leq \frac{1}{2} \int_0^1 \|(\nabla^2 f(\boldsymbol{x} + s(\boldsymbol{y} - \boldsymbol{x})) - \nabla^2 f(\boldsymbol{x}))(\boldsymbol{y} - \boldsymbol{x})\| \, \|\boldsymbol{y} - \boldsymbol{x}\|\, ds$$

$$\leq \frac{1}{2} \int_0^1 s \|\nabla^2 f(\boldsymbol{x} + s(\boldsymbol{y} - \boldsymbol{x})) - \nabla^2 f(\boldsymbol{x})\| \, \|\boldsymbol{y} - \boldsymbol{x}\|^2\, ds$$

$$\leq \frac{\rho_u}{2} \|\boldsymbol{x} - \boldsymbol{y}\|^3 \int_0^1 s^2 ds = \frac{\rho_u}{6} \|\boldsymbol{x} - \boldsymbol{y}\|^3.$$

$\square$

**Lemma 2.** *Let $u : \mathbb{R}^d \to \mathbb{R}$ be $\nu$-weakly convex and differentiable. Then $\forall \boldsymbol{x}, \boldsymbol{y} \in \mathbb{R}^d$,*

$$u(\boldsymbol{y}) \geq u(\boldsymbol{x}) + \langle \nabla u(\boldsymbol{x}), \boldsymbol{y} - \boldsymbol{x} \rangle - \frac{\nu}{2} \|\boldsymbol{x} - \boldsymbol{y}\|^2.$$

*Proof.* By definition of the weak convexity, $f = u + \frac{\nu}{2}\left\|\cdot\right\|^2$ is convex. Then, for all $\boldsymbol{x}, \boldsymbol{y} \in \mathbb{R}^d$, we have

$$f(\boldsymbol{x}) \geq f(\boldsymbol{y}) + \langle \boldsymbol{x} - \boldsymbol{y}, \nabla f(\boldsymbol{y}) \rangle$$

$$u(\boldsymbol{x}) + \frac{\nu}{2}\left\|\boldsymbol{x}\right\|^2 \geq u(\boldsymbol{y}) + \frac{\nu}{2}\left\|\boldsymbol{y}\right\|^2 + \langle \boldsymbol{x} - \boldsymbol{y}, \nabla u(\boldsymbol{y}) + \nu\boldsymbol{y} \rangle$$

$$u(\boldsymbol{y}) \geq u(\boldsymbol{x}) + \langle \nabla u(\boldsymbol{x}), \boldsymbol{y} - \boldsymbol{x} \rangle - \frac{\nu}{2}\left\|\boldsymbol{x} - \boldsymbol{y}\right\|^2.$$

$\square$

## C  CONVERGENCE ANALYSIS OF RED

In this Section, we give a proof of the convergence of (2) stated in Proposition 1. This proof's technique is rather classical (Nesterov, 2013, Section 1.2.3). Proposition 1 adapts it to the RED-GM framework.

**Proposition.** *Under Assumptions 1-2, RED-GM in algorithm* (2) *with $n$ iterations, with $\eta = \frac{1}{L}$, outputs a point $\tilde{\boldsymbol{x}}$ such that*

$$\left\|\nabla F(\tilde{\boldsymbol{x}})\right\| \leq \frac{A_0}{\sqrt{n}} = \mathcal{O}(n^{-1/2}),$$

*with $A_0 = \sqrt{2L(F(\boldsymbol{x}^0) - \min F)}$.*

*Proof.* Recall the iterates of RED-GM

$$\boldsymbol{x}^{k+1} = \boldsymbol{x}^k - \eta\big(\nabla f(\boldsymbol{x}^k) + \tau(\boldsymbol{x} - \mathsf{D}_\sigma(\boldsymbol{x}^k))\big).$$

Because the score function is defined as $\mathsf{S}(\boldsymbol{x}) = -\tau\big(\boldsymbol{x} - \mathsf{D}_\sigma(\boldsymbol{x})\big)$, by Assumption 1, we can write $\tau\big(\boldsymbol{x} - \mathsf{D}_\sigma(\boldsymbol{x})\big) = \nabla g(\boldsymbol{x})$, such that RED-GM becomes

$$\boldsymbol{x}^{k+1} = \boldsymbol{x}^k - \eta\nabla f(\boldsymbol{x}^k) - \eta\nabla g(\boldsymbol{x}^k) = \boldsymbol{x}^k - \eta\nabla F(\boldsymbol{x}^k).$$

Writing the algorithm in this form allows us to apply a classical gradient descent result. For completeness, we detail the remainder of the proof.

By Assumption 2, $F$ is $L := L_f + L_g$ Lipschitz. Hence, Lemma 1-*(i)* can be applied to $u = F$ at $\boldsymbol{x} = \boldsymbol{x}^k$ and $\boldsymbol{y} = \boldsymbol{x}^{k+1}$, yielding

$$F(\boldsymbol{x}^{k+1}) \leq F(\boldsymbol{x}^k) + \langle \nabla F(\boldsymbol{x}^k), \boldsymbol{x}^{k+1} - \boldsymbol{x}^k \rangle + \frac{L}{2}\left\|\boldsymbol{x}^k - \boldsymbol{x}^{k+1}\right\|^2.$$

By definition, $\boldsymbol{x}^{k+1} = \boldsymbol{x}^k - \eta\nabla F(\boldsymbol{x}^k)$, thus

$$F(\boldsymbol{x}^{k+1}) \leq F(\boldsymbol{x}^k) - \eta\left\|\nabla F(\boldsymbol{x}^k)\right\|^2 + \frac{L\eta^2}{2}\left\|\nabla F(\boldsymbol{x}^k)\right\|^2 = F(\boldsymbol{x}^k) - \eta\left(1 - \frac{L\eta}{2}\right)\left\|\nabla F(\boldsymbol{x}^k)\right\|^2.$$

Note that $\left(1 - \frac{L\eta}{2}\right) \geq 0$ as long as $\eta \leq \frac{1}{L}$. Then, rearranging and summing over $k = 0, \dots, n-1$, we get

$$\sum_{k=0}^{n}\left\|\nabla F(\boldsymbol{x}^k)\right\|^2 \leq \frac{F(\boldsymbol{x}^0) - F(\boldsymbol{x}^n)}{\eta\left(1 - \frac{L\eta}{2}\right)} \leq \frac{F(\boldsymbol{x}^0) - \min F}{\eta\left(1 - \frac{L\eta}{2}\right)}.$$

We use the elementary relation $\sum_{k=1}^{n}\left\|\nabla F(\boldsymbol{x}^k)\right\|^2 \geq n\min_{k\in\{1,\dots,n\}}\left\|\nabla F(\boldsymbol{x}^k)\right\|^2$, such that

$$\min_{k\in\{1,\dots,n\}}\left\|\nabla F(\boldsymbol{x}^k)\right\|^2 \leq \frac{1}{n}\frac{F(\boldsymbol{x}^0) - \min F}{\eta\left(1 - \frac{L\eta}{2}\right)}.$$

To conclude, noting $\tilde{\boldsymbol{x}} := \arg\min_{\boldsymbol{x}^k, k\in\{0,\dots,n-1\}}\left\|\nabla F(\boldsymbol{x}^k)\right\|^2$, and fixing $\eta = \frac{1}{L}$, the following holds

$$\left\|\nabla F(\tilde{\boldsymbol{x}})\right\| \leq \frac{1}{\sqrt{n}}\sqrt{2L(F(\boldsymbol{x}^0) - \min F)}.$$

$\square$

Existing works studying gradient descent under Assumptions 2 and 3 does not manage to improve over this convergence speed (Jin et al., 2017). This indicates that assuming the Hessian of the function is Lipschitz does not help to achieve faster convergence when only assuming the gradient is Lipschitz, making the inertial mechanism crucial to do so. We next show that a similar result is obtained when using RED-Prox instead of RED-GM.

**Proposition.** *Under Assumptions 1,2 and 4, RED-PROX in algorithm* (3) *with $n$ iterations, with $\eta = \frac{1}{L}$, outputs a point $\tilde{x}$ such that*

$$\|\nabla F(\tilde{x})\| \leq \frac{2}{\sqrt{n}} \sqrt{2L(F(x^0) - \min F)}.$$

*Proof.* Because the score function is defined as $\mathsf{S}(x) = -\tau(x - \mathsf{D}_\sigma(x))$, by Assumption 1, we can rewrite RED-Prox as a proximal gradient descent algorithm

$$x^{k+1} = \mathsf{prox}_{\eta f}\left(x^k - \tau \cdot \eta(x - \mathsf{D}_\sigma(x^k))\right) \longrightarrow x^{k+1} = \mathsf{prox}_{\eta f}\left(x^k - \eta \nabla g(x^k)\right).$$

Under Assumption 2 and 4, the iterates of this algorithm verify

$$F(x^{k+1}) - F(x^k) \leq -\eta \left(1 - \frac{L\eta}{2}\right) \left\|\frac{1}{\eta}\left(x^{k+1} - x^k\right)\right\|^2,$$

see (Ghadimi et al., 2016, Equation 25). Summing over $k = 0, \ldots n - 1$, we deduce

$$\sum_{k=0}^{n-1} \left\|\frac{1}{\eta}\left(x^{k+1} - x^k\right)\right\|^2 \leq \frac{F(x^0) - F(x^n)}{\eta\left(1 - \frac{L\eta}{2}\right)} \leq \frac{F(x^0) - \min F}{\eta\left(1 - \frac{L\eta}{2}\right)}.$$

We use the elementary relation $\sum_{k=1}^{n} \left\|\frac{1}{\eta}\left(x^{k+1} - x^k\right)\right\|^2 \geq n \min_{k \in \{1, \ldots, n\}} \|\nabla F(x^k)\|^2$, and noting $k_0 := \arg\min_{k \in \{0, \ldots, n-1\}} \left\|\frac{1}{\eta}\left(x^{k+1} - x^k\right)\right\|^2$, we deduce

$$\|x^{k_0+1} - x^{k_0}\| \leq \frac{\eta}{\sqrt{n}} \sqrt{\frac{F(x^0) - \min F}{\eta\left(1 - \frac{L\eta}{2}\right)}}. \tag{8}$$

By the caracterization of the proximal operator, we have

$$\frac{1}{\eta}\left(x^{k_0+1} - x^{k_0}\right) = -\nabla g(x^{k_0}) - \nabla f(x^{k_0+1}). \tag{9}$$

Then, by the smoothness of $f$ (Assumption 2) and (9), we relate the gradient of $F$ to the residuals

$$\begin{aligned}
\|\nabla F(x^{k_0})\| &= \|\nabla g(x^{k_0}) + \nabla f(x^{k_0}) - \nabla f(x^{k_0+1}) + \nabla f(x^{k_0+1})\| \\
&\leq \|\nabla g(x^{k_0}) + \nabla f(x^{k_0+1})\| + \|\nabla f(x^{k_0+1}) - \nabla f(x^{k_0})\| \\
&\leq \left\|\frac{1}{\eta}\left(x^{k_0+1} - x^{k_0}\right)\right\| + L\|x^{k_0+1} - x^{k_0}\| \\
&= \left(\frac{1}{\eta} + L\right)\|x^{k_0+1} - x^{k_0}\|
\end{aligned} \tag{10}$$

Combining (8) and (10), we get

$$\|\nabla F(x^{k_0})\| \leq \left(\frac{1}{\eta} + L\right)^{-1} \frac{\eta}{\sqrt{n}} \sqrt{\frac{F(x^0) - \min F}{\eta\left(1 - \frac{L\eta}{2}\right)}}.$$

Fixing $\eta = \frac{1}{L}$, and noting $\tilde{x} := x^{k_0}$, it becomes

$$\|\nabla F(\tilde{x})\| \leq \frac{2}{\sqrt{n}} \sqrt{2L(F(x^0) - \min F)}.$$

$\square$

Assumption 4 is only used for the convexity of the data fidelity $f$.

## D    CONVERGENCE ANALYSIS OF RISP-GM

In this section, we prove Theorem 1, which establishes the convergence of RISP-GM (Algorithm 1). The key idea is that Assumption 1 allows RISP-GM to be reformulated in a way that permits the application of an optimization result from (Li & Lin, 2023).

**Theorem.** *Let Assumptions 1-3 hold. Let $\eta = 1/(4L)$, $B = \sqrt{\varepsilon/\rho}$, $\theta = 4(\varepsilon\rho\eta^2)^{1/4} \in (0,1)$, and $K = \theta^{-1}$. Then, with at most $n$ iterations, RISP-GM outputs a point $\hat{z}$ such that*

$$\|\nabla F(\hat{z})\| \leq 82\varepsilon = \mathcal{O}(n^{-4/7}),$$

*where $\varepsilon = 2^{4/7}\Delta_F^{4/7}L^{2/7}\rho^{1/7}n^{-4/7} + L^2\rho^{-1}n^{-4}$.*

*Proof.* By Assumption 1, the third line of Algorithm 1 (RISP-GM) writes

$$\boldsymbol{x}^{k+1} = \boldsymbol{z}^k - \eta\nabla f(\boldsymbol{z}^k) - \eta\nabla g(\boldsymbol{y^k}) = \boldsymbol{z}^k - \eta\nabla F(\boldsymbol{z}^k),$$

where $F$ satisfies Assumptions 2 and 3. Then, we can apply (Li & Lin, 2023, Theorem 1) to RISP-GM, which ensures that the algorithm stops in at most $\frac{\Delta_F L^{\frac{1}{2}}\rho^{\frac{1}{4}}}{\varepsilon^{\frac{7}{4}}} + \frac{1}{2}\left(\frac{L^2}{\varepsilon\rho}\right)^{\frac{1}{4}}$ iterations. Moreover, it outputs a point $\hat{z}$ that verifies $\|\nabla F(\hat{z})\| \leq 82\varepsilon$. So, considering a fixed bugdet of $n$ iterations, we fix

$$\varepsilon := 2^{4/7}\Delta_F^{4/7}L^{2/7}\rho^{1/7}n^{-4/7} + L^2\rho^{-1}n^{-4}. \tag{11}$$

We plug (51) in the upper bound of the number of iterations, namely $\frac{\Delta_F L^{\frac{1}{2}}\rho^{\frac{1}{4}}}{\varepsilon^{\frac{7}{4}}} + \frac{1}{2}\left(\frac{L^2}{\varepsilon\rho}\right)^{\frac{1}{4}}$. We have

$$
\begin{aligned}
\frac{\Delta_F L^{\frac{1}{2}}\rho^{\frac{1}{4}}}{\varepsilon^{\frac{7}{4}}} &= \frac{\Delta_F L^{\frac{1}{2}}\rho^{\frac{1}{4}}}{\left(2^{4/7}\Delta_F^{4/7}L^{2/7}\rho^{1/7}n^{-4/7} + L^2\rho^{-1}n^{-4}\right)^{\frac{7}{4}}} \\
&\leq \frac{\Delta_F L^{\frac{1}{2}}\rho^{\frac{1}{4}}}{\left(2^{4/7}\Delta_F^{4/7}L^{2/7}\rho^{1/7}n^{-4/7}\right)^{\frac{7}{4}}} \\
&= \frac{n}{2},
\end{aligned}
\tag{12}
$$

and

$$
\begin{aligned}
\frac{1}{2}\left(\frac{L^2}{\varepsilon\rho}\right)^{\frac{1}{4}} &= \frac{1}{2}\left(\frac{L^2}{\left(2^{4/7}\Delta_F^{4/7}L^{2/7}\rho^{1/7}n^{-4/7} + L^2\rho^{-1}n^{-4}\right)\rho}\right)^{\frac{1}{4}} \\
&\leq \frac{1}{2}\left(\frac{L^2}{L^2\rho^{-1}n^{-4}\rho}\right)^{\frac{1}{4}} \\
&= \frac{n}{2}.
\end{aligned}
\tag{13}
$$

Combining (12) and (13) ensures the algorithm ends at most within the fixed budget of $n$ iterations. Finally, plugging our choice of $\varepsilon$ in the bound $\|\nabla F(\hat{z})\| \leq 82\varepsilon$, we deduce

$$\|\nabla F(\hat{z})\| \leq 82 \cdot 2^{4/7}\Delta_F^{4/7}L^{2/7}\rho^{1/7}n^{-4/7} + 82L^2\rho^{-1}n^{-4}.$$

$\square$

**Remark 1.** *In (Li & Lin, 2023, Theorem 1), it is actually stated that the algorithm stops in at most $\frac{\Delta_F L^{\frac{1}{2}}\rho^{\frac{1}{4}}}{\varepsilon^{\frac{7}{4}}}$ iterations, without the supplementary term $\frac{1}{2}\left(\frac{L^2}{\varepsilon\rho}\right)^{\frac{1}{4}}$ that appears in our proof. We believe this precision has been omitted in their original proof because the main term remains the one of order $\mathcal{O}(\varepsilon^{-7/4})$.*

# E CONVERGENCE ANALYSIS OF RISP-PROX

This section is dedicated to the proof of Theorem 2.

**Theorem.** *Let Assumptions 1-4 hold. Let* $\nu \leq 8(\varepsilon\rho)^{1/4}\sqrt{L}$, $\eta = 1/(8L)$, $B = \sqrt{\varepsilon/(4\rho)}$, $\theta = 4(\varepsilon\rho\eta^2)^{1/4} \in (0,1)$, *and* $K = \theta^{-1}$. *Then, with at most $n$ iterations, RISP-Prox outputs a point $\hat{z}$ such that*

$$\|\nabla F(\hat{z})\| \leq 45\varepsilon = \mathcal{O}(n^{-4/7}),$$

*where* $\varepsilon = \Delta_F^{4/7}(2L)^{2/7}\rho^{1/7}n^{-4/7} + 4L^2\rho^{-1}n^{-4}$.

## E.1 PROOF SKETCH

The overall strategy of proof is adapted from (Li & Lin, 2023, Theorem 1) to analyze algorithm 2. We first prove a reformulation of Theorem 2.

**Theorem 4.** *Under Assumptions 1-4 with* $\nu \leq 8(\varepsilon\rho)^{\frac{1}{4}}\sqrt{L_f + L_g}$, *Algorithm 2 with parameter choice* $\eta = \frac{1}{8(L_f+L_g)}$, $B = \frac{1}{2}\sqrt{\frac{\varepsilon}{\rho}}$, $\theta = 4(\varepsilon\rho\eta^2)^{1/4} \in (0,1)$, *and* $K = \theta^{-1}$ *outputs a point $\hat{z}$ that verify* $\|\nabla F(\hat{z})\| \leq 45\varepsilon$ *in at most* $\frac{\Delta_F L^{\frac{1}{2}}\rho^{\frac{1}{4}}}{\sqrt{2}\varepsilon^{\frac{7}{4}}} + \frac{1}{\sqrt{2}}\left(\frac{L^2}{\varepsilon\rho}\right)^{\frac{1}{4}}$ *iterations.*

The main challenge stems from the presence of an additional proximal operator in Algorithm 2.

First, we prove that if the restarting criterion is reached before the maximum number of iterations $K$, the objective function has a sufficient decrease. This first part of the proof is splitted into two cases, if the gradient mapping at the last iterate before restarting is large (Section E.3) and if the gradient mapping at the last iterate before restarting is small (Section E.4). The case of small gradient is more technical due to the proximal operator and needs the use of the Hessian Lipschitz property.

Then, this sufficient decreasing property allows us to conclude that an $\varepsilon$-stationary point is attained in $\mathcal{O}(\varepsilon^{-\frac{7}{4}})$ iterations.

Moreover, we show that if the restarting criterion is not reached before the maximum number of iterations $K$ then the output of the algorithm is an $\varepsilon$-stationary point (Section E.5).

Finally, we show that Theorem 4 is a reformulation of Theorem 2.

**Remark 2.** *Assuming* $\theta = 4(\varepsilon\rho\eta^2)^{1/4} \in (0,1)$ *in Theorem 2 induces the bound* $\varepsilon < \frac{1}{256\rho\eta^2} = \frac{1}{4}\frac{L^2}{\rho}$.

## E.2 NOTATIONS

As in (Li & Lin, 2023), we consider the restart time

$$\mathcal{K} = \min_k \left\{ k \geq 1 \,\middle|\, k\sum_{t=0}^{k-1}\|\boldsymbol{x}^{t+1} - \boldsymbol{x}^t\|^2 > B^2 \right\}. \tag{14}$$

We call an *epoch* of Algorithm 2 all successive iterates $k \in \{0, \ldots, \mathcal{K}-1\}$. Using Assumption 1, such that $\mathsf{S} = -\nabla g$, we have that inside an epoch, the iterates of Algorithm 2 verify the following recursive formula

$$\begin{cases} \boldsymbol{z}^k = \boldsymbol{x}^k + (1-\theta)(\boldsymbol{x}^k - \boldsymbol{x}^{k-1}) \\ \boldsymbol{x}^{k+1} = \mathsf{prox}_{\eta f}(\boldsymbol{z}^k - \eta\nabla g(\boldsymbol{z}^k)) \end{cases} \tag{15}$$

From the definition of the Proximal operator, the following equation is verified

$$\frac{\boldsymbol{z}^k - \boldsymbol{x}^{k+1}}{\eta} = \nabla f(\boldsymbol{x}^{k+1}) + \nabla g(\boldsymbol{z}^k). \tag{16}$$

By definition of $\mathcal{K}$, we have

$$\mathcal{K} \geq 1, \qquad \mathcal{K}\sum_{t=0}^{\mathcal{K}-1}\|\boldsymbol{x}^{t+1} - \boldsymbol{x}^t\|^2 > B^2. \tag{17}$$

It induces a control on the distance of the $k$-th iterate to the starting point of the epoch $\boldsymbol{x}^0$, $\forall k \in [0, \mathcal{K})$

$$\|\boldsymbol{x}^k - \boldsymbol{x}^0\|^2 = \left\|\sum_{t=0}^{k-1} \boldsymbol{x}^{t+1} - \boldsymbol{x}^t\right\|^2 \leq k \sum_{t=0}^{k-1} \|\boldsymbol{x}^{t+1} - \boldsymbol{x}^t\|^2 \leq B^2, \tag{18}$$

and we have, $\forall k \in [0, \mathcal{K})$

$$\|\boldsymbol{z}^k - \boldsymbol{x}^0\| \leq \|\boldsymbol{x}^k - \boldsymbol{x}^0\| + \|\boldsymbol{x}^k - \boldsymbol{x}^{k-1}\| \leq 2B, \tag{19}$$

where we used that by the definition of $\boldsymbol{z}^k$, one has

$$\left\|\boldsymbol{x}^k - \boldsymbol{z}^k\right\| \leq \left\|\boldsymbol{x}^k - \boldsymbol{x}^{k-1}\right\|. \tag{20}$$

**Gradient mapping.** When considering algorithms of the form (15), it is natural to consider the gradient mapping

$$\boldsymbol{z}^k - \mathrm{prox}_{\eta f}(\boldsymbol{z}^k - \eta \nabla g(\boldsymbol{z}^k)) = \boldsymbol{z}^k - \boldsymbol{x}^{k+1} \overset{(16)}{=} \eta \nabla f(\boldsymbol{x}^{k+1}) + \eta \nabla g(\boldsymbol{z}^k). \tag{21}$$

It can be thought as a generalization of the gradient in the composite case, as if $f = 0$ it reduced to $\eta \nabla g(\boldsymbol{z}^k)$. Importantly, if the gradient mapping is zero, *i.e.* $\boldsymbol{z}^k - \boldsymbol{x}^{k+1} = 0$, then

$$\nabla f(\boldsymbol{x}^{k+1}) + \nabla g(\boldsymbol{z}^k) = \nabla f(\boldsymbol{x}^{k+1}) + \nabla g(\boldsymbol{x}^{k+1}) = \nabla F(\boldsymbol{x}^{k+1}) = 0,$$

*i.e.* we found a stationary point of $F$, which justifies this analogy.

In the two following sections, we consider two case regarding on the value of the gradient mapping at the end of the epoch $\|\boldsymbol{x}^{\mathcal{K}} - \boldsymbol{z}^{\mathcal{K}-1}\|$.

### E.3 LARGE GRADIENT MAPPING IN THE LAST ITERATION OF THE EPOCH

The goal of this section is to prove Corollary 1, which shows that $F$ decrease sufficiently when the gradient mapping $\|\boldsymbol{z}^{\mathcal{K}-1} - \boldsymbol{x}^{\mathcal{K}}\| \geq B$ is large enough.

The following Lemma is a prox version of the classic descent Lemma, see *e.g.* (1.2.19) in (Nesterov, 2013).

**Lemma 3.** *Let $\boldsymbol{x}^k$, $\boldsymbol{z}^k$ be the iterates defined in Equation* (15)*, we have for any $k \geq 1$*

$$F(\boldsymbol{x}^{k+1}) \leq F(\boldsymbol{z}^k) - \left(\frac{1}{\eta} - \frac{3}{2}L\right) \left\|\boldsymbol{x}^{k+1} - \boldsymbol{z}^k\right\|^2,$$

*with $L = L_f + L_g$.*

*Proof.*

$$F(\boldsymbol{x}^{k+1})$$

$$\leq F(\boldsymbol{z}^k) + \left\langle \nabla F(\boldsymbol{z}^k), \boldsymbol{x}^{k+1} - \boldsymbol{z}^k \right\rangle + \frac{L}{2} \left\|\boldsymbol{x}^{k+1} - \boldsymbol{z}^k\right\|^2$$

$$= F(\boldsymbol{z}^k) + \left\langle \nabla g(\boldsymbol{z}^k) + \nabla f(\boldsymbol{z}^k), \boldsymbol{x}^{k+1} - \boldsymbol{z}^k \right\rangle + \frac{L}{2} \left\|\boldsymbol{x}^{k+1} - \boldsymbol{z}^k\right\|^2$$

$$= F(\boldsymbol{z}^k) + \left\langle \nabla g(\boldsymbol{z}^k) + \nabla f(\boldsymbol{x}^{k+1}), \boldsymbol{x}^{k+1} - \boldsymbol{z}^k \right\rangle - \left\langle \nabla f(\boldsymbol{x}^{k+1}) - \nabla f(\boldsymbol{z}^k), \boldsymbol{x}^{k+1} - \boldsymbol{z}^k \right\rangle$$

$$+ \frac{L}{2} \left\|\boldsymbol{x}^{k+1} - \boldsymbol{z}^k\right\|^2$$

$$= F(\boldsymbol{z}^k) + \left\langle -\frac{\boldsymbol{x}^{k+1} - \boldsymbol{z}^k}{\eta}, \boldsymbol{x}^{k+1} - \boldsymbol{z}^k \right\rangle - \left\langle \nabla f(\boldsymbol{x}^{k+1}) - \nabla f(\boldsymbol{z}^k), \boldsymbol{x}^{k+1} - \boldsymbol{z}^k \right\rangle$$

$$+ \frac{L}{2} \left\|\boldsymbol{x}^{k+1} - \boldsymbol{z}^k\right\|^2,$$

where we use (16) in the last equality. Using Cauchy-Schwarz and the fact that $\nabla f$ is $L$-Lipschitz, we have

$$- \left\langle \nabla f(\boldsymbol{x}^{k+1}) - \nabla f(\boldsymbol{z}^k), \boldsymbol{x}^{k+1} - \boldsymbol{z}^k \right\rangle \leq \left\|\nabla f(\boldsymbol{x}^{k+1}) - \nabla f(\boldsymbol{z}^k)\right\| \left\|\boldsymbol{x}^{k+1} - \boldsymbol{z}^k\right\|$$

$$\leq L \left\|\boldsymbol{x}^{k+1} - \boldsymbol{z}^k\right\|^2.$$

Thus we obtain the desired inequality

$$F(\boldsymbol{x}^{k+1}) \leq F(\boldsymbol{z}^k) - \left(\frac{1}{\eta} - \frac{3}{2}L\right) \left\|\boldsymbol{x}^{k+1} - \boldsymbol{z}^k\right\|^2.$$

$\square$

The following Lemma indicates that the decrease of $F$ at the last iteration of the current epoch can be quantified using the cumulated norms of gradient mapping along these iterates, and the cumulated distance between iterates along this epoch.

**Lemma 4.** *We have*

$$F(\boldsymbol{x}^{\mathcal{K}}) - F(\boldsymbol{x}^0) \leq \frac{1}{2}\left(\frac{1}{\eta} + L\right)\sum_{k=0}^{\mathcal{K}-1}\left\|\boldsymbol{x}^{k+1} - \boldsymbol{x}^k\right\|^2 - \left(\frac{1}{2\eta} - \frac{\eta L_f^2}{2} - \frac{3}{2}L\right)\sum_{k=0}^{\mathcal{K}-1}\left\|\boldsymbol{z}^k - \boldsymbol{x}^{k+1}\right\|^2.$$

*Proof.* By Lemma 3, we have

$$F(\boldsymbol{x}^{k+1}) \leq F(\boldsymbol{z}^k) - \left(\frac{1}{\eta} - \frac{3}{2}L\right)\left\|\boldsymbol{x}^{k+1} - \boldsymbol{z}^k\right\|^2. \tag{22}$$

As $F$ is $L$-smooth, we have

$$F(\boldsymbol{x}^k) \geq F(\boldsymbol{z}^k) + \left\langle \nabla F(\boldsymbol{z}^k), \boldsymbol{x}^k - \boldsymbol{z}^k \right\rangle - \frac{L}{2}\left\|\boldsymbol{z}^k - \boldsymbol{x}^k\right\|^2. \tag{23}$$

Summing Equations (22) and (23) we get

$$F(\boldsymbol{x}^{k+1}) - F(\boldsymbol{x}^k) \leq -\left\langle \nabla F(\boldsymbol{z}^k), \boldsymbol{x}^k - \boldsymbol{z}^k \right\rangle + \frac{L}{2}\left\|\boldsymbol{z}^k - \boldsymbol{x}^k\right\|^2 - \left(\frac{1}{\eta} - \frac{3}{2}L\right)\left\|\boldsymbol{z}^k - \boldsymbol{x}^{k+1}\right\|^2. \tag{24}$$

From the characterization of Equation (16), making $\nabla F(\boldsymbol{z}^k)$ appear we have

$$\nabla F(\boldsymbol{z}^k) = \frac{\boldsymbol{z}^k - \boldsymbol{x}^{k+1}}{\eta} + \nabla f(\boldsymbol{z}^k) - \nabla f(\boldsymbol{x}^{k+1}). \tag{25}$$

Combining Equations (24) and (25), we have

$$\begin{aligned}
F(\boldsymbol{x}^{k+1}) - F(\boldsymbol{x}^k) &\leq -\left\langle \frac{\boldsymbol{z}^k - \boldsymbol{x}^{k+1}}{\eta} + \nabla f(\boldsymbol{z}^k) - \nabla f(\boldsymbol{x}^{k+1}), \boldsymbol{x}^k - \boldsymbol{z}^k \right\rangle \\
&\quad + \frac{L}{2}\left\|\boldsymbol{z}^k - \boldsymbol{x}^k\right\|^2 - \left(\frac{1}{\eta} - \frac{3}{2}L\right)\left\|\boldsymbol{z}^k - \boldsymbol{x}^{k+1}\right\|^2 \\
&= \frac{1}{\eta}\left\langle \boldsymbol{x}^{k+1} - \boldsymbol{z}^k, \boldsymbol{x}^k - \boldsymbol{z}^k \right\rangle + \left\langle \nabla f(\boldsymbol{x}^{k+1}) - \nabla f(\boldsymbol{z}^k), \boldsymbol{x}^k - \boldsymbol{z}^k \right\rangle \\
&\quad + \frac{L}{2}\left\|\boldsymbol{z}^k - \boldsymbol{x}^k\right\|^2 - \left(\frac{1}{\eta} - \frac{3}{2}L\right)\left\|\boldsymbol{z}^k - \boldsymbol{x}^{k+1}\right\|^2. 
\end{aligned} \tag{26}$$

We have the algebraic identity

$$\left\langle \boldsymbol{x}^{k+1} - \boldsymbol{z}^k, \boldsymbol{x}^k - \boldsymbol{z}^k \right\rangle = \frac{1}{2}\left(\left\|\boldsymbol{x}^{k+1} - \boldsymbol{z}^k\right\|^2 + \left\|\boldsymbol{x}^k - \boldsymbol{z}^k\right\|^2 - \left\|\boldsymbol{x}^{k+1} - \boldsymbol{x}^k\right\|^2\right). \tag{27}$$

Moreover for some $\lambda > 0$, the $L_f$-Lipschitz continuity of $\nabla f$ gives

$$\begin{aligned}
&\left\langle \nabla f(\boldsymbol{x}^{k+1}) - \nabla f(\boldsymbol{z}^k), \boldsymbol{x}^k - \boldsymbol{z}^k \right\rangle \\
&\leq \frac{\lambda}{2}\left\|\nabla f(\boldsymbol{x}^{k+1}) - \nabla f(\boldsymbol{z}^k)\right\|^2 + \frac{1}{2\lambda}\left\|\boldsymbol{x}^k - \boldsymbol{z}^k\right\|^2 \leq \frac{\lambda L_f^2}{2}\left\|\boldsymbol{x}^{k+1} - \boldsymbol{z}^k\right\|^2 + \frac{1}{2\lambda}\left\|\boldsymbol{x}^k - \boldsymbol{z}^k\right\|^2.
\end{aligned} \tag{28}$$

By injecting Equations (27) and (28) into Equation (26), we get

$$F(\boldsymbol{x}^{k+1}) - F(\boldsymbol{x}^k)$$

$$\leq \frac{1}{2\eta}\left(\left\|\boldsymbol{x}^{k+1} - \boldsymbol{z}^k\right\|^2 + \left\|\boldsymbol{x}^k - \boldsymbol{z}^k\right\|^2 - \left\|\boldsymbol{x}^{k+1} - \boldsymbol{x}^k\right\|^2\right) + \frac{\lambda L_f^2}{2}\left\|\boldsymbol{x}^{k+1} - \boldsymbol{z}^k\right\|^2$$

$$+ \frac{1}{2\lambda}\left\|\boldsymbol{x}^k - \boldsymbol{z}^k\right\|^2 + \frac{L}{2}\left\|\boldsymbol{z}^k - \boldsymbol{x}^k\right\|^2 - \left(\frac{1}{\eta} - \frac{3}{2}L\right)\left\|\boldsymbol{z}^k - \boldsymbol{x}^{k+1}\right\|^2$$

$$= -\frac{1}{2\eta}\left\|\boldsymbol{x}^{k+1} - \boldsymbol{x}^k\right\|^2 + \frac{1}{2}\left(\frac{1}{\eta} + \frac{1}{\lambda} + L\right)\left\|\boldsymbol{x}^k - \boldsymbol{z}^k\right\|^2$$

$$- \left(\frac{1}{2\eta} - \frac{\lambda L_f^2}{2} - \frac{3}{2}L\right)\left\|\boldsymbol{z}^k - \boldsymbol{x}^{k+1}\right\|^2 .$$

By using Equation (20), we get

$$F(\boldsymbol{x}^{k+1}) - F(\boldsymbol{x}^k)$$

$$\leq -\frac{1}{2\eta}\left\|\boldsymbol{x}^{k+1} - \boldsymbol{x}^k\right\|^2 + \frac{1}{2}\left(\frac{1}{\eta} + \frac{1}{\lambda} + L\right)\left\|\boldsymbol{x}^k - \boldsymbol{x}^{k-1}\right\|^2$$

$$- \left(\frac{1}{2\eta} - \frac{\lambda L_f^2}{2} - \frac{3}{2}L\right)\left\|\boldsymbol{z}^k - \boldsymbol{x}^{k+1}\right\|^2$$

$$= -\frac{1}{2\eta}\left\|\boldsymbol{x}^{k+1} - \boldsymbol{x}^k\right\|^2 + \frac{1}{2}\left(\frac{1}{\eta} + 2L\right)\left\|\boldsymbol{x}^k - \boldsymbol{x}^{k-1}\right\|^2 - \left(\frac{1}{2\eta} - \frac{L_f^2}{2L} - \frac{3}{2}L\right)\left\|\boldsymbol{z}^k - \boldsymbol{x}^{k+1}\right\|^2 .$$

where we fix $\lambda = \frac{1}{L}$ in the last equality. Using $\boldsymbol{x}^0 = \boldsymbol{x}^{-1}$, we sum on $k = 0, \ldots \mathcal{K} - 1$

$$F(\boldsymbol{x}^{\mathcal{K}}) - F(\boldsymbol{x}^0) \leq L \sum_{k=0}^{\mathcal{K}-1}\left(-\frac{1}{2\eta}\left\|\boldsymbol{x}^{k+1} - \boldsymbol{x}^k\right\|^2 + \frac{1}{2}\left(\frac{1}{\eta} + 2L\right)\left\|\boldsymbol{x}^k - \boldsymbol{x}^{k-1}\right\|^2\right)$$

$$- \left(\frac{1}{2\eta} - \frac{\eta L_f^2}{2} - \frac{3}{2}L\right)\sum_{k=1}^{\mathcal{K}-1}\left\|\boldsymbol{z}^k - \boldsymbol{x}^{k+1}\right\|^2$$

$$= \sum_{k=0}^{\mathcal{K}-2}\left(-\frac{1}{2\eta}\left\|\boldsymbol{x}^{k+1} - \boldsymbol{x}^k\right\|^2 + \frac{1}{2}\left(\frac{1}{\eta} + 2L\right)\left\|\boldsymbol{x}^{k+1} - \boldsymbol{x}^k\right\|^2\right) + \frac{1}{2}\left(\frac{1}{\eta} + 2L\right)\left\|\boldsymbol{x}^0 - \boldsymbol{x}^{-1}\right\|$$

$$- \frac{1}{2\eta}\left\|\boldsymbol{x}^{\mathcal{K}} - \boldsymbol{x}^{\mathcal{K}-1}\right\| - \left(\frac{1}{2\eta} - \frac{\eta L_f^2}{2} - \frac{3}{2}L\right)\sum_{k=0}^{\mathcal{K}-1}\left\|\boldsymbol{z}^k - \boldsymbol{x}^{k+1}\right\|^2$$

$$\leq L \sum_{k=0}^{\mathcal{K}-2}\left\|\boldsymbol{x}^{k+1} - \boldsymbol{x}^k\right\|^2 - \left(\frac{1}{2\eta} - \frac{\eta L_f^2}{2} - \frac{3}{2}L\right)\sum_{k=0}^{\mathcal{K}-1}\left\|\boldsymbol{z}^k - \boldsymbol{x}^{k+1}\right\|^2$$

$$\square$$

Using Lemma 4, we use the restart criterion ((14)) to ensure the decrease of $F$ in the case of sufficiently large gradient mapping at the last iterate of the epoch.

**Corollary 1.** *If the gradient mapping is large* $\left\|\boldsymbol{z}^{\mathcal{K}-1} - \boldsymbol{x}^{\mathcal{K}}\right\| \geq B$*, then*

$$F(\boldsymbol{x}^{\mathcal{K}}) - F(\boldsymbol{x}^0) \leq -\frac{B^2}{8\eta}.$$

*Proof.* By Lemma 4, we have

$$F(\boldsymbol{x}^{\mathcal{K}}) - F(\boldsymbol{x}^0) \leq L \sum_{k=0}^{\mathcal{K}-2}\left\|\boldsymbol{x}^{k+1} - \boldsymbol{x}^k\right\|^2 - \left(\frac{1}{2\eta} - \frac{\eta L_f^2}{2} - \frac{3}{2}L\right)\sum_{k=0}^{\mathcal{K}-1}\left\|\boldsymbol{z}^k - \boldsymbol{x}^{k+1}\right\|^2$$

$$\leq L \sum_{k=0}^{\mathcal{K}-2}\left\|\boldsymbol{x}^{k+1} - \boldsymbol{x}^k\right\|^2 - \left(\frac{1}{2\eta} - \frac{\eta L_f^2}{2} - \frac{3}{2}L\right)\left\|\boldsymbol{z}^{\mathcal{K}-1} - \boldsymbol{x}^{\mathcal{K}}\right\|^2$$

If $\mathcal{K} = 1$, $\sum_{k=0}^{\mathcal{K}-2} \left\| \boldsymbol{x}^{k+1} - \boldsymbol{x}^k \right\|^2$ is an empty sum, equal to zero. Otherwise, one has

$$F(\boldsymbol{x}^{\mathcal{K}}) - F(\boldsymbol{x}^0) \leq L\frac{B^2}{\mathcal{K}-1} - \left( \frac{1}{2\eta} - \frac{\eta L_f^2}{2} - \frac{3}{2}L \right) B^2$$

$$\leq LB^2 - \left( \frac{1}{2\eta} - \frac{\eta L_f^2}{2} - \frac{3}{2}L \right) B^2$$

$$= \left( L + \frac{\eta L_f^2}{2} + \frac{3}{2}L - \frac{1}{2\eta} \right) B^2$$

$$\leq \left( L + \frac{\eta L_f^2}{2} + \frac{3}{2}L - \frac{1}{2\eta} \right) B^2$$

$$\leq -\frac{B^2}{8\eta},$$

where we used $\eta \leq \frac{1}{8L}$. $\qquad\qquad\square$

**Remark 3.** *Note that to obtain Corollary 1, it is not necessary that the large gradient mapping occur in the last iterate. This choice of iterate will however be important in the next section, in order to control the distance between iterates at the beginning and end of the epoch, see (29).*

### E.4 SMALL GRADIENT MAPPING FOR THE LAST ITERATE OF THE EPOCH

The goal of this section is to prove Corollary 2, which shows that $F$ decrease sufficiently when $\left\| \boldsymbol{z}^{\mathcal{K}-1} - \boldsymbol{x}^{\mathcal{K}} \right\| \leq B$. Authors of (Li & Lin, 2023) use the Hessian Lipschitz Assumption to approximate the function by a quadratic function, and the fact that its Hessian matrix is diagonal up to a change of basis

In our case considering a sum of two functions and a proximal operator, the choice of the basis must be made carefully.

**Preliminary.** If the gradient mapping $\|\boldsymbol{x}^{\mathcal{K}} - \boldsymbol{z}^{\mathcal{K}-1}\| \leq B$, then from Equation (19) we have

$$\|\boldsymbol{x}^{\mathcal{K}} - \boldsymbol{x}^0\| \leq \|\boldsymbol{z}^{\mathcal{K}-1} - \boldsymbol{x}^0\| + \|\boldsymbol{x}^{\mathcal{K}} - \boldsymbol{z}^{\mathcal{K}-1}\| \leq 3B. \tag{29}$$

For each epoch, we denote $\mathbf{H}_g = \nabla^2 g(\boldsymbol{x}^0)$ to be the Hessian matrix at the starting iterate $\boldsymbol{x}^0$ and $\mathbf{H}_g = \mathbf{U}\boldsymbol{\Lambda}_g \mathbf{U}^T$ to be its eigenvalue decomposition with $\mathbf{U}, \boldsymbol{\Lambda}_g \in \mathbb{R}^{d \times d}$, $\mathbf{U}^T\mathbf{U} = \mathsf{I}$ and $\boldsymbol{\Lambda}_g$ being diagonal. We denote $\lambda_j$ the $j$th eigenvalue and $\widetilde{\boldsymbol{x}} = \mathbf{U}^T\boldsymbol{x}$, $\widetilde{\boldsymbol{z}} = \mathbf{U}^T\boldsymbol{y}$, and $\widetilde{\nabla} f(\boldsymbol{y}) = \mathbf{U}^T\nabla f(\boldsymbol{y})$ the different objects in the basis of diagonalization for $\mathbf{H}_g$.

By Lemma 1 (ii), we have

$$g(\boldsymbol{x}^{\mathcal{K}}) - g(\boldsymbol{x}^0) \leq \langle \nabla g(\boldsymbol{x}^0), \boldsymbol{x}^{\mathcal{K}} - \boldsymbol{x}^0 \rangle + \frac{1}{2}(\boldsymbol{x}^{\mathcal{K}} - \boldsymbol{x}^0)^T \mathbf{H}_g(\boldsymbol{x}^{\mathcal{K}} - \boldsymbol{x}^0) + \frac{\rho_g}{6} \|\boldsymbol{x}^{\mathcal{K}} - \boldsymbol{x}^0\|^3$$

$$= \left\langle \widetilde{\nabla} g(\boldsymbol{x}^0), \widetilde{\boldsymbol{x}}^{\mathcal{K}} - \widetilde{\boldsymbol{x}}^0 \right\rangle + \frac{1}{2}(\widetilde{\boldsymbol{x}}^{\mathcal{K}} - \widetilde{\boldsymbol{x}}^0)^T \boldsymbol{\Lambda}_g(\widetilde{\boldsymbol{x}}^{\mathcal{K}} - \widetilde{\boldsymbol{x}}^0) + \frac{\rho_g}{6} \|\boldsymbol{x}^{\mathcal{K}} - \boldsymbol{x}^0\|^3 \tag{30}$$

$$\leq \hat{g}(\widetilde{\boldsymbol{x}}^{\mathcal{K}}) - \hat{g}(\widetilde{\boldsymbol{x}}^0) + \frac{9}{2}\rho_g B^3,$$

where we used Equation (29), and we denote for $x \in \mathbb{R}^d$, $x_j$ the $j$-th component of $\boldsymbol{x}$ and $t \in \mathbb{R}$

$$\hat{g}(\boldsymbol{x}) = \left\langle \widetilde{\nabla} g(\boldsymbol{x}^0), \boldsymbol{x} - \widetilde{\boldsymbol{x}}^0 \right\rangle + \frac{1}{2}(\boldsymbol{x} - \widetilde{\boldsymbol{x}}^0)^T \boldsymbol{\Lambda}_g(\boldsymbol{x} - \widetilde{\boldsymbol{x}}^0) = \sum_{j=1}^d \hat{g}_j(x_j)$$

$$\hat{g}_j(t) = \widetilde{\nabla}_j g(\boldsymbol{x}^0)(t - \widetilde{x}_j^0) + \frac{1}{2}\lambda_j(t - \widetilde{x}_j^0)^2. \tag{31}$$

By following the same change of basis for the function $f$, we get

$$f(\boldsymbol{x}^{\mathcal{K}}) - f(\boldsymbol{x}^0) \leq \left\langle \nabla f(\boldsymbol{x}^0), \boldsymbol{x}^{\mathcal{K}} - \boldsymbol{x}^0 \right\rangle + \frac{1}{2}(\boldsymbol{x}^{\mathcal{K}} - \boldsymbol{x}^0)^T \mathbf{H}_f(\boldsymbol{x}^{\mathcal{K}} - \boldsymbol{x}^0) + \frac{\rho_f}{6}\|\boldsymbol{x}^{\mathcal{K}} - \boldsymbol{x}^0\|^3$$

$$= \left\langle \widetilde{\nabla} f(\boldsymbol{x}^0), \widetilde{\boldsymbol{x}}^{\mathcal{K}} - \widetilde{\boldsymbol{x}}^0 \right\rangle + \frac{1}{2}(\widetilde{\boldsymbol{x}}^{\mathcal{K}} - \widetilde{\boldsymbol{x}}^0)^T \mathbf{U}^T \mathbf{H}_f \mathbf{U}(\widetilde{\boldsymbol{x}}^{\mathcal{K}} - \widetilde{\boldsymbol{x}}^0) + \frac{\rho_f}{6}\|\boldsymbol{x}^{\mathcal{K}} - \boldsymbol{x}^0\|^3 \qquad (32)$$

$$\leq \hat{f}(\widetilde{\boldsymbol{x}}^{\mathcal{K}}) - \hat{f}(\widetilde{\boldsymbol{x}}^0) + \frac{9}{2}\rho_f B^3,$$

where we used Equation (29), and we denote for $x \in \mathbb{R}^d$

$$\hat{f}(x) = \left\langle \widetilde{\nabla} f(\boldsymbol{x}^0), x - \widetilde{\boldsymbol{x}}^0 \right\rangle + \frac{1}{2}(x - \widetilde{\boldsymbol{x}}^0)^T \mathbf{U}^T \mathbf{H}_f \mathbf{U}(x - \widetilde{\boldsymbol{x}}^0) \qquad (33)$$

By adding Equation (30) and Equation (32), we have

$$F(\boldsymbol{x}^{\mathcal{K}}) - F(\boldsymbol{x}^0) \leq \hat{F}(\widetilde{\boldsymbol{x}}^{\mathcal{K}}) - \hat{F}(\widetilde{\boldsymbol{x}}^0) + \frac{9}{2}\rho B^3, \qquad (34)$$

where we used Equation (29) and $\hat{F} = \hat{g} + \hat{f}$.

**Writting $\widetilde{\boldsymbol{x}}^{k+1}, \widetilde{\boldsymbol{z}}^k$ as an inexact forward backward algorithm on $\hat{F}$.** We show now that the iterates $\widetilde{\boldsymbol{x}}^{k+1}, \widetilde{\boldsymbol{z}}^k$ in the basis of diagonalization of $\mathbf{H}_g$ follow a inexact forward backward algorithm on the quadratic function $\hat{F} = \hat{g} + \hat{f}$, up to an error term.

By applying the change of basis in equation (15), we get

$$\widetilde{\boldsymbol{x}}^{k+1} = \widetilde{\boldsymbol{z}}^k - \eta \mathbf{U}^T \nabla g(\boldsymbol{z}^k) - \eta \mathbf{U}^T \nabla f(\boldsymbol{x}^{k+1}) \qquad (35)$$

$$= \widetilde{\boldsymbol{z}}^k - \eta \nabla \hat{g}(\widetilde{\boldsymbol{z}}^k) - \eta \nabla \hat{f}(\widetilde{\boldsymbol{x}}^{k+1}) + \eta \widetilde{\delta}^k, \qquad (36)$$

where

$$\widetilde{\delta}^k = \nabla \hat{g}(\widetilde{\boldsymbol{z}}^k) + \nabla \hat{f}(\widetilde{\boldsymbol{x}}^{k+1}) - \mathbf{U}^T \nabla g(\boldsymbol{z}^k) - \mathbf{U}^T \nabla f(\boldsymbol{x}^{k+1}).$$

Then, Algorithm 15 can be written as

$$\begin{cases} \widetilde{\boldsymbol{z}}^k = \widetilde{\boldsymbol{x}}^k + (1 - \theta)(\widetilde{\boldsymbol{x}}^k - \widetilde{\boldsymbol{x}}^{k-1}) \\ \widetilde{\boldsymbol{x}}^{k+1} = \mathsf{prox}_{\eta\hat{f}}(\widetilde{\boldsymbol{z}}^k - \eta \nabla \hat{g}(\widetilde{\boldsymbol{z}}^k) + \eta \widetilde{\delta}^k) \end{cases} \qquad (37)$$

To ensure good properties of the iterates of (37), it is necessary to controll the norm of the error $\left\|\widetilde{\delta}^k\right\|$, as done in the following.

**Upper bound on $\left\|\widetilde{\delta}^k\right\|$.** The norm of the error term $\widetilde{\delta}^k$ can be controlled using the Hessian Lipschitz assumption.

$$\left\|\widetilde{\delta}^k\right\| = \left\|\nabla \hat{g}(\widetilde{\boldsymbol{z}}^k) + \nabla \hat{f}(\widetilde{\boldsymbol{x}}^{k+1}) - \mathbf{U}^T \nabla g(\boldsymbol{z}^k) - \mathbf{U}^T \nabla f(\boldsymbol{x}^{k+1})\right\|$$

$$\leq \left\|\nabla \hat{g}(\widetilde{\boldsymbol{z}}^k) - \mathbf{U}^T \nabla g(\boldsymbol{z}^k)\right\| + \left\|\nabla \hat{f}(\widetilde{\boldsymbol{x}}^{k+1}) - \mathbf{U}^T \nabla f(\boldsymbol{x}^{k+1})\right\|.$$

For $k \in [0, \mathcal{K})$, using the Hessian of $g$ is $\rho_g$ Lipschitz, the fact that $\|\boldsymbol{U}^T \boldsymbol{x}\| = \|\boldsymbol{x}\|$ for $\boldsymbol{x} \in \mathbb{R}^d$ and equation (19), we have

$$\left\|\nabla \hat{g}(\widetilde{\boldsymbol{z}}^k) - \mathbf{U}^t \nabla g(\boldsymbol{z}^k)\right\| = \left\|\widetilde{\nabla} g(\boldsymbol{x}^0) + \boldsymbol{\Lambda}_g(\widetilde{\boldsymbol{z}}^k - \widetilde{\boldsymbol{x}}^0) - \mathbf{U}^T \nabla g(\boldsymbol{z}^k)\right\|$$

$$= \left\|\nabla g(\boldsymbol{x}^0) + \mathbf{H}_g(\boldsymbol{z}^k - \boldsymbol{x}^0) - \nabla g(\boldsymbol{z}^k)\right\|$$

$$= \left\|\left(\int_0^1 \nabla^2 g(\boldsymbol{x}^0 + t(\boldsymbol{z}^k - \boldsymbol{x}^0)) - \mathbf{H}_f\right)(\boldsymbol{z}^k - \boldsymbol{x}^0)dt\right\|$$

$$\leq \frac{\rho_g}{2}\left\|\boldsymbol{x}^0 - \boldsymbol{z}^k\right\|^2 \leq 2\rho_g B^2, \qquad (38)$$

Similarly on $f$, using equation (29), we get

$$
\begin{aligned}
\left\| \nabla \hat{f}(\widetilde{\boldsymbol{x}}^{k+1}) - \mathbf{U}^t \nabla f(\boldsymbol{x}^{k+1}) \right\| &= \left\| \widetilde{\nabla} f(\boldsymbol{x}^0) + \mathbf{U}^T \mathbf{H}_f \mathbf{U}(\widetilde{\boldsymbol{x}}^{k+1} - \widetilde{\boldsymbol{x}}^0) - \mathbf{U}^T \nabla f(\boldsymbol{x}^{k+1}) \right\| \\
&= \left\| \nabla f(\boldsymbol{x}^0) + \mathbf{H}_f(\boldsymbol{x}^{k+1} - \boldsymbol{x}^0) - \nabla f(\boldsymbol{x}^{k+1}) \right\| \\
&= \left\| \left( \int_0^1 \nabla^2 f(\boldsymbol{x}^0 + t(\boldsymbol{x}^{k+1} - \boldsymbol{x}^0)) - \mathbf{H}_f \right)(\boldsymbol{x}^{k+1} - \boldsymbol{x}^0) dt \right\| \\
&\leq \frac{\rho_f}{2} \left\| \boldsymbol{x}^0 - \boldsymbol{x}^{k+1} \right\|^2 \leq \frac{9}{2}\rho_f B^2.
\end{aligned}
\tag{39}
$$

By combining equation (39) and equation (38), we have

$$
\left\| \widetilde{\delta}^k \right\| \leq \left( 2\rho_g + \frac{9}{2}\rho_f \right) B^2 \leq \frac{9}{2}\rho B^2,
\tag{40}
$$

where $\rho := \rho_f + \rho_g$.

In the following Lemma, we use the control on $\left\| \widetilde{\delta}^k \right\|$ to upper bound the decrease of $\hat{F}$ between the first and last iterate of the current epoch. We will make use of the fact that $\hat{g}$ is separable, thanks to the change of basis.

**Lemma 5.** *Assume $H_g \succeq -\frac{\theta}{\eta}\mathsf{I} = -8(\varepsilon\rho)^{\frac{1}{4}}\sqrt{L}\mathsf{I}$. We have*

$$
\hat{F}(\widetilde{\boldsymbol{x}}^{\mathcal{K}}) \leq \hat{F}(\widetilde{\boldsymbol{x}}^0) - \frac{3\theta}{8\eta} \sum_{k=0}^{\mathcal{K}-1} \left\| \widetilde{\boldsymbol{x}}^{k+1} - \widetilde{\boldsymbol{x}}^k \right\|^2 + \frac{8\eta\rho^2 B^4 \mathcal{K}}{\theta}.
$$

*Proof.* Since $\hat{g}_j(x)$, defined in equation (31), is quadratic, we have

$$
\hat{g}_j(\widetilde{x}_j^{k+1}) = \hat{g}_j(\widetilde{x}_j^k) + \left\langle \nabla \hat{g}_j(\widetilde{x}_j^k), \widetilde{x}_j^{k+1} - \widetilde{x}_j^k \right\rangle + \frac{\lambda_j}{2}|\widetilde{x}_j^{k+1} - \widetilde{x}_j^k|^2
$$

Convexity of $f$ induces convexity of $\hat{f}$, such that

$$
\hat{f}(\widetilde{\boldsymbol{x}}^{k+1}) \leq \hat{f}(\widetilde{\boldsymbol{x}}^k) + \left\langle \nabla \hat{f}(\widetilde{\boldsymbol{x}}^{k+1}), \widetilde{\boldsymbol{x}}^{k+1} - \widetilde{\boldsymbol{x}}^k \right\rangle.
$$

Then, summing over $j$, using the separability of $\hat{g}$ we have

$$
\hat{f}(\widetilde{\boldsymbol{x}}^{k+1}) + \sum_j \hat{g}_j(\widetilde{x}_j^{k+1}) = \hat{f}(\widetilde{\boldsymbol{x}}^{k+1}) + \hat{g}(\widetilde{\boldsymbol{x}}^{k+1}) = \hat{F}(\widetilde{\boldsymbol{x}}^{k+1}),
$$

and

$$
\begin{aligned}
\sum_j &\left( \hat{g}_j(\widetilde{x}_j^k) + \left\langle \nabla \hat{g}_j(\widetilde{x}_j^k), \widetilde{x}_j^{k+1} - \widetilde{x}_j^k \right\rangle + \frac{\lambda_j}{2}|\widetilde{x}_j^{k+1} - \widetilde{x}_j^k|^2 \right) \\
&= \hat{g}(\widetilde{\boldsymbol{x}}^k) + \left\langle \nabla \hat{g}(\widetilde{\boldsymbol{x}}^k), \widetilde{\boldsymbol{x}}^k - \widetilde{\boldsymbol{x}}^{k+1} \right\rangle + \frac{1}{2}\sum_j \lambda_j |\widetilde{x}_j^{k+1} - \widetilde{x}_j^k|^2.
\end{aligned}
$$

Combining the three previous equations, we get

$$
\begin{aligned}
\hat{F}(\widetilde{\boldsymbol{x}}^{k+1}) \leq &\hat{g}(\widetilde{\boldsymbol{x}}^k) + \left\langle \nabla \hat{g}(\widetilde{\boldsymbol{x}}^k), \widetilde{\boldsymbol{x}}^{k+1} - \widetilde{\boldsymbol{x}}^k \right\rangle + \frac{1}{2}\sum_j \lambda_j |\widetilde{x}_j^{k+1} - \widetilde{x}_j^k|^2 \\
&+ \hat{f}(\widetilde{\boldsymbol{x}}^k) + \left\langle \nabla \hat{f}(\widetilde{\boldsymbol{x}}^{k+1}), \widetilde{\boldsymbol{x}}^{k+1} - \widetilde{\boldsymbol{x}}^k \right\rangle \\
= &\hat{F}(\widetilde{\boldsymbol{x}}^k) + \left\langle \nabla \hat{g}(\widetilde{\boldsymbol{x}}^k) - \nabla \hat{g}(\widetilde{\boldsymbol{z}}^k), \widetilde{\boldsymbol{x}}^{k+1} - \widetilde{\boldsymbol{x}}^k \right\rangle + \frac{1}{2}\sum_j \lambda_j |\widetilde{x}_j^{k+1} - \widetilde{x}_j^k|^2 \\
&+ \left\langle \nabla \hat{g}(\widetilde{\boldsymbol{z}}^k) + \nabla \hat{f}(\widetilde{\boldsymbol{x}}^{k+1}), \widetilde{\boldsymbol{x}}^{k+1} - \widetilde{\boldsymbol{x}}^k \right\rangle.
\end{aligned}
$$

However, by (36), we have

$$\nabla \hat{g}(\widetilde{\boldsymbol{z}}^k) + \nabla \hat{f}(\widetilde{\boldsymbol{x}}^{k+1}) = -\frac{1}{\eta}\left(\widetilde{\boldsymbol{x}}^{k+1} - \widetilde{\boldsymbol{z}}^k - \eta\widetilde{\delta}^k\right),$$

from which we deduce

$$\hat{F}(\widetilde{\boldsymbol{x}}^{k+1}) \leq \hat{F}(\widetilde{\boldsymbol{x}}^k) + \left\langle \nabla\hat{g}(\widetilde{\boldsymbol{x}}^k) - \nabla\hat{g}(\widetilde{\boldsymbol{z}}^k), \widetilde{\boldsymbol{x}}^{k+1} - \widetilde{\boldsymbol{x}}^k \right\rangle + \frac{1}{2}\sum_j \lambda_j |\widetilde{x}_j^{k+1} - \widetilde{x}_j^k|^2$$

$$- \frac{1}{\eta}\left\langle \widetilde{\boldsymbol{x}}^{k+1} - \widetilde{\boldsymbol{z}}^k - \eta\widetilde{\delta}^k, \widetilde{\boldsymbol{x}}^{k+1} - \widetilde{\boldsymbol{x}}^k \right\rangle.$$

In the expression above, only $\hat{F}$ is not separable. So we have

$$\hat{F}(\widetilde{\boldsymbol{x}}^{k+1}) - \hat{F}(\widetilde{\boldsymbol{x}}^k)$$

$$\leq \sum_j \left( (\nabla\hat{g}_j(\widetilde{x}_j^k) - \nabla\hat{g}_j(\widetilde{z}_j^k))(\widetilde{x}_j^{k+1} - \widetilde{x}_j^k) + \frac{1}{2}\lambda_j |\widetilde{x}_j^{k+1} - \widetilde{x}_j^k|^2 \right. \tag{41}$$

$$\left. -\frac{1}{\eta}(\widetilde{x}_j^{k+1} - \widetilde{z}_j^k - \eta\widetilde{\delta}_j^k)(\widetilde{x}_j^{k+1} - \widetilde{x}_j^k) \right) \tag{42}$$

As $\nabla\hat{g}_j(\widetilde{x}_j^k) - \nabla\hat{g}_j(\widetilde{z}_j^k) = \lambda_j(\widetilde{x}_j^k - \widetilde{z}_j^k)$, we consider

$$-\frac{1}{\eta}(\widetilde{x}_j^{k+1} - \widetilde{z}_j^k)(\widetilde{x}_j^{k+1} - \widetilde{x}_j^k) + \widetilde{\delta}_j^k(\widetilde{x}_j^{k+1} - \widetilde{x}_j^k) + \lambda_j(\widetilde{x}_j^k - \widetilde{z}_j^k)(\widetilde{x}_j^{k+1} - \widetilde{x}_j^k) + \frac{\lambda_j}{2}|\widetilde{x}_j^{k+1} - \widetilde{x}_j^k|^2$$

$$= \frac{1}{2\eta}\left(|\widetilde{x}_j^k - \widetilde{z}_j^k|^2 - |\widetilde{x}_j^{k+1} - \widetilde{z}_j^k|^2 - |\widetilde{x}_j^{k+1} - \widetilde{x}_j^k|^2\right) + \widetilde{\delta}_j^k(\widetilde{x}_j^{k+1} - \widetilde{x}_j^k)$$

$$+ \frac{\lambda_j}{2}\left(|\widetilde{x}_j^{k+1} - \widetilde{z}_j^k|^2 - |\widetilde{x}_j^k - \widetilde{z}_j^k|^2\right)$$

$$\leq \frac{1}{2\eta}\left(|\widetilde{x}_j^k - \widetilde{z}_j^k|^2 - |\widetilde{x}_j^{k+1} - \widetilde{z}_j^k|^2 - |\widetilde{x}_j^{k+1} - \widetilde{x}_j^k|^2\right)$$

$$+ \frac{1}{2\alpha}|\widetilde{\delta}_j^k|^2 + \frac{\alpha}{2}|\widetilde{x}_j^{k+1} - \widetilde{x}_j^k|^2 + \frac{\lambda_j}{2}\left(|\widetilde{x}_j^{k+1} - \widetilde{z}_j^k|^2 - |\widetilde{x}_j^k - \widetilde{z}_j^k|^2\right)$$

for some positive constant $\alpha$ to be specified later, where we use (36) in the first equality.

As we assumed $L\mathsf{I} \succeq H_g \succeq -\frac{\theta}{\eta}\mathsf{I}$, for all $j \in \{1, \ldots, d\}$ $L \geq \lambda_j \geq -\frac{\theta}{\eta}$. If $\eta \leq \frac{1}{8L}$, we get that $\left(-\frac{1}{2\eta} + \frac{\lambda_j}{2}\right)|\widetilde{x}_j^{k+1} - \widetilde{z}_j^k|^2 \leq \left(-4L + \frac{L}{2}\right)|\widetilde{x}_j^{k+1} - \widetilde{z}_j^k|^2 \leq 0$. By injecting these properties into equation (41), we have

$$\hat{F}(\widetilde{\boldsymbol{x}}^{k+1})$$

$$\leq \hat{F}(\widetilde{\boldsymbol{x}}^k) + \sum_j \left( (\nabla\hat{g}_j(\widetilde{x}_j^k) - \nabla\hat{g}_j(\widetilde{z}_j^k))(\widetilde{x}_j^{k+1} - \widetilde{x}_j^k) + \frac{1}{2}\lambda_j|\widetilde{x}_j^{k+1} - \widetilde{x}_j^k|^2 \right)$$

$$- \sum_j \left( \frac{1}{\eta}(\widetilde{x}_j^{k+1} - \widetilde{z}_j^k - \eta\widetilde{\delta}_j^k)(\widetilde{x}_j^{k+1} - \widetilde{x}_j^k) \right)$$

$$\leq \hat{F}(\widetilde{\boldsymbol{x}}^k) + \sum_j \left( \frac{1}{2\eta}\left(|\widetilde{x}_j^k - \widetilde{z}_j^k|^2 - |\widetilde{x}_j^{k+1} - \widetilde{x}_j^k|^2\right) + \frac{1}{2\alpha}|\widetilde{\delta}_j^k|^2 + \frac{\alpha}{2}|\widetilde{x}_j^{k+1} - \widetilde{x}_j^k|^2 + \frac{\theta}{2\eta}|\widetilde{x}_j^k - \widetilde{z}_j^k|^2 \right)$$

$$\leq \hat{F}(\widetilde{\boldsymbol{x}}^k) + \sum_j \left( \frac{(1+\theta)(1-\theta)^2}{2\eta}|\widetilde{x}_j^k - \widetilde{x}_j^{k-1}|^2 - \left(\frac{1}{2\eta} - \frac{\alpha}{2}\right)|\widetilde{x}_j^{k+1} - \widetilde{x}_j^k|^2 + \frac{1}{2\alpha}|\widetilde{\delta}_j^k|^2 \right)$$

$$= \hat{F}(\widetilde{\boldsymbol{x}}^k) + \sum_j \left( \frac{(1+\theta)(1-\theta)^2}{2\eta}|\widetilde{x}_j^k - \widetilde{x}_j^{k-1}|^2 - \left(\frac{1}{2\eta} - \frac{\alpha}{2}\right)|\widetilde{x}_j^{k+1} - \widetilde{x}_j^k|^2 + \frac{1}{2\alpha}|\widetilde{\delta}_j^k|^2 \right).$$

We have, using $\boldsymbol{x}^{-1} = \boldsymbol{x}^0$,

$$\sum_{k=0}^{\mathcal{K}-1} \left( \frac{(1+\theta)(1-\theta)^2}{2\eta} |\widetilde{x}_j^k - \widetilde{x}_j^{k-1}|^2 - \left( \frac{1}{2\eta} - \frac{\alpha}{2} \right) |\widetilde{x}_j^{k+1} - \widetilde{x}_j^k|^2 \right)$$

$$\leq -\sum_{k=0}^{\mathcal{K}-1} \left( \frac{1}{2\eta} - \frac{\alpha}{2} - \frac{(1+\theta)(1-\theta)^2}{2\eta} \right) |\widetilde{x}_j^{k+1} - \widetilde{x}_j^k|^2$$

$$\leq -\frac{3\theta}{8\eta} \sum_{k=0}^{\mathcal{K}-1} |\widetilde{x}_j^{k+1} - \widetilde{x}_j^k|^2$$

where we let $\alpha = \frac{\theta}{4\eta}$ in the last inequality, such that

$$\frac{1}{2\eta} - \frac{\theta}{8\eta} - \frac{(1+\theta)(1-\theta)^2}{2\eta} = \frac{3\theta}{8\eta} + \frac{\theta^2}{2\eta} - \frac{\theta^3}{2\eta} \geq \frac{3\theta}{8\eta}.$$

So, summing on $k = 0, \ldots, \mathcal{K}-1$ we have

$$\hat{F}(\widetilde{\boldsymbol{x}}^{\mathcal{K}}) \leq \hat{F}(\widetilde{\boldsymbol{x}}^0) + \sum_j \left( -\frac{3\theta}{8\eta} \sum_{k=0}^{\mathcal{K}-1} |\widetilde{x}_j^{k+1} - \widetilde{x}_j^k|^2 + \frac{2\eta}{\theta} \sum_{k=0}^{\mathcal{K}-1} \left\| \widetilde{\delta}_j^k \right\|^2 \right)$$

$$\overset{d}{\leq} \hat{F}(\widetilde{\boldsymbol{x}}^0) - \frac{3\theta}{8\eta} \sum_{k=0}^{\mathcal{K}-1} \left\| \widetilde{\boldsymbol{x}}^{k+1} - \widetilde{\boldsymbol{x}}^k \right\|^2 + \frac{81\eta\rho^2 B^4 \mathcal{K}}{2\theta}$$

where we use (40) in $\overset{d}{\leq}$. $\qquad \square$

**Lemma 6.** *Assuming* $\left\| \boldsymbol{z}^{\mathcal{K}-1} - \boldsymbol{x}^{\mathcal{K}} \right\| \leq B$, *we have*

$$F(\boldsymbol{x}^{\mathcal{K}}) - F(\boldsymbol{x}^0) \leq -\frac{3\theta B^2}{8\eta K} + \frac{81\eta\rho^2 B^4 K}{2\theta} + \frac{9}{2}\rho B^3.$$

*Proof.* By Lemma 5, we have

$$\hat{F}(\widetilde{\boldsymbol{x}}^{\mathcal{K}}) - \hat{F}(\widetilde{\boldsymbol{x}}^0) \leq -\frac{3\theta}{8\eta} \sum_{k=0}^{\mathcal{K}-1} \left\| \widetilde{\boldsymbol{x}}^{k+1} - \widetilde{\boldsymbol{x}}^k \right\|^2 + \frac{81\eta\rho^2 B^4 \mathcal{K}}{2\theta}$$

$$= -\frac{3\theta}{8\eta} \sum_{k=0}^{\mathcal{K}-1} \left\| \boldsymbol{x}^{k+1} - \boldsymbol{x}^k \right\|^2 + \frac{81\eta\rho^2 B^4 \mathcal{K}}{2\theta}$$

Using (34), we get

$$F(\boldsymbol{x}^{\mathcal{K}}) - F(\boldsymbol{x}^0) \leq -\frac{3\theta}{8\eta} \sum_{k=0}^{\mathcal{K}-1} \left\| \widetilde{x}_j^{k+1} - \widetilde{x}_j^k \right\|^2 + \frac{81\eta\rho^2 B^4 \mathcal{K}}{2\theta} + \frac{9}{2}\rho B^3$$

$$\leq -\frac{3\theta B^2}{8\eta \mathcal{K}} + \frac{81\eta\rho^2 B^4 \mathcal{K}}{2\theta} + \frac{9}{2}\rho B^3$$

$$\leq -\frac{3\theta B^2}{8\eta K} + \frac{81\eta\rho^2 B^4 K}{2\theta} + \frac{9}{2}\rho B^3,$$

where the last inequality uses $\mathcal{K} \leq K$. $\qquad \square$

**Corollary 2.** *If the restart criterion is reached at* $\mathcal{K} < K$, *assuming* $\varepsilon \leq \frac{1}{64\rho\eta^2}$, *we have*

$$F(\boldsymbol{x}^{\mathcal{K}}) - F(\boldsymbol{x}^0) \leq -\frac{\varepsilon^{\frac{3}{2}}}{\sqrt{\rho}}.$$

*Proof.* Combining Corollary 1 and Lemma 6, and using parameter choice $\eta = \frac{1}{8L}$, $B = \frac{1}{2}\sqrt{\frac{\varepsilon}{\rho}}$, $\theta = 4(\varepsilon\rho\eta^2)^{1/4} \in (0, 1)$, and $K = \frac{1}{\theta}$, we have

$$\frac{3\theta B^2}{8\eta K} = \frac{3}{8\eta}\theta^2 B^2 = \frac{3}{2}\frac{\varepsilon^{\frac{3}{2}}}{\sqrt{\rho}},$$

$$\frac{81\eta\rho^2 B^4 K}{2\theta} = \frac{81}{2\theta^2}\eta\rho^2 B^4 = \frac{3^4}{2^9}\frac{\varepsilon^{\frac{3}{2}}}{\sqrt{\rho}},$$

$$\frac{9}{2}\rho B^3 = \frac{3^2}{2^4}\frac{\varepsilon^{\frac{3}{2}}}{\sqrt{\rho}},$$

$$\frac{B^2}{8\eta} = \frac{\varepsilon}{2^5\eta\rho}.$$

Such that

$$F(\boldsymbol{x}^{\mathcal{K}}) - F(\boldsymbol{x}^0) \leq -\min\left\{\frac{3\theta B^2}{8\eta K} - \frac{81\eta\rho^2 B^4 K}{2\theta} - \frac{9}{2}\rho B^3, \frac{B^2}{8\eta}\right\}$$

$$\leq -\frac{1}{2}\min\left\{\frac{\varepsilon^{\frac{3}{2}}}{\sqrt{\rho}}, \frac{\varepsilon}{16\eta\rho}\right\}.$$

Assuming $\varepsilon \leq \frac{1}{64\rho\eta^2}$ we have $\frac{\varepsilon^{\frac{3}{2}}}{\sqrt{\rho}} \leq \frac{\varepsilon}{16\eta\rho}$. This condition is verified as long as $\theta \leq 1$, such that $\varepsilon \leq \frac{1}{256\rho\eta^2}$. $\square$

### E.5   WHEN THE RESTARTING IS NOT TRIGGERED IN THE $K$ FIRST ITERATIONS

**Lemma 7.** *If the restart criterion of Algorithm 2 is not reached, we have*
$$\|\nabla F(\hat{\boldsymbol{z}})\| \leq 45\varepsilon.$$

*Proof.* The proof follows three steps. First, we bound the quantity $\|\nabla g(\hat{\boldsymbol{z}}) + \nabla f(\hat{\boldsymbol{x}})\|$. The second step bound the quantity $\|\hat{\boldsymbol{x}} - \hat{\boldsymbol{z}}\|$, which allows to bound the term $\|\nabla F(\hat{\boldsymbol{z}})\|$.

**(i) Bounding** $\|\nabla g(\hat{\boldsymbol{z}}) + \nabla f(\hat{\boldsymbol{x}})\|$. Denote $\widetilde{\boldsymbol{z}} = \mathbf{U}^T\hat{\boldsymbol{z}} = \frac{1}{K_0+1}\sum_{k=0}^{K_0}\mathbf{U}^T\boldsymbol{z}^k = \frac{1}{K_0+1}\sum_{k=0}^{K_0}\widetilde{\boldsymbol{z}}^k$ and $\widetilde{\boldsymbol{x}} = \mathbf{U}^T\hat{\boldsymbol{x}} = \frac{1}{K_0+1}\sum_{k=0}^{K_0}\mathbf{U}^T\boldsymbol{x}^{k+1} = \frac{1}{K_0+1}\sum_{k=0}^{K_0}\widetilde{\boldsymbol{x}}^{k+1}$. Since, $\hat{g}$ and $\hat{f}$ are quadratic and $\boldsymbol{x}^{-1} = \boldsymbol{x}^0$, we get

$$\left\|\nabla\hat{g}(\widetilde{y}) + \nabla\hat{f}(\widetilde{x})\right\| = \left\|\frac{1}{K_0+1}\sum_{k=0}^{K_0}\nabla\hat{g}(\widetilde{\boldsymbol{z}}^k) + \nabla\hat{f}(\widetilde{\boldsymbol{x}}^{k+1})\right\|$$

$$= \frac{1}{\eta(K_0+1)}\left\|\sum_{k=0}^{K_0}(\widetilde{\boldsymbol{x}}^{k+1} - \widetilde{\boldsymbol{z}}^k + \eta\widetilde{\delta}^k)\right\| \tag{43}$$

$$= \frac{1}{\eta(K_0+1)}\left\|\sum_{k=0}^{K_0}\left(\widetilde{\boldsymbol{x}}^{k+1} - \widetilde{\boldsymbol{x}}^k - (1-\theta)(\widetilde{\boldsymbol{x}}^k - \widetilde{\boldsymbol{x}}^{k-1}) + \eta\widetilde{\delta}^k\right)\right\|$$

$$= \frac{1}{\eta(K_0+1)}\left\|\widetilde{\boldsymbol{x}}^{K_0+1} - \widetilde{\boldsymbol{x}}^0 - (1-\theta)(\widetilde{\boldsymbol{x}}^{K_0} - \widetilde{\boldsymbol{x}}^0) + \eta\sum_{k=0}^{K_0}\widetilde{\delta}^k\right\|$$

$$= \frac{1}{\eta(K_0+1)}\left\|\widetilde{\boldsymbol{x}}^{K_0+1} - \widetilde{\boldsymbol{x}}^{K_0} + \theta(\widetilde{\boldsymbol{x}}^{K_0} - \widetilde{\boldsymbol{x}}^0) + \eta\sum_{k=0}^{K_0}\widetilde{\delta}^k\right\|$$

$$\leq \frac{1}{\eta(K_0+1)}\left(\|\widetilde{\boldsymbol{x}}^{K_0+1} - \widetilde{\boldsymbol{x}}^{K_0}\| + \theta\|\widetilde{\boldsymbol{x}}^{K_0} - \widetilde{\boldsymbol{x}}^0\| + \eta\sum_{k=0}^{K_0}\|\widetilde{\delta}^k\|\right)$$

$$\leq \frac{2}{\eta K}\|\boldsymbol{x}^{k_0+1} - \boldsymbol{x}^{k_0}\| + \frac{2\theta B}{\eta K} + 2\rho B^2, \tag{44}$$

Where we used in the last inequality the fact that $K_0 = \arg\min_{\lfloor \frac{K}{2} \rfloor \leq k \leq K-1} \|x^{k+1} - x^k\| \leq K-1$, equation (18) and equation (40).

Moreover, the fact that $K_0 = \arg\min_{\lfloor \frac{K}{2} \rfloor \leq k \leq K-1} \|x^{k+1} - x^k\|$ gives

$$
\begin{aligned}
\|x^{k_0+1} - x^{k_0}\|^2 &\leq \frac{1}{K - \lfloor K/2 \rfloor} \sum_{k=\lfloor K/2 \rfloor}^{K-1} \|x^{k+1} - x^k\|^2 \\
&\leq \frac{1}{K - \lfloor K/2 \rfloor} \sum_{k=0}^{K-1} \|x^{k+1} - x^k\|^2 \\
&\leq \frac{1}{K - \lfloor K/2 \rfloor} \frac{B^2}{K} \leq \frac{2B^2}{K^2},
\end{aligned}
\tag{45}
$$

where we use in the last inequality the fact that the restarting criterion is not reached at iteration $K$.

We can now obtain a control on $\|\nabla g(\hat{z}) + \nabla f(\hat{x})\|$

$$
\begin{aligned}
\|\nabla g(\hat{z}) + \nabla f(\hat{x})\| &= \left\| \widetilde{\nabla} g(\hat{z}) + \widetilde{\nabla} f(\hat{x}) \right\| \\
&\leq \left\| \nabla \hat{g}(\widetilde{z}) + \nabla \hat{f}(\widetilde{x}) \right\| + \left\| \widetilde{\nabla} g(\hat{z}) - \nabla \hat{g}(\widetilde{z}) \right\| + \left\| \widetilde{\nabla} f(\hat{x}) - \nabla \hat{f}(\widetilde{x}) \right\|.
\end{aligned}
\tag{46}
$$

Using Hessian Lipschitz properties, we already showed in (38)-(39) that we have

$$
\left\| \widetilde{\nabla} g(\hat{z}) - \nabla \hat{g}(\widetilde{z}) \right\| \leq \frac{\rho_g}{2} \left\| \hat{z} - x^0 \right\|^2
\tag{47}
$$

$$
\left\| \widetilde{\nabla} f(\hat{x}) - \nabla \hat{f}(\widetilde{x}) \right\| \leq \frac{\rho_f}{2} \left\| \hat{x} - x^0 \right\|^2.
\tag{48}
$$

As the restart criterion did not activate for all $k < K$, we have $\left\| \hat{x} - x^0 \right\| \leq \frac{1}{K_0+1} \sum_{k=0}^{K_0} \left\| x^{k+1} - x^0 \right\| \leq B$ and $\left\| \hat{z} - x^0 \right\| \leq \frac{1}{K_0+1} \sum_{k=0}^{K_0} \left\| z^k - x^0 \right\| \leq 2B$. By injecting (44), (45), (47) and (48) into (46), we get

$$
\|\nabla g(\hat{z}) + \nabla f(\hat{x})\| \leq \frac{2\sqrt{2}B}{\eta K^2} + \frac{2\theta B}{\eta K} + 2\rho B^2 + 2\rho B^2.
$$

Recalling the parameter choice $\eta = \frac{1}{8L}$, $B = \frac{1}{2}\sqrt{\frac{\varepsilon}{\rho}}$, $\theta = 4(\varepsilon\rho\eta^2)^{1/4} \in (0,1)$, and $K = \frac{1}{\theta}$, we have

$$
\begin{aligned}
\frac{2\sqrt{2}B}{\eta K^2} &= \frac{2\sqrt{2}\sqrt{\varepsilon}}{\eta\sqrt{\rho}} \theta^2 = 16\sqrt{2}\varepsilon \\
\frac{2\theta B}{\eta K} &= \frac{2\theta^2 B}{\eta} = 16\varepsilon \\
2\rho B^2 &= \frac{\varepsilon}{2},
\end{aligned}
$$

such that

$$
\|\nabla g(\hat{z}) + \nabla f(\hat{x})\| \leq (\sqrt{2}16 + 16 + 1)\varepsilon \leq 40\varepsilon.
\tag{49}
$$

**(ii) Bounding $\|\hat{x} - \hat{z}\|$.** Starting from (43), removing the $\widetilde{\delta}^k$ term insider the norm and multiplying b $\eta$, the exact same computations can be derived to get

$$
\begin{aligned}
\|\hat{x} - \hat{z}\| = \frac{1}{K_0+1} \left\| \sum_{k=0}^{K_0} \widetilde{x}^{k+1} - \widetilde{z}^k \right\| &\leq \frac{1}{K_0+1} \left( \|\widetilde{x}^{K_0+1} - \widetilde{x}^{K_0}\| + \theta\|\widetilde{x}^{K_0} - \widetilde{x}^0\| \right) \\
&\leq \frac{2}{K} \|\widetilde{x}^{K_0+1} - \widetilde{x}^{K_0}\| + \frac{2\theta B}{K} \\
&\leq \frac{2\sqrt{2}B}{K^2} + \frac{2\theta B}{K},
\end{aligned}
$$

where we used $K_0 \geq \frac{K}{2}$. Recalling the parameter choice $\eta = \frac{1}{8L}$, $B = \frac{1}{2}\sqrt{\frac{\varepsilon}{\rho}}$, $\theta = 4(\varepsilon\rho\eta^2)^{1/4} \in (0,1)$, and $K = \frac{1}{\theta}$, we have

$$\frac{2\sqrt{2}B}{K^2} = 2\sqrt{2}\theta^2 B = 16\sqrt{2}\eta\varepsilon$$

$$\frac{2\theta B}{K} = 2\theta^2 B = 16\eta\varepsilon,$$

such that

$$\|\hat{\boldsymbol{x}} - \hat{\boldsymbol{z}}\| \leq (16 + 16\sqrt{2})\eta\varepsilon \leq 39\eta\varepsilon. \tag{50}$$

**(iii) Bounding** $\|\nabla F(\hat{\boldsymbol{z}})\|$**.** Note that we have

$$\|\nabla F(\hat{\boldsymbol{z}})\| = \|\nabla f(\hat{\boldsymbol{z}}) + \nabla g(\hat{\boldsymbol{z}})\| = \|\nabla f(\hat{\boldsymbol{z}}) - \nabla f(\hat{\boldsymbol{x}}) + \nabla f(\hat{\boldsymbol{x}}) + \nabla g(\hat{\boldsymbol{z}})\|.$$

Using the triangular inequality, we get

$$\|\nabla F(\hat{\boldsymbol{z}})\| \leq \|\nabla f(\hat{\boldsymbol{z}}) - \nabla f(\hat{\boldsymbol{x}})\| + \|\nabla f(\hat{\boldsymbol{x}}) + \nabla g(\hat{\boldsymbol{z}})\|$$
$$\leq L_f \|\hat{\boldsymbol{z}} - \hat{\boldsymbol{x}}\| + \|\nabla f(\hat{\boldsymbol{x}}) + \nabla g(\hat{\boldsymbol{z}})\|,$$

where the last inequality used the $L_f$-smooth property of $f$. To conclude, we use (49)-(50) and $\eta = \frac{1}{8L}$

$$\|\nabla F(\hat{\boldsymbol{z}})\| \leq L_f 39\eta\varepsilon + 40\varepsilon$$
$$\leq \left(\frac{39}{8} + 40\right)\varepsilon$$
$$\leq 45\varepsilon.$$

$\square$

### E.6 PROOF OF THEOREMS 2 AND 4

We prove Theorem 4 first. Theorem 2 is a corollary of Theorem 4.

**Proof of Theorem 4.** For each epoch such that the restart criterion is met, we have by Corollary 2

$$F(\boldsymbol{x}^{\mathcal{K}}) - F(\boldsymbol{x}^0) \leq -\frac{\varepsilon^{\frac{3}{2}}}{\sqrt{\rho}}.$$

Using that $\boldsymbol{x}^0$ is set to be $\boldsymbol{x}^{\mathcal{K}}$ of the previous epoch, and that $\min F \leq f(\boldsymbol{x}^{\mathcal{K}})$ for any epoch, summing over $N$ epochs we get

$$\min F - F(\boldsymbol{x}^0) \leq -N\frac{\varepsilon^{\frac{3}{2}}}{\sqrt{\rho}},$$

where $\boldsymbol{x}^{\text{init}}$ is the starting point $\boldsymbol{x}^0$ of the first epoch. So, noting $\Delta_F := F(\boldsymbol{x}^0) - \min F$, we deduce that the number of epochs $N$ such that the restart criterion is met is upper bounded

$$N \leq \frac{\Delta_F \sqrt{\rho}}{\varepsilon^{\frac{3}{2}}},$$

leading to a total number of epoch upper-bounded by $\frac{\Delta_F \sqrt{\rho}}{\varepsilon^{\frac{3}{2}}} + 1$. Since each epoch requires at most $K = \frac{1}{4(\varepsilon\rho\eta^2)^{1/4}} = \frac{1}{\sqrt{2}}\left(\frac{L^2}{\varepsilon\rho}\right)^{\frac{1}{4}}$, we then deduce there is at most $\frac{\Delta_F L^{\frac{1}{2}}\rho^{\frac{1}{4}}}{\sqrt{2}\varepsilon^{\frac{7}{4}}} + \frac{1}{\sqrt{2}}\left(\frac{L^2}{\varepsilon\rho}\right)^{\frac{1}{4}}$ gradient iterations needed before the algorithm stops and output a point $\hat{\boldsymbol{z}}$. Lemma 7 ensures that this output verifies $\|\nabla F(\hat{\boldsymbol{z}})\| \leq 45\varepsilon$, which concludes the proof of Theorem 4.

**Proof of Theorem 2.** Assume we fix a budget of $n$ iterations. We fix

$$\varepsilon := \frac{\Delta_F^{\frac{4}{7}}(2L)^{\frac{2}{7}}\rho^{\frac{1}{7}}}{n^{\frac{4}{7}}} + \frac{4L^2}{\rho n^4}. \tag{51}$$

Theorem 4 ensures the algorithm runs for at most $\frac{\Delta_F L^{\frac{1}{2}} \rho^{\frac{1}{4}}}{\sqrt{2}\varepsilon^{\frac{7}{4}}} + \frac{1}{\sqrt{2}} \left(\frac{L^2}{\varepsilon\rho}\right)^{\frac{1}{4}}$ iterations. We plug (51) in this bound. We have

$$
\begin{aligned}
\frac{\Delta_F L^{\frac{1}{2}} \rho^{\frac{1}{4}}}{\sqrt{2}\varepsilon^{\frac{7}{4}}} &= \frac{\Delta_F L^{\frac{1}{2}} \rho^{\frac{1}{4}}}{\sqrt{2}\left(\frac{\Delta_F^{\frac{4}{7}}(2L)^{\frac{2}{7}}\rho^{\frac{1}{7}}}{n^{\frac{4}{7}}} + \frac{4L^2}{\rho n^4}\right)^{\frac{7}{4}}} \\
&\leq \frac{\Delta_F L^{\frac{1}{2}} \rho^{\frac{1}{4}}}{\sqrt{2}\left(\frac{\Delta_F^{\frac{4}{7}}(2L)^{\frac{2}{7}}\rho^{\frac{1}{7}}}{n^{\frac{4}{7}}}\right)^{\frac{7}{4}}} \\
&= \frac{n}{2},
\end{aligned}
\tag{52}
$$

and

$$
\begin{aligned}
\frac{1}{\sqrt{2}} \left(\frac{L^2}{\varepsilon\rho}\right)^{\frac{1}{4}} &= \frac{1}{\sqrt{2}} \left(\frac{L^2}{\left(\frac{\Delta_F^{\frac{4}{7}}(2L)^{\frac{2}{7}}\rho^{\frac{1}{7}}}{n^{\frac{4}{7}}} + \frac{4L^2}{\rho n^4}\right)\rho}\right)^{\frac{1}{4}} \\
&\leq \frac{1}{\sqrt{2}} \left(\frac{L^2}{\frac{4L^2}{\rho n^4}\rho}\right)^{\frac{1}{4}} \\
&= \frac{n}{2}.
\end{aligned}
\tag{53}
$$

Combining (52) and (53) ensures the algorithm end at most within the fixed budget of $n$ iterations. Moreover, we know from Theorem 4 that the outputs verifies $\|\nabla F(\hat{z})\| \leq 45\varepsilon$, or with our choice of $\varepsilon$, see (51), we obtain

$$
\|\nabla F(\hat{z})\| \leq 45 \cdot 2^{\frac{2}{7}} \frac{\Delta_F^{\frac{4}{7}} L^{\frac{2}{7}} \rho^{\frac{1}{7}}}{n^{\frac{4}{7}}} + 180\frac{L^2}{\rho n^4}.
$$

## F   CONVERGENCE ANALYSIS OF CONTINUOUS RISP

Let $F : \mathbb{R}^d \to \mathbb{R}$. The Nesterov accelerated gradient

$$
\begin{cases}
z^k = x^k + (1 - \theta_n)(x^k - x^{k-1}) \\
x^{k+1} = z^k - \eta\nabla F(z^k)
\end{cases}
\tag{NAG}
$$

can be seen as a discretization of the following continuous dynamical system,

$$
\ddot{x}_t + \alpha(t)\dot{x}_t + \nabla F(x_t) = 0,
\tag{DS}
$$

where $\dot{x}_t$ and $\ddot{x}_t$ are respectively the first and second order derivative with respect to $t \in \mathbb{R}_+$, see (Su et al., 2016) for a seminal work, or (Kim & Yang, 2023, Section B.2.2) for a more general result. Equation DS defines a damping system, with damping coefficient $\alpha(t)$. Intuitively, $(x_t)_{t\geq 0}$ is a rolling object on the surface defined by the image of $f$, subject to some friction. The higher $\alpha(t)$, the higher the friction. We will consider the constant friction version, called the heavy ball equation (Polyak, 1964)

$$
\ddot{x}_t + \alpha\dot{x}_t + \nabla F(x_t) = 0.
\tag{HB}
$$

We can think of $(x_t)_{t\geq 0}$ in HB as a continuous version of $\{x^k\}_{k\geq 0}$ defined in NAG, with constant sequence $\theta_n = \theta$ for all $n \in \mathbb{N}$. There are several reasons to study $(x_t)_{t\geq 0}$:

1. Dealing with continuous system offers tools that do not exists in the discrete settings, such as derivation, which may leads to smoother computations. If one's goal is to study $\{x^k\}_{k\geq 0}$, one can start to study $(x_t)_{t\geq 0}$ in order to gain intuition.

2. The dynamical systems can be thought as a generalization of the algorithms, as there exists several way to discretize a dynamical system. By instance, considering the gradient flow equation

$$\dot{\boldsymbol{x}} = -\nabla F(\boldsymbol{x}_t), \qquad \text{(GF)}$$

one can consider gradient descent $\boldsymbol{x}^{k+1} = \boldsymbol{x}^k - \eta \nabla F(\boldsymbol{x}^k)$ as an explicit discretization of GF, while the proximal algorithm $\boldsymbol{x}^{k+1} = \mathsf{prox}_{\eta f}(\boldsymbol{x}^k) = \boldsymbol{x}^k - \eta \nabla F(\boldsymbol{x}^{k+1})$ can be thought as an implicit discretization of GF.

## F.1 CONTINOUS HEAVY BALL

In this section, we show how the inertial mechanisms of RISP-Prox (Algorithm 2) can be thought as discretization of the HB equation. The same derivation for RISP-GM (Algorithm 2) is deduced in a similar way, and has already been done, see (Su et al., 2016) or (Kim & Yang, 2023, Section B.2.2).

**From RISP-Prox to Heavy Ball.** Under Assumption 1, there exists $g$ which verifies $\mathsf{S} = -\nabla g$, such that the inertial mechanism of RISP-Prox (Algorithm 2) writes

$$\boldsymbol{z}^k = \boldsymbol{x}^k + (1 - \theta(\eta))(\boldsymbol{x}^k - \boldsymbol{x}^{k-1})$$
$$\boldsymbol{x}^{k+1} = \mathsf{prox}_{\eta f}(\boldsymbol{z}^k - \eta \nabla g(\boldsymbol{z}^k)),$$

where we make explicit the dependancy of $\theta$ in the algorithm step-size $\eta$. Because of the characterization of the proximal operator, and the fact that $f$ is differentiable, we can rewrite it in the following way

$$\boldsymbol{z}^k = \boldsymbol{x}^k + (1 - \theta(\eta))(\boldsymbol{x}^k - \boldsymbol{x}^{k-1}) \qquad (54)$$
$$\boldsymbol{x}^{k+1} = \boldsymbol{z}^k - \eta \nabla g(\boldsymbol{z}^k) - \eta \nabla f(\boldsymbol{x}^{k+1}) \qquad (55)$$

Merging Equations 54 and 55, we divide by $\sqrt{\eta}$ and rearrange to obtain

$$\frac{\boldsymbol{x}^{k+1} - \boldsymbol{x}^k}{\sqrt{\eta}} = (1 - \theta(\eta))\frac{\boldsymbol{x}^k - \boldsymbol{x}^{k-1}}{\sqrt{\eta}} - \sqrt{\eta}\nabla g(\boldsymbol{z}^k) - \sqrt{\eta}\nabla f(\boldsymbol{x}^{k+1}) \qquad (56)$$

We want to identify $\{\boldsymbol{x}^k\}_{k \geq 0}$ with a continuous curve $(\boldsymbol{x}_t)_{t \geq 0}$ through the identification $\boldsymbol{x}_{t_k} = \boldsymbol{x}^k$, where $(t_k)_{k \in \mathbb{N}}$ is a positive and increasing sequence depending of the algorithm stepsize $\eta$. A good choice for $(t_k)_{k \in \mathbb{N}}$ is given by $t_k = \sqrt{\eta}k$, see (Su et al., 2016). Then, using Taylor expensions, we have

$$\frac{\boldsymbol{x}^{k+1} - \boldsymbol{x}^k}{\sqrt{\eta}} = \frac{\boldsymbol{x}_{t_{k+1}} - \boldsymbol{x}_{t_k}}{\sqrt{\eta}} = \dot{\boldsymbol{x}}_{t_k} + \frac{1}{2}\ddot{\boldsymbol{x}}_{t_k} + o(\sqrt{\eta}) \qquad (57)$$

$$\frac{\boldsymbol{x}^k - \boldsymbol{x}^{k-1}}{\sqrt{\eta}} = \frac{\boldsymbol{x}_{t_k} - \boldsymbol{x}_{t_{k-1}}}{\sqrt{\eta}} = \dot{\boldsymbol{x}}_{t_k} - \frac{1}{2}\ddot{\boldsymbol{x}}_{t_k} + o(\sqrt{\eta}). \qquad (58)$$

Also, rewriting (54) we get

$$\frac{\boldsymbol{z}^k - \boldsymbol{x}^k}{\sqrt{\eta}} = (1 - \theta(\eta))\frac{\boldsymbol{x}^k - \boldsymbol{x}^{k-1}}{\sqrt{\eta}} = (1 - \theta(\eta))\dot{\boldsymbol{x}}_{t_k} + o(\sqrt{\eta}).$$

Multiplying by $\sqrt{\eta}$ on each side, we obtain

$$\boldsymbol{z}^k - \boldsymbol{x}^k = (1 - \theta(\eta))\sqrt{\eta}\dot{\boldsymbol{x}}_{t_k} + o(\sqrt{\eta}).$$

This allows us to use the Lipschitz gradient property of $g$, to write

$$\nabla g(\boldsymbol{z}^k) = \nabla g(\boldsymbol{x}^k) + O(\sqrt{\eta}) = \nabla g(\boldsymbol{x}_{t_k}) + O(\sqrt{\eta}). \qquad (59)$$

By (57), we also have

$$\nabla f(\boldsymbol{x}^{k+1}) = \nabla f(\boldsymbol{x}^k) + O(\sqrt{\eta}) = \nabla f(\boldsymbol{x}_{t_k}) + O(\sqrt{\eta}). \qquad (60)$$

Note that precisely in (60) appears that the proximal operator acts as an implicit discretization. In the case of RISP instead of RISP-Prox, we can replace the computations of (60) by those in (59). Injecting (57)-(60) in (56), we obtain

$$\dot{\boldsymbol{x}}_{t_k} + \frac{1}{2}\sqrt{\eta}\ddot{\boldsymbol{x}}_{t_k} + o(\sqrt{\eta}) = (1 - \theta(\eta))\left(\dot{\boldsymbol{x}}_{t_k} - \frac{1}{2}\sqrt{\eta}\ddot{\boldsymbol{x}}_{t_k}\right) - \sqrt{\eta}\nabla g(\boldsymbol{x}_{t_k}) - \sqrt{\eta}\nabla f(\boldsymbol{x}_{t_k}) + O(\sqrt{\eta}).$$

Rearranging, dividing by $\sqrt{\eta}$, and using $F = f + g$, we deduce

$$\ddot{\boldsymbol{x}}_{t_k} + \frac{\theta(\eta)}{\sqrt{\eta}}\dot{\boldsymbol{x}}_{t_k} + \nabla F(\boldsymbol{x}_{t_k}) = O(\sqrt{\eta}).$$

We can conclude by taking $\eta \to 0$, assuming $\lim_{\eta \to 0} \frac{\theta(\eta)}{\sqrt{\eta}} := \alpha$ exists.

**Remark 4.** *In the case of $\mu$-strongly convex functions with $L$-Lipschitz gradient, a classical choice of $\theta(\eta)$ is $\frac{2\sqrt{\mu\eta}}{1+\sqrt{\mu\eta}}$. Then, $\lim_{\eta \to 0} \frac{\theta(\eta)}{\sqrt{\eta}} = 2\sqrt{\mu}$, and we recover the classical choice of friction $\alpha(\cdot)$ of this setting, see (Siegel, 2019).*

### F.2 RESTARTED HEAVY BALL: CONTINOUS RISP

Inspired by Algorithm 1 in (Li & Lin, 2023), we consider the following continuous restart procedure. Let $\boldsymbol{x}^0 \in \mathbb{R}^d$. We define $\{\boldsymbol{x}_{t,1}\}_{t \in \mathbb{R}_+}$ such that it verifies (6)

$$\ddot{\boldsymbol{x}}_{t,1} + \alpha\dot{\boldsymbol{x}}_{t,1} + \nabla F(\boldsymbol{x}_{t,1}) = 0, \tag{HB}$$

with initial conditions $(\boldsymbol{x}_{0,1}, \dot{\boldsymbol{x}}_{0,1} = \boldsymbol{x}^0, 0)$. We then define the restart time

$$T_1 = \inf_t \{t \int_0^t \|\dot{\boldsymbol{x}}_{t,1}\|^2 \, ds = B^2\}.$$

We define recursively $\{\boldsymbol{x}_{t,k}\}_{t \in \mathbb{R}^+}$ as verifying (6) with initial conditions $(\boldsymbol{x}_{(0,k)}, \dot{\boldsymbol{x}}_{(0,k)}) = (\boldsymbol{x}_{(T_{k-1}, k-1)}, 0)$, and

$$T_k = \inf_t \{t \int_0^t \|\dot{\boldsymbol{x}}_{t,k}\|^2 \, ds = B^2\}.$$

Finally, we define $(\boldsymbol{x}_t^c)_{t \geq 0}$ the concatenation of all the trajectories $\{\boldsymbol{x}_{t,k}\}_{t \in [0, T_k]}$, such that one has for $0 \leq t \leq T_k$

$$\boldsymbol{x}_{t + \sum_{i=1}^{k-1} T_i}^c = \boldsymbol{x}_{t,k}.$$

For some $T_{\max} > 0$, we denote $K := \arg\min_k \{T_{\max} \int_0^{T_{\max}} \|\dot{\boldsymbol{x}}_{t,k}\|^2 \, ds > B^2\} - 1$ the number of restart. Below we repeat Theorem 3 for convenience.

**Theorem.** *Let Assumptions 1 and 3 hold. Consider the $(\boldsymbol{x}_t^c)_{t \geq 0}$ running for a total execution time $T \in \mathbb{R}_+$. Define the output $\hat{\boldsymbol{x}}$ as the average $\hat{\boldsymbol{x}} := \frac{1}{K_0} \int_0^{K_0} \boldsymbol{x}_{t + \sum_{i=1}^K T_i}^c \, dt$, where $K_0 = \arg\min_{t \in \left[\frac{T_{\max}}{2}, T_{\max}\right]} \left\| \dot{\boldsymbol{x}}_{t + \sum_{i=1}^K T_i}^c \right\|$. Set the parameters as $\alpha = (\varepsilon\rho)^{1/4}$, $T_{\max} = (\varepsilon\rho)^{-1/4}$, and $B = \sqrt{\varepsilon/\rho}$. Then, under these conditions, the gradient norm satisfies*

$$\|\nabla F(\hat{\boldsymbol{x}})\| \leq 5\varepsilon = \mathcal{O}(T^{-4/7}),$$

*with $\varepsilon = 2^{4/7}\rho^{1/7}\Delta_F^{4/7}T^{-4/7} + 2^4\rho^{-1}T^{-4}$.*

The theorem is proven in the next section. Note that is does not require the Lipschitz gradient Assumption in its proof. However, a local Lipschitz gradient property of $F$, that is the Lipschitz Hessian property, is needed to ensure that each trajectory $\{\boldsymbol{x}_t t, k\}_{t \in \mathbb{R}_+}$ can be well defined.

**Remark 5.** *A similar convergence rate for the vanilla heavy ball (6) without restart is provided in (Okamura et al., 2024, Theorem 1). As these authors do not achieve a similar result in the discrete setting, Theorem 3 highlights the crucial role of the restart mechanism for (6).*

### F.3 PROOF OF THEOREM 3

In order to prove Theorem 3, we first prove the following intermediate result, which bounds the number of epoch such that the restart criterion is met.

**Theorem.** *Consider $(\boldsymbol{x}_t^c)_{t \geq 0}$ with $\alpha = (\varepsilon\rho)^{1/4}$, $T_{\max} = (\varepsilon\rho)^{-1/4}$ and $B = \sqrt{\varepsilon/\rho}$ for some $\varepsilon > 0$. Denote $K$ the total number of restarts. Then, $K$ is finite, and defining $\hat{\boldsymbol{x}} = \frac{1}{K_0} \int_0^{K_0} \boldsymbol{x}_{t + \sum_{i=1}^K T_i}^c \, dt$ with $K_0 = \arg\min_{t \in \left[\frac{T_{\max}}{2}, T_{\max}\right]} \left\| \dot{x}_{t + \sum_{i=1}^K T_i}^c \right\|$, we have*

1. $\sum_{i=1}^{K} T_i + T_{\max} \leq (F(\boldsymbol{x}_{init}) - \min F)\varepsilon^{-7/4}\rho^{-1/4} + (\varepsilon\rho)^{-1/4}$;

2. $\|\nabla F(\hat{\boldsymbol{x}})\| \leq 5\varepsilon$.

From a high perspective, the proof follows similar steps as in the discrete case. We start to show that the total number of restart is of the order $\mathcal{O}(\varepsilon^{-3/2})$. As one epoch ends at $t \leq T_{\max} = \mathcal{O}(\varepsilon^{-1/4})$, the total amount of running time of the process is $\mathcal{O}(\varepsilon^{-7/4})$. Interestingly, the latter can be shown without assuming smoothness properties. Finally, it remains to show that the process allows to find a point with small gradient norm. This last part requires the Lipschitz property of the Hessian matrix.

**Sufficient decrease in each epoch.** Similarly to the discrete case, we show that each time the restart criterion is triggered, then we have a sufficient decrease of the function value over the epoch.

**Lemma 8.** *Let* $(\boldsymbol{x}_t)_{t \geq 0}$ *verifies HB with initial condition* $(\boldsymbol{x}_0, 0)$. *Assume* $T \int_0^T \|\dot{\boldsymbol{x}}_t\|^2 \, dt = B^2$, *for* $T \leq T_{\max}$. *We then have*

$$F(\boldsymbol{x}_T) - F(\boldsymbol{x}_0) \leq -\alpha \frac{B^2}{T_{\max}}.$$

*Proof.* Set the following Lyapunov function

$$E_t = F(\boldsymbol{x}_t) + \frac{1}{2}\|\dot{\boldsymbol{x}}_t\|^2.$$

We differentiate $(E_t)_{t \geq 0}$

$$\dot{E}_t = \langle \nabla F(\boldsymbol{x}_t), \dot{\boldsymbol{x}}_t \rangle + \langle \dot{\boldsymbol{x}}_t, \ddot{\boldsymbol{x}}_t \rangle$$
$$\overset{(HB)}{=} \langle \nabla F(\boldsymbol{x}_t), \dot{\boldsymbol{x}}_t \rangle - \langle \dot{\boldsymbol{x}}_t, \alpha\dot{\boldsymbol{x}}_t + \nabla F(\boldsymbol{x}_t) \rangle = -\alpha\|\dot{\boldsymbol{x}}_t\|^2.$$

Integrating for $t \in [0, T]$, we get

$$E_T - E_0 = -\alpha \int_0^T \|\dot{\boldsymbol{x_t}}\|^2 \, dt.$$

Because $T \int_0^T \|\dot{\boldsymbol{x}}_t\|^2 \, dt = B^2$ with $T \leq T_{\max}$, we have

$$-\alpha \int_0^T \|\dot{\boldsymbol{x_t}}\|^2 \, dt = -\alpha\frac{B^2}{T} \leq -\alpha\frac{B^2}{T_{\max}}.$$

Finally, noting that assuming $\dot{\boldsymbol{x}}_0 = 0$ implies $E_T - E_0 \geq F(\boldsymbol{x}_T) - F(\boldsymbol{x}_0)$, we have

$$F(\boldsymbol{x}_T) - F(\boldsymbol{x}_0) \leq -\alpha\frac{B^2}{T_{\max}}.$$

□

Lemma 8 will allow to bounds the total number of epochs. Note that in this continuous setting, compared to the discrete setting many things simplify. By instance, there is no need to use quadratic approximation and to split the space according to the eigenvalues of the Hessian matrix of $f$, see for comparison Section E.4.

**Small gradient in last epoch.** In this section, we show that in the last epoch, *i.e.* if the restart criterion is not triggered for all $t \leq T_{\max}$, we can find a point with small gradient norm. This part is a bit trickier than the previous one, as we need to use a different argument from the discrete case. In the discrete case, the linearity of the quadratic approximation was crucial to get the desired result, see Section E.5. Since no such linearity property holds for the gradient in our setting, we proceed differently. We begin with Lemma 9, which provides a suitable bound on the average of the gradients.

**Lemma 9.** *Let* $(\boldsymbol{x}_t)_{t \geq 0}$ *verifies HB with initial condition* $\dot{\boldsymbol{x}}_0 = 0$. *Assume* $\forall T \in [0, T_{\max}]$, *one has* $T \int_0^T \|\dot{\boldsymbol{x}}_t\|^2 \, dt < B^2$. *Then, defining* $K_0 = \arg\min_{t \in \left[\frac{T_{\max}}{2}, T_{\max}\right]} \|\dot{\boldsymbol{x}}_t\|$, *we have*

$$\left\|\frac{1}{K_0}\int_0^{K_0} \nabla F(\boldsymbol{x}_s)ds\right\| \leq \frac{2\sqrt{2}B}{T_{\max}^2} + \frac{2\alpha B}{T_{\max}}.$$

*Proof.*

$$\left\| \frac{1}{K_0} \int_0^{K_0} \nabla F(\boldsymbol{x}_s) ds \right\| = \frac{1}{K_0} \left\| \int_0^{K_0} (\ddot{\boldsymbol{x}}_s + \alpha \dot{\boldsymbol{x}}_s) ds \right\|$$

$$= \frac{1}{K_0} \left\| \dot{\boldsymbol{x}}_{K_0} - \dot{\boldsymbol{x}}_0 + \alpha(\boldsymbol{x}_{K_0} - \boldsymbol{x}_0) \right\|$$

$$\leq \frac{2}{T_{\mathsf{max}}} \left\| \dot{\boldsymbol{x}}_{K_0} \right\| + \frac{2\alpha}{T_{\mathsf{max}}} \left\| \boldsymbol{x}_{K_0} - \boldsymbol{x}_0 \right\|, \tag{61}$$

where the first equality is by definition of (HB), and the last inequality is by triangular inequality, using $\dot{\boldsymbol{x}}_0 = 0$ and because $K_0 \in [\frac{T_{\mathsf{max}}}{2}, T_{\mathsf{max}}]$. Then,

$$\left\| \boldsymbol{x}_{K_0} - \boldsymbol{x}_0 \right\|^2 = \left\| \int_0^{K_0} \dot{\boldsymbol{x}}_s ds \right\|^2 \leq K_0 \int_0^{K_0} \left\| \dot{\boldsymbol{x}}_s \right\|^2 ds \leq B^2, \tag{62}$$

where the last inequality is because we assumed $\forall T \in [0, T_{\mathsf{max}}]$, $T \int_0^T \left\| \dot{\boldsymbol{x}}_t \right\|^2 dt < B^2$. Also, as $K_0 = \arg\min_{t \in \left[\frac{T_{\mathsf{max}}}{2}, T_{\mathsf{max}}\right]} \left\| \dot{\boldsymbol{x}}_t \right\|$, one has

$$\left\| \dot{\boldsymbol{x}}_{K_0} \right\|^2 \leq \frac{1}{T_{\mathsf{max}} - \frac{T_{\mathsf{max}}}{2}} \int_{\frac{T_{\mathsf{max}}}{2}}^{T_{\mathsf{max}}} \left\| \dot{\boldsymbol{x}}_t \right\|^2 dt$$

$$\leq \frac{1}{T_{\mathsf{max}} - \frac{T_{\mathsf{max}}}{2}} \int_0^{T_{\mathsf{max}}} \left\| \dot{\boldsymbol{x}}_t \right\|^2 dt$$

$$\leq \frac{1}{T_{\mathsf{max}} - \frac{T_{\mathsf{max}}}{2}} \frac{B^2}{T_{\mathsf{max}}} \leq \frac{2B^2}{T_{\mathsf{max}}^2}, \tag{63}$$

where used $\int_0^{T_{\mathsf{max}}} \left\| \dot{\boldsymbol{x}}_t \right\|^2 dt \leq \frac{B^2}{T_{\mathsf{max}}}$. Injecting (62) and (63) in (61), we obtain

$$\left\| \frac{1}{K_0} \int_0^{K_0} \nabla F(\boldsymbol{x}_s) ds \right\| \leq \frac{2\sqrt{2}B}{T_{\mathsf{max}}^2} + \frac{2\alpha B}{T_{\mathsf{max}}}.$$

$\square$

To conclude we need to bound the gradient of an average of the trajectory. But the gradient is not linear so $\nabla F(\frac{1}{K_0} \int_0^{K_0} \boldsymbol{x}_s ds) \neq \frac{1}{K_0} \int_0^{K_0} \nabla F(\boldsymbol{x}_s) ds$. This is where the Hessian Lipschitz property steps in, as it allows us to bound the gap between these two quantities. We use the following result.

**Lemma 10.** *(Okamura et al., 2024, Lemma 1) For $t > 0$, let $\boldsymbol{z} : [0,t] \to \mathbb{R}^d$, $f$ a $\rho$-Lipschitz Hessian function, $w : [0,t] \to [0,\infty)$ a measurable function satisfies $\int_0^t w(s) ds = 1$, and $\bar{\boldsymbol{z}} := \int_0^t w(s) \boldsymbol{z}(s) ds$. Then*

$$\left\| \nabla F(\bar{\boldsymbol{z}}) - \int_0^t w(s) \nabla F(\boldsymbol{z}(s)) ds \right\| \leq \frac{\rho}{2} \int_0^t \left\| \dot{\boldsymbol{z}}(s) \right\|^2 \left( \int_0^s \int_s^t w(\sigma) w(\tau)(\tau - \sigma) d\sigma d\tau \right) ds.$$

As in (Okamura et al., 2024) we will apply this result with $(\boldsymbol{z}_t)_{t \in \mathbb{R}_+} = (\boldsymbol{x}_t)_{t \in \mathbb{R}_+}$ defined by (HB), but with a different choice of weight function $w$.

**Lemma 11.** *Let $(\boldsymbol{x}_t)_{t \geq 0}$ verifies HB with initial condition $(\boldsymbol{x}_0, 0)$. Assume $\forall T \in [0, T_{\mathsf{max}}]$, one has $T \int_0^T \left\| \dot{\boldsymbol{x}}_t \right\|^2 dt < B^2$. Then, defining $\hat{\boldsymbol{x}} = \frac{1}{K_0} \int_0^{K_0} \boldsymbol{x}_t dt$ with $K_0 = \arg\min_{t \in \left[\frac{T_{\mathsf{max}}}{2}, T_{\mathsf{max}}\right]} \left\| \dot{\boldsymbol{x}}_t \right\|$, one has*

$$\left\| \nabla F(\hat{\boldsymbol{x}}) - \frac{1}{K_0} \int_0^{K_0} \nabla F(\boldsymbol{x}_t) dt \right\| \leq \frac{\rho B^2}{16}.$$

*Proof.* Applying Lemma 10 to $(\boldsymbol{x}_t)_{t\in[0,K_0]}$ and $w(\cdot) = \frac{1}{K_0}$, we get

$$\left\| \nabla F(\hat{\boldsymbol{x}}) - \frac{1}{K_0} \int_0^{K_0} \nabla F(\boldsymbol{x}_s) ds \right\| \leq \frac{\rho}{2} \int_0^{K_0} \|\dot{\boldsymbol{x}}_s\|^2 \left( \int_0^s \int_s^{K_0} w(\sigma)w(\tau)(\tau - \sigma)d\sigma d\tau \right) ds$$

$$\leq \frac{\rho}{2K_0^2} \int_0^{K_0} \|\dot{\boldsymbol{x}}_s\|^2 \left( \int_0^s \int_s^{K_0} (\tau - \sigma)d\sigma d\tau \right) ds. \quad (64)$$

Then, we have

$$\int_0^s \int_s^{K_0} (\tau - \sigma)d\sigma d\tau = \int_s^{K_0} \left( \tau s - \frac{s^2}{2} \right) d\tau$$

$$= \int_s^{K_0} \tau s \, d\tau - \int_s^{K_0} \frac{s^2}{2} d\tau$$

$$= \frac{s}{2} \left( K_0^2 - s^2 \right) - \frac{s^2}{2}(K_0 - s)$$

$$= \frac{sK_0(K_0 - s)}{2},$$

where in the last equality we use the elementary fact that $a^2 - b^2 = (a+b)(a-b)$ for any $a, b \in \mathbb{R}$. The above quantity is maximized for $s = \frac{K_0}{2}$, such that

$$\int_0^s \int_s^{K_0} (\tau - \sigma)d\sigma d\tau \leq \frac{K_0^3}{8}. \quad (65)$$

By injecting (65) into (64), we obtain

$$\left\| \nabla F(\hat{\boldsymbol{x}}) - \frac{1}{K_0} \int_0^{K_0} \nabla F(\boldsymbol{x}_s) ds \right\| \leq \frac{\rho}{16} K_0 \int_0^{K_0} \|\dot{\boldsymbol{x}}_s\|^2 ds.$$

Now, because $K_0 \int_0^{K_0} \|\dot{\boldsymbol{x}}_t\|^2 dt < B^2$, we conclude that

$$\left\| \nabla F(\hat{\boldsymbol{x}}) - \frac{1}{K_0} \int_0^{K_0} \nabla F(\boldsymbol{x}_s) ds \right\| \leq \frac{\rho B^2}{16}.$$

$\square$

Now, we can prove Theorem 3

*Proof of Theorem 3.* If the $k$-th epoch is such that $T_k :=$ $\arg\inf_{t\in[0,T_{\max}]} \left\{ t \int_0^t \left\| \dot{\boldsymbol{x}}^c_{s+\sum_{i=1}^{k-1} T_i} \right\|^2 ds = B^2 \right\}$ exists, then Lemma 8 applied to $\left( \boldsymbol{x}^c_{t+\sum_{i=1}^{k-1} T_i} \right)_{t\geq 0}$ ensures that the following decrease holds

$$f\left( \boldsymbol{x}^c_{\sum_{i=1}^k T_i} \right) - f\left( \boldsymbol{x}^c_{\sum_{i=1}^{k-1} T_i} \right) \leq -\alpha \frac{B^2}{T_{\max}} = -\varepsilon^{\frac{3}{2}} \rho^{\frac{1}{2}},$$

where we used our choice of parameters. Summing over $1 \leq k \leq K$, we get

$$\min F - F(\boldsymbol{x}^0) \leq f\left( \boldsymbol{x}^c_{\sum_{i=1}^K T_i} \right) - F(\boldsymbol{x}^0) \leq -\varepsilon^{\frac{3}{2}} \rho^{\frac{1}{2}} K.$$

This concludes that $K \leq (F(\boldsymbol{x}^0) - \min F)\varepsilon^{-3/2}\rho^{1/2}$. In other words, the restart number is upper bounded by $(F(\boldsymbol{x}^0) - \min F)\varepsilon^{-3/2}\rho^{1/2}$, such that the epoch number is upper bounded by $(F(\boldsymbol{x}^0) - \min F)\varepsilon^{-3/2}\rho^{1/2} + 1$. In particular, it ensures that $K < +\infty$. Since an epoch duration is at most $T_{\max} = (\varepsilon\rho)^{-1/4}$, this means that the process $(\boldsymbol{x}^c_t)_{t\geq 0}$ stops at a total amount of time upper bounded by $(F(\boldsymbol{x}^0) - \min F)\varepsilon^{-7/4}\rho^{1/4} + (\varepsilon\rho)^{-1/4}$.

**Small gradient norm of the outputs.** Now, applying Lemma 9 and Lemma 11 to $\left(\boldsymbol{x}_{t+\sum_{i=1}^{K} T_i}^{c}\right)_{t\geq 0}$, we deduce that in the last epoch such that the restart criterion is not triggered. This is because that for $\hat{\boldsymbol{x}} = \frac{1}{K_0} \int_0^{K_0} \boldsymbol{x}_{t+\sum_{i=1}^{K} T_i}^{c} dt$ with $K_0 = \arg\min_{t\in\left[\frac{T_{\max}}{2}, T_{\max}\right]} \left\|\dot{\boldsymbol{x}}_{t+\sum_{i=1}^{K} T_i}^{c}\right\|$, we have

$$\|\nabla F(\hat{\boldsymbol{x}})\| \leq \left\|\nabla F(\hat{\boldsymbol{x}}) - \frac{1}{K_0} \int_0^{K_0} \nabla F(\boldsymbol{x}_{t+\sum_{i=1}^{K} T_i}^{c}) dt\right\| + \left\|\frac{1}{K_0} \int_0^{K_0} \nabla F(\boldsymbol{x}_{t+\sum_{i=1}^{K} T_i}^{c}) dt\right\|$$

$$\leq \frac{\rho B^2}{16} + \frac{2\sqrt{2}B}{T_{\max}^2} + \frac{2\alpha B}{T_{\max}}.$$

According to our choice of parameters, we have

$$\frac{\rho B^2}{16} = \frac{1}{16}\varepsilon, \quad \frac{2\sqrt{2}B}{T_{\max}^2} = 2\sqrt{2}\varepsilon, \quad \frac{2\alpha B}{T_{\max}} = 2\varepsilon,$$

such that

$$\|\nabla F(\hat{\boldsymbol{x}})\| \leq 5\varepsilon.$$

We are ready to conclude by expressing the result as a decrease in term of number of iterations.

**Result for a fixed budget $T > 0$.** We fix a budget of $T > 0$ computational time. Let

$$\varepsilon := 2^{\frac{4}{7}} \rho^{\frac{1}{7}} (F(\boldsymbol{x}^0) - \min F)^{\frac{4}{7}} T^{-\frac{4}{7}} + 2^4 \rho^{-1} T^{-4}. \tag{66}$$

We showed earlier that the total computational running time of the process is at most $(F(\boldsymbol{x}^0) - \min F)\varepsilon^{-7/4}\rho^{1/4} + (\varepsilon\rho)^{-1/4}$. Plugging our choice of $\varepsilon$ of (66), we have

$$(F(\boldsymbol{x}^0) - \min F)\varepsilon^{-\frac{7}{4}}\rho^{\frac{1}{4}} = (F(\boldsymbol{x}^0) - \min F)\left(2^{\frac{4}{7}} \rho^{\frac{1}{7}} (F(\boldsymbol{x}^0) - \min F)^{\frac{4}{7}} T^{-\frac{4}{7}} + 2^4 \rho^{-1} T^{-4}\right)^{-\frac{7}{4}} \rho^{\frac{1}{4}}$$

$$\leq (F(\boldsymbol{x}^0) - \min F)\left(2^{\frac{4}{7}} \rho^{\frac{1}{7}} (F(\boldsymbol{x}^0) - \min F)^{\frac{4}{7}} T^{-\frac{4}{7}}\right)^{-\frac{7}{4}} \rho^{\frac{1}{4}} \tag{67}$$

$$= \frac{T}{2},$$

and

$$(\varepsilon\rho)^{-\frac{1}{4}} = \left(2^{\frac{4}{7}} \rho^{\frac{1}{7}} (F(\boldsymbol{x}^0) - \min F)^{\frac{4}{7}} T^{-\frac{4}{7}} + 2^4 \rho^{-1} T^{-4}\right)^{-\frac{1}{4}} \rho^{-\frac{1}{4}}$$

$$\leq \left(2^4 \rho^{-1} T^{-4}\right)^{-\frac{1}{4}} \rho^{-\frac{1}{4}} \tag{68}$$

$$= \frac{T}{2}.$$

Combining (67) and (68) ensures the process ends in at most $T$ computational time. As the output $\hat{\boldsymbol{x}}$ verifies $\|\nabla F(\hat{\boldsymbol{x}})\| \leq 5\varepsilon$, then (66) ensures that

$$\|\nabla F(\hat{\boldsymbol{x}})\| \leq 5 \cdot 2^{\frac{4}{7}} \rho^{\frac{1}{7}} (F(\boldsymbol{x}^0) - \min F)^{\frac{4}{7}} T^{-\frac{4}{7}} + 5 \cdot 2^4 \rho^{-1} T^{-4}.$$

$\square$

### F.4 CONVERGENCE ANALYSIS OF CONTINUOUS RED

As mentioned in the introduction of Appendix F, the gradient descent algorithm

$$\boldsymbol{x}^{k+1} = \boldsymbol{x}^k - \eta \nabla F(\boldsymbol{x}^k) \tag{69}$$

can be seen as a discretization of a continuous dynamical system, namely the gradient flow equation

$$\dot{\boldsymbol{x}} = -\nabla F(\boldsymbol{x}_t). \tag{GF}$$

Under Assumption 1, the gradient flow equation can be used to formulate a continuous version of (2)

$$\dot{\boldsymbol{x}} = -\nabla f(\boldsymbol{x}_t) + S(\boldsymbol{x}_t), \tag{Cont-RED}$$

where (Cont-RED) is a gradient flow for $\nabla F = \nabla f + \nabla g$. This enables us to formulate a convergence rate for this continuous version of RED, in order to confirm that the benefit of the inertial mechanism also holds in the continuous setting.

| **Algorithm 4** RED-GM | **Algorithm 5** RED-Prox |
|---|---|
| **Require:** $\boldsymbol{x}^0 \in \mathbb{R}^d, n > 0, \eta > 0$, and $\tau > 0$ | **Require:** $\boldsymbol{x}^0 \in \mathbb{R}^d, n > 0, \eta > 0$, and $\tau > 0$ |
| 1: $k = 0$ | 1: $k = 0$ |
| 2: **while** $k < n$ **do** | 2: **while** $k < n$ **do** |
| 3: $\quad \boldsymbol{x}^{k+1} = \boldsymbol{x}^k - \eta\big(\nabla f(\boldsymbol{x}^k) + \tau(\boldsymbol{x} - \mathsf{D}_\sigma(\boldsymbol{x}^k))\big)$ | 3: $\quad \boldsymbol{x}^{k+1} = \mathsf{prox}_{\eta f}\big(\boldsymbol{x}^k - \tau \cdot \eta(\boldsymbol{x} - \mathsf{D}_\sigma(\boldsymbol{x}^k))\big)$ |
| 4: $\quad k = k + 1$ | 4: $\quad k = k + 1$ |
| 5: **end while** | 5: **end while** |
| 6: $K_0 = \arg\min_{0 \le k < n} \|\nabla F(\boldsymbol{x}^k)\|$ | 6: $K_0 = \arg\min_{0 \le k < n} \|\nabla F(\boldsymbol{x}^k)\|$ |
| 7: **return** $\boldsymbol{x}^{K_0}$ | 7: **return** $\boldsymbol{x}^{K_0}$ |

**Theorem 5.** *Let $(\boldsymbol{x}_t)_{t \ge 0}$ be a gradient flow (GF) for $\nabla F = \nabla f + \nabla g$. In at most $T$ execution time, the process achieves a point $\tilde{x}$ such that $\|\nabla F(\tilde{x})\| \le \sqrt{\frac{F(\boldsymbol{x}_0) - \min F}{T}} = \mathcal{O}(T^{-1/2})$.*

*Proof.* Let
$$E_t = F(\boldsymbol{x}_t) - \min F.$$
We derivate $E_t$
$$\dot{E}_t = \langle \nabla F(\boldsymbol{x}_t), \dot{\boldsymbol{x}} \rangle = -\|\nabla F(\boldsymbol{x}_t)\|^2.$$
We integrate between $0$ and $T$
$$\int_0^T \dot{E}_t ds = -\int_0^T \|\nabla F(\boldsymbol{x}_t)\|^2 dt = E_T - E_0 \ge \min F - F(\boldsymbol{x}_0).$$
Rearranging, and taking the min, we get
$$T \min_{t \in [0,T]} \|\nabla F(\boldsymbol{x}_t)\|^2 \le \int_0^T \|\nabla F(\boldsymbol{x}_t)\|^2 dt \le F(\boldsymbol{x}_0) - \min F.$$
Dividing by $T$, and considering $\tilde{T} := \arg\min_{t \in [0,T]} \|\nabla F(\boldsymbol{x}_t)\|^2$, we obtain
$$\|\nabla f(\boldsymbol{x}_{\tilde{T}})\| \le \sqrt{\frac{F(\boldsymbol{x}_0) - \min F}{T}}.$$
$\square$

The result of Theorem 5 highlights that without the inertial mechanism, we lose the $\mathcal{O}(T^{-4/7})$ convergence rate (Theorem 3) for a $\mathcal{O}(T^{-1/2})$ convergence rate. Note also that Theorem 5 is consistent with Theorem 1, with respectively a convergence rate of $\mathcal{O}(T^{-1/2})$ for the continuous time, and $\mathcal{O}(n^{-1/2})$ for the discrete time.

**Remark 6.** *As it was the case when considering the restarted heavy ball equation, there is no need to assume the gradient Lipschitz. In fact, this gradient Lipschitzness is commonly useful to ensure that the discretization preserves the continuous behavior.*

## G  ADDITIONAL EXPERIMENTS

First, we write explicitly below the algorithms RED-GM (Algorithm 4) and RED-Prox (Algorithm 5) for more clarity.

### G.1  ON THE DENOISER

In this part, we discuss in detail how the assumptions on the regularization can be verified in practice and which denoiser weights and architecture are used in our experiments.

**On the denoiser assumptions.**  In our experiments (except for linear inverse scattering), we use regularizer $g_\sigma$ proposed by (Hurault et al., 2022a)
$$g_\sigma(\boldsymbol{x}) = \frac{1}{2\sigma^2}\|\boldsymbol{x} - \mathsf{N}_\sigma(\boldsymbol{x})\|^2,$$
where $\boldsymbol{x} \in \mathbb{R}^d$ and $\mathsf{N}_\sigma$ is a neural network with DRUNet architecture (Zhang et al., 2021).

**Proposition 2.** *The regularization $g_\sigma(\boldsymbol{x}) = \frac{1}{2\sigma^2}\|\boldsymbol{x} - \mathsf{N}_\sigma(\boldsymbol{x})\|^2$ induces by gradient-step denoiser, with SoftPlus activations, verifies Assumption 2-3.*

Proposition 2 shows that the gradient step denoiser with SoftPlus activations verifies the assumptions that are necessary to obtain convergence guarantees. However, if the activation function is different, for instance `ReLU` of `eLU`, then regularization does not necessarily verify the assumption 2-3. Therefore, the theoretical assumptions are ensured in experiments on deblurring, inpainting, super-resolution and Rician noise removal but not in MRI and ODT.

*Proof.* We assume that $\mathsf{N}_\sigma$ has `SoftPlus` activation functions. The SoftPlus function is defined for $x \in \mathbb{R}$, by

$$\mathtt{SoftPlus}(x) = \log\left(1 + e^x\right).$$

We have the following derivatives for the Softplus activation function, for $x \in \mathbb{R}$,

$$|\mathtt{SoftPlus}^{(1)}(x)| = |\frac{1}{1 + e^{-x}}| \le 1$$

$$|\mathtt{SoftPlus}^{(2)}(x)| = |\frac{1}{4}\cosh^{-2}(x)| \le \frac{1}{4}$$

$$|\mathtt{SoftPlus}^{(3)}(x)| = |\frac{1}{2}\tanh(x)\cosh^{-2}(x)| \le \frac{1}{2}.$$

Because `SoftPlus`$^{(2)}$ and `SoftPlus`$^{(3)}$ are bounded, we deduce that `SoftPlus`$^{(1)}$ and `SoftPlus`$^{(2)}$ are Lipschitz and bounded. Therefore, by composition and sum, $J_{\mathsf{N}_\sigma}$, the Jacobian of $\mathsf{N}_\sigma$, and $d_{J_{\mathsf{N}_\sigma}^T}$, the differential of $J_{\mathsf{N}_\sigma}^T$, are Lipschitz and bounded.

Then the gradient of the regularization $g_\sigma$ is computed by

$$\sigma^2 \nabla g_\sigma(\boldsymbol{x}) = \boldsymbol{x} - \mathsf{N}_\sigma(\boldsymbol{x}) - [J_{\mathsf{N}_\sigma}(\boldsymbol{x})]^T (\boldsymbol{x} - \mathsf{N}_\sigma(\boldsymbol{x})),$$

with $J_{N_\sigma}(\boldsymbol{x})$ the jacobian matrix of $\mathsf{N}_\sigma$ at the point $\boldsymbol{x}$ and the Hessian of the regularization is

$$\sigma^2 \nabla^2 g_\sigma(\boldsymbol{x}) = \mathsf{I} - J_{\mathsf{N}_\sigma}(\boldsymbol{x}) - d_{J_{\mathsf{N}_\sigma}^T}[\boldsymbol{x}](\boldsymbol{x} - \mathsf{N}_\sigma(\boldsymbol{x})) - [J_{\mathsf{N}_\sigma}(\boldsymbol{x})]^T (\mathsf{I} - J_{\mathsf{N}_\sigma}(\boldsymbol{x})),$$

where $d_{J_{\mathsf{N}_\sigma}^T}[\boldsymbol{x}] : \mathbb{R}^d \to \mathbb{R}^{d \times d}$ is the differential of $J_{\mathsf{N}_\sigma}$ at point $\boldsymbol{x}$, thus $d_{J_{\mathsf{N}_\sigma}^T}[\boldsymbol{x}](\boldsymbol{x} - \mathsf{N}_\sigma(\boldsymbol{x})) \in \mathbb{R}^{d \times d}$.

Finally, Assumptions 2-3 on the regularization are verified. $\square$

**On the pre-trained weights.** For image deblurring, inpainting, super-resolution and Rician noise removal, we use the pre-trained weights proposed in (Hurault et al., 2022b) for GS-DRUNet with `SoftPlus` activation. For MRI reconstruction, we use the pre-trained weights provided in (Hurault et al., 2022a) trained on gray-scale natural images with `eLU` activations functions. We do not find pre-trained weights with `SoftPlus` activations and MRI images, therefore we use these ones. All links to download the weights are provided in the README file of the code.

### G.2 LINEAR INVERSE PROBLEM

**Experimental setup.** For RISP methods, we set the inertia parameter to $\theta = 0.2$. We note that the algorithm's performance is robust to this choice, as the restart mechanism enhances stability and reduces the need for extensive parameter tuning; see Figure 16 in Appendix G.2.1. We note that due to the non-convex nature of the score-based prior, different methods may converge to distinct local solutions, which can lead to different final PSNR values. Details on the calculation of $\nabla f$ and $\mathrm{prox}_{\eta f}$, and more results for each problem are given in Appendices G.2.1-G.2.3. The specific configurations for each linear inverse problem are outlined below.

- *Deblurring.* Following (Zhang et al., 2017b; Hurault et al., 2022a), experiments are conducted on the CBSD10 dataset (Martin et al., 2001) with noise level $12.5/255$. We use 8 motion kernels from (Levin et al., 2009), a $9 \times 9$ uniform kernel, and a $25 \times 25$ Gaussian kernel with $\sigma = 1.6$.

- *Inpainting.* Experiments are performed on CBSD68 with $80\%$ of pixels randomly masked and additive Gaussian noise of $\sigma = 1/255$.

- *SISR.* We test RISP and baselines on CBSD10. Each image is processed with an anti-aliasing blur kernel followed by $2\times$ downsampling. A total of ten motion and fixed kernels are used.

- *MRI.* An $8\times$ undersampling scheme is applied to 10 images from the fastMRI dataset (Zbontar et al., 2018); $4\times$ results are provided in Appendix G.2.4. The $k$-space noise level is set to $1/255$. Due to the absence of publicly available MRI-specific models with `SoftPlus`, the score network S uses a GS-DRUNet with `eLu` activations, pre-trained on natural grayscale images.

### G.2.1 DEBLURRING

The image deblurring inverse problem can be formulated as

$$\boldsymbol{y} = \boldsymbol{Ax} + \boldsymbol{n},$$

with $\boldsymbol{n} \sim \mathcal{N}(0, \sigma_y \mathsf{I})$ the additive Gaussian noise and $\boldsymbol{A} = \boldsymbol{F}^\star \boldsymbol{\Lambda} \boldsymbol{F} \in \mathbb{R}^{d \times d}$ the observation matrix where $\boldsymbol{F}$ is the discrete Fourier transform matrix, $\boldsymbol{F}^\star$ its inverse and $\boldsymbol{\Lambda}$ a diagonal matrix. The data-fidelity is then defined by

$$f(\boldsymbol{x}) = \frac{1}{2}\|\boldsymbol{Ax} - \boldsymbol{y}\|^2.$$

And its gradient and proximal operator can be computed in closed-form using the following formula, for $\boldsymbol{x} \in \mathbb{R}^d$ and $\eta > 0$

$$\nabla f(\boldsymbol{x}) = \boldsymbol{F}^\star \boldsymbol{\Lambda}^\star \left(\boldsymbol{\Lambda} \boldsymbol{F} \boldsymbol{x} - \boldsymbol{F} \boldsymbol{y}\right)$$
$$\mathrm{prox}_{\eta f}(\boldsymbol{x}) = \boldsymbol{F}^\star \left(\mathsf{I} + \eta \boldsymbol{\Lambda} \boldsymbol{\Lambda}^\star\right)^{-1} \boldsymbol{F} \left(\boldsymbol{x} + \eta \boldsymbol{A} \boldsymbol{y}\right). \tag{70}$$

For linear degradation, Assumption 2-3-4 on $f$ are verified. However, it is not possible to test in practice the inequality $\nu \leq Cn^{-1/7}\rho^{-2/7}L^{4/7}$. Therefore, we can not ensure that Theorem 2 applies.

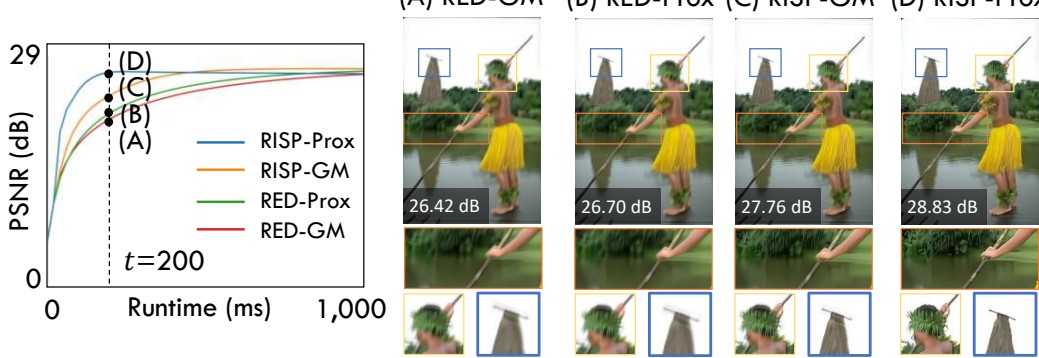

Figure 6: Convergence of RISP methods compared with baselines for deblurring with a noise level of $12.5/255$ averaged on CBSD10, and 10 different blur kernels. Time are given on a NVIDIA A100 Tensor Core GPU. On the right, qualitative restoration of an image after 200 ms. Note that the inertial mechanism allows to accelerate significantly the convergence.

**Experimental set-up.** As in (Zhang et al., 2017b; Hurault et al., 2022a), 10 blur kernels are tested, including the 8 motion kernels proposed by (Levin et al., 2009), $9 \times 9$ uniform kernel and the $25 \times 25$ Gaussian blur kernel with standard deviation 1.6. The noise level is chosen to be $\sigma_y = 12.5/255$. The tested dataset (CBSD10) is composed of a subset of 10 images of the CBSD68 data set (Martin et al., 2001).

Figure 6 shows that RISP-Prox achieves a $5\times$ acceleration over RED-GM. Qualitative comparisons further indicate that RISP-GM and RISP-Prox recover sharper structures within a short time budget (200 ms).

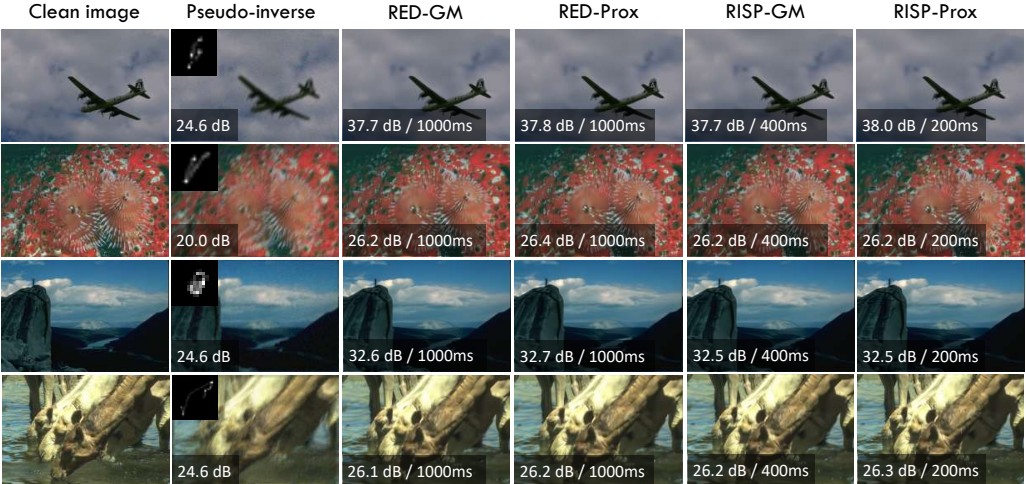

Figure 7: Qualitative results for image deblurring with various methods after their convergence. Time are computed on a NVIDIA A100 Tensor Core GPU.

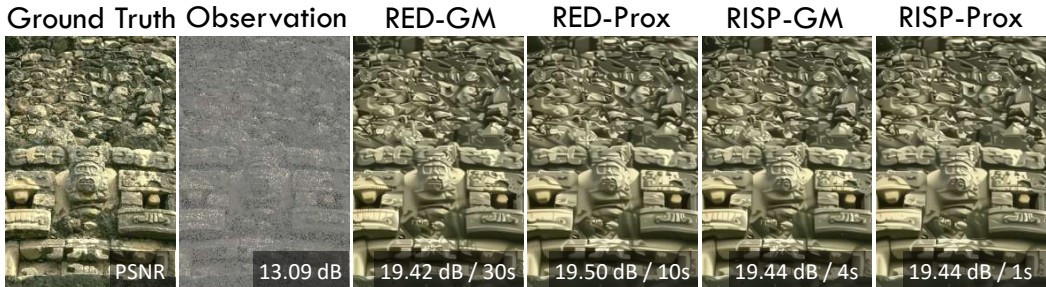

Figure 8: Qualitative results for image inpainting with $80\%$ random masked pixels and a noise level of $\sigma_y = 1/255$ and various methods after their convergence. Time are computed for a NVIDIA A100 Tensor Core GPU.

### G.2.2 INPAINTING

For image inpainting, the forward model can be written as

$$\boldsymbol{y} = \boldsymbol{A}\boldsymbol{x} + \boldsymbol{n},$$

with $\boldsymbol{n} \sim \mathcal{N}(0, \sigma_y \mathsf{I}_d)$ and $\boldsymbol{A} \in \mathbb{R}^{d \times d}$ a diagonal matrix with diagonal coefficients in $\{0, 1\}$. Then the data-fidelity $f(\boldsymbol{x}) = \frac{1}{2}\|\boldsymbol{A}\boldsymbol{x} - \boldsymbol{y}\|^2$ have the following formula to compute its gradient and proximal operator, for $\boldsymbol{x} \in \mathbb{R}^d$ and $\eta > 0$,

$$\nabla f(\boldsymbol{x}) = \boldsymbol{A}\left(\boldsymbol{x} - \boldsymbol{y}\right),$$
$$\mathsf{prox}_{\eta f}(\boldsymbol{x}) = \left(\mathsf{I}_d + \eta \boldsymbol{A}\right)^{-1}\left(\boldsymbol{x} + \eta \boldsymbol{A}\boldsymbol{y}\right).$$

**Experimental set-up.** In our experiments, we tackle the inpainting problem with $80\%$ random missing pixels. The noise level is chosen to be $\sigma_y = 1/255$. The validation set is the entire CBSD68 dataset (Martin et al., 2001).

**Results.** Figure 8 shows the visual results by different methods after convergence. It can be seen that, RISP-GM accelerates RED-GM by $7.5\times$, while RISP-Prox has a $10\times$ acceleration over RED-Prox. This clearly show the efficiency of the proposed RISP methods.

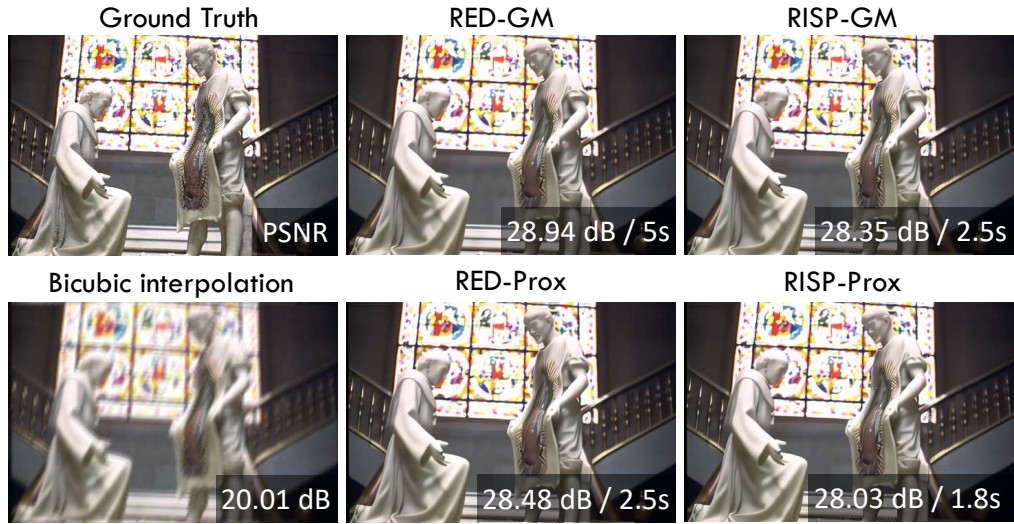

Figure 9: Qualitative results for super-resolution with various methods. On each images, PSNR values and reconstruction time (in second) are provided.

### G.2.3 SINGLE IMAGE SUPER RESOLUTION

For image super-resolution, the forward model can be written as

$$\boldsymbol{y} = \boldsymbol{SHx} + \boldsymbol{n},$$

with $\boldsymbol{H} \in \mathbb{R}^{d \times d}$ the matrix of the convolution with an anti-aliasing kernel of blur, $\boldsymbol{S} \in \mathbb{R}^{m \times d}$ the standard $s$-dowsampling matrix, with $d = s^2 m$ and $\boldsymbol{n} \sim \mathcal{N}(0, \sigma_y^2 \mathsf{I}_m)$. Therefore, we have the following gradients and proximal operator for the data-fidelity $f(\boldsymbol{x}) = \frac{1}{2} \| \boldsymbol{SHx} - \boldsymbol{y} \|$, given by (Zhao et al., 2016), for $\boldsymbol{x} \in \mathbb{R}^d$ and $\eta > 0$,

$$\nabla f(\boldsymbol{x}) = \boldsymbol{H}^T \boldsymbol{S}^T (\boldsymbol{SHx} - \boldsymbol{y})$$

$$\mathsf{prox}_{\eta f}(\boldsymbol{x}) = \boldsymbol{z} - \frac{1}{s^2} \boldsymbol{F}^\star \boldsymbol{\Lambda}^\star \left( \mathsf{I}_d + \frac{\eta}{s^2} \boldsymbol{\Lambda \Lambda}^\star \right)^{-1} \boldsymbol{\Lambda F z},$$

with $\boldsymbol{z} = \boldsymbol{x} + \eta \boldsymbol{H}^T \boldsymbol{S}^T \boldsymbol{y}$, $\boldsymbol{\Lambda}$ a block-diagonal decomposition of the $s \times s$ downsampled matrix in the Fourier domain and $\boldsymbol{F}$ the discrete Fourier transform matrix.

**Experimental set-up.** We study $2\times$ super-resolution with 10 blur kernels introduced in the deblurring experiment with a noise level of $\sigma_y = 1/255$. The tested data set (CBSD10) is composed of a subset of 10 images of the CBSD68 data set (Martin et al., 2001).

### G.2.4 MRI

For Magnetic Resonance Imaging (MRI), the data-fidelity term can be expressed in the same form as in deblurring, with a matrix $\boldsymbol{\Lambda}$ whose diagonal entries belong to $\{0, 1\}$. Consequently, the formulas in (70) remain valid in this setting.

**Experimental set-up.** In our experiments, we tackle the MRI reconstruction problem with $4\times$ and $8\times$ acceleration. The noise level in the $k$ space is chosen to be $\sigma_y = 1/255$. For validation, we use 10 knee images from the FastMRI dataset (Zbontar et al., 2018), following the same protocol as in (Sun et al., 2019a).

**Results.** Figure 10 demonstrates that RISP-GM and RISP-Prox yield comparable qualitative reconstructions on par with RED-GM and RED-Prox, while exhibiting faster convergence. The quantitative results for $4\times$ acceleration are reported in Figure 11. In particular, RISP-Prox is the most efficient among the compared methods.

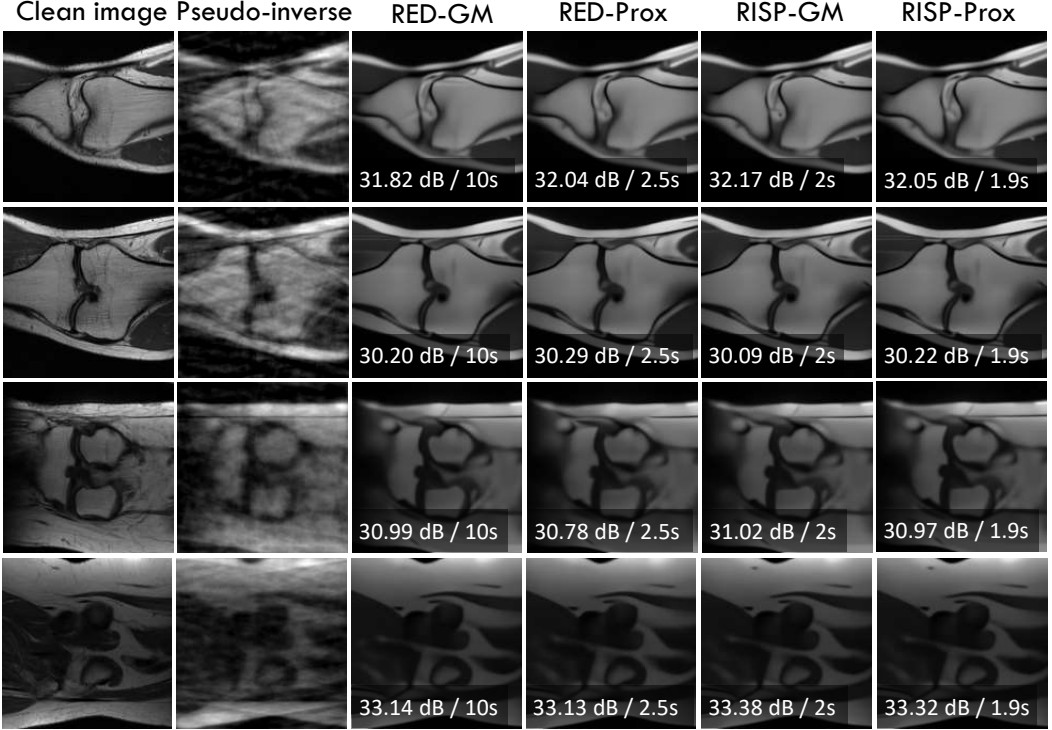

Figure 10: Visual results for $8\times$ MRI reconstruction by RISP and baselines. For each image, PSNR values and reconstruction time (in second) are provided. Compared with the baseline methods, RISP can significantly accelerate the MRI reconstruction, while maintaining the reconstruction quality.

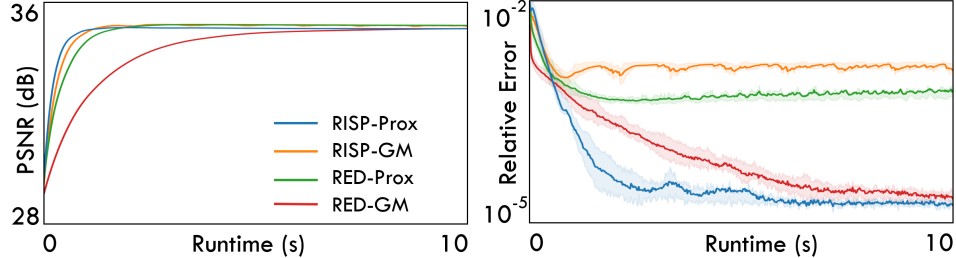

Figure 11: PSNR and relative error curves on with $4\times$ MRI reconstruction by RISP methods and baselines. The colored area are $25\%$-$75\%$ quantiles for the relative errors $\|\boldsymbol{x}^{k+1} - \boldsymbol{x}^k\|/\|\boldsymbol{x}^0\|$.

### G.3  NONLINEAR INVERSE PROBLEM

In this section, we provide additional details for the Rician noise removal task. Let $\boldsymbol{n}_{\mathrm{real}}$ and $\boldsymbol{n}_{\mathrm{imag}}$ denote two *i.i.d.* Gaussian noises with zero mean and standard derivation $\sigma_y$. The Rician noise corrupted observation $y$ is obtained by:

$$\boldsymbol{y} = \sqrt{(\boldsymbol{x} + \boldsymbol{n}_{\mathrm{real}})^2 + \boldsymbol{n}_{\mathrm{imag}}^2}.$$

According to (Gudbjartsson & Patz, 1995), the conditional probability density function (PDF) of $\boldsymbol{y}$ is

$$p(\boldsymbol{y}|\boldsymbol{x}) = \frac{\boldsymbol{y}}{\sigma_y^2} \exp\left(-\frac{\boldsymbol{x}^2 + \boldsymbol{y}^2}{2\sigma_y^2}\right) I_0\left(\frac{\boldsymbol{x} \odot \boldsymbol{y}}{\sigma_y^2}\right),$$

which is known as *Rice* or *Rician* distribution. Here $I_0$ is the modified Bessel function of the first kind with order zero. Consider the modified Bessel's differential equations:

$$x^2 y'' + xy' - (x^2 + n^2)y = 0, n \geq 0.$$

The solutions of the previous equation $I_n(x) = y(x)$ continuous in zero are called the modified Bessel function of the first kind with order $n$.

After omitting some constant terms, the data fidelity term $f$ for Rician noise removal is

$$f(\boldsymbol{x}) = \left\langle \boldsymbol{1}, \frac{\boldsymbol{x}^2}{2\sigma_y^2} - \log I_0\left(\frac{\boldsymbol{x} \odot \boldsymbol{y}}{\sigma_y^2}\right) \right\rangle.$$

Based on the recurrence formulas of the derivative of $I_n$ (Bowman, 2012), we have that $\forall x \in \mathbb{R}$,

$$I_0'(x) = I_1(x), I_1'(x) = \frac{1}{2}(I_0(x) + I_2(x)), I_2(x) = I_0(x) - \frac{2}{x}I_1(x).$$

Thus we have

$$I_1'(x) = I_0(x) - \frac{1}{x}I_1(x).$$

Let $B(x) = (\log I_0(x))' = \frac{I_0'(x)}{I_0(x)}$. Then $B(x) = \frac{I_1(x)}{I_0(x)}$. According to the proof of Proposition 2.1 in (Wei & Li, 2022), we know

$$B'(x) = 1 - \frac{1}{x}B(x) - B^2(x).$$

According to Proposition 2.1 in (Wei & Li, 2022), $B(x)$ is a increasing concave function on $[0, \infty)$ with $B(0) = 0$, $B(\infty) = 1$. $B'(x)$ is a decreasing function with $B'(0) = 0.5$, $B'(\infty) = 0$. Based on this, we further have

$$
\begin{aligned}
B''(x) = & \left(1 - \frac{1}{x}B(x) - B^2(x)\right)' = \frac{1}{x^2}B(x) - \frac{1}{x}B'(x) - 2B(x)B'(x) \\
= & \frac{1}{x^2}B(x) - \frac{1}{x}\left(1 - \frac{1}{x}B(x) - B^2(x)\right) - 2B(x)\left(1 - \frac{1}{x}B(x) - B^2(x)\right) \\
= & -\frac{1}{x} + \left(\frac{2}{x^2} - 2\right)B(x) + \frac{3}{x}B^2(x) + 2B^3(x).
\end{aligned}
$$

In Figure 12, we plot the functions $I_0$, $B$, $B'$, $B''$. It is clear that $B''(x)$ is bounded by $[-0.25, 0]$. Based on this, we now prove that $f$ has a Lipschitz gradient and Hessian.

**Lemma 12.** *Let $I_0$ be the modified Bessel function of the first kind with order zero. Let $\boldsymbol{y}$ be the observation, $f(\boldsymbol{x})$ be the data fidelity term for the Rician noise removal task, that is,*

$$f(\boldsymbol{x}) = \left\langle \boldsymbol{1}, \frac{\boldsymbol{x}^2}{2\sigma_y^2} - \log I_0\left(\frac{\boldsymbol{x} \odot \boldsymbol{y}}{\sigma_y^2}\right) \right\rangle.$$

*Then, $f$ has a Lipschitz gradient and Hessian.*

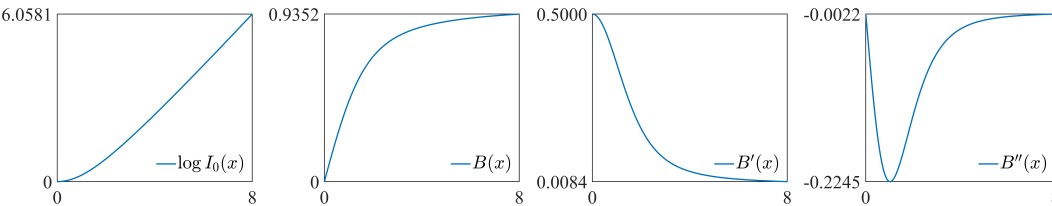

Figure 12: Visualizations of function $I_0$, $B$, $B'$, $B''$.

*Proof.* To prove that $f$ has a Lipschitz gradient and Hessian, we only need to prove it in the one dimensional case, because the function is separable. Thus, we only need to consider $\boldsymbol{x} = x \in [0, \infty)$.

Clearly, we have that

$$f'(x) = \frac{1}{\sigma_y^2} x - \frac{y}{\sigma_y^2} B\left(\frac{xy}{\sigma_y^2}\right).$$

In order to prove that $f'$ is Lipschitz, we only need to prove that $f''$ is bounded.

$$f''(x) = \frac{1}{\sigma_y^2} - \frac{y^2}{\sigma_y^4} B'\left(\frac{xy}{\sigma_y^2}\right).$$

Since $B'$ is bounded, we know that $f$ has a Lipschitz gradient.

To prove the Lipschitz property of $f''$, we only need to prove the boundedness of $f^{(3)}$. Note that

$$f^{(3)}(x) = -\frac{y^3}{\sigma_y^6} B''\left(\frac{xy}{\sigma_y^2}\right),$$

and that $B''$ is bounded by $[-0.25, 0]$, we complete the proof. $\qquad\square$

Based on the above derivations, we have already shown how to compute $\nabla f$:

$$\nabla f(\boldsymbol{x}) = \frac{1}{\sigma_y^2} \boldsymbol{x} - \frac{\boldsymbol{y}}{\sigma_y^2} B\left(\frac{\boldsymbol{x} \odot \boldsymbol{y}}{\sigma_y^2}\right).$$

**Computation of** $\text{prox}_{\eta f}$. Based on the technique used by (Wei et al., 2023), we are able to solve $\text{prox}_{\eta f}$ efficiently by the Iterative Reweighted $L^1$ method (IRL1) given in (Ochs et al., 2015). Consider the objective function of $\text{prox}_{\eta f}$:

$$\text{prox}_{\eta f}(\boldsymbol{z}) = \arg\min_x \frac{1}{2} \|\boldsymbol{x} - \boldsymbol{z}\|^2 + \eta\left\langle \mathbf{1}, \frac{\boldsymbol{x}^2}{2\sigma_y^2} - \log I_0\left(\frac{\boldsymbol{x} \odot \boldsymbol{y}}{\sigma_y^2}\right)\right\rangle.$$

According to the Lemma 2 in (Wei et al., 2023), it can be rewritten as the sum of a convex function $f_1$ and a non-decreasing concave function $f_2$:

$$f_1(\boldsymbol{x}) + f_2(\boldsymbol{x}) = \frac{1}{2}\|\boldsymbol{x} - \boldsymbol{z}\|^2 + \eta\left\langle \mathbf{1}, \frac{\boldsymbol{x}^2}{2\sigma_y^2} - \log I_0\left(\frac{\boldsymbol{x} \odot \boldsymbol{y}}{\sigma_y^2}\right)\right\rangle,$$

where

$$\begin{aligned} f_1(\boldsymbol{x}) &= \eta\left\langle \mathbf{1}, \frac{\boldsymbol{x}^2}{2\sigma_y^2} - \frac{\boldsymbol{x} \odot \boldsymbol{y}}{\sigma_y^2}\right\rangle + \frac{1}{2}\|\boldsymbol{x} - \boldsymbol{z}\|^2, \\ f_2(\boldsymbol{x}) &= \eta\left\langle \mathbf{1}, \frac{\boldsymbol{x}\boldsymbol{y}}{\sigma_y^2} - \log I_0\left(\frac{\boldsymbol{x} \odot \boldsymbol{y}}{\sigma_y^2}\right)\right\rangle. \end{aligned}$$

IRL1 algorithm linearizes the non-convex part $f_2$, and solve the resulted weighted convex subproblem. The IRL1 algorithm for solving $\text{prox}_{\eta f}(\boldsymbol{z})$ is given as follows:

$$\begin{aligned} \boldsymbol{x}^{t+\frac{1}{2}} &= \frac{\boldsymbol{y}}{\sigma_y^2} B\left(\frac{\boldsymbol{x}^t \odot \boldsymbol{y}}{\sigma_y^2}\right) \\ \boldsymbol{x}^{t+1} &= \arg\min_x \eta\left\langle \mathbf{1}, \frac{\boldsymbol{x}^2}{2\sigma_y^2} - \frac{\boldsymbol{x} \odot \boldsymbol{y}}{\sigma_y^2}\right\rangle + \frac{1}{2}\|\boldsymbol{x} - \boldsymbol{z}\|^2 - \eta\langle \boldsymbol{x}^{t+\frac{1}{2}}, \boldsymbol{x}\rangle \\ &= \frac{\boldsymbol{z} + \frac{\eta}{\sigma_y^2}\boldsymbol{y} + \eta\boldsymbol{x}^{t+\frac{1}{2}}}{1 + \frac{\eta}{\sigma_y^2}}. \end{aligned}$$

In experiments, we set the maximum inner iteration number for IRL1 to be 10 because it usually solves the proximal sub-problem within 5 iterations.

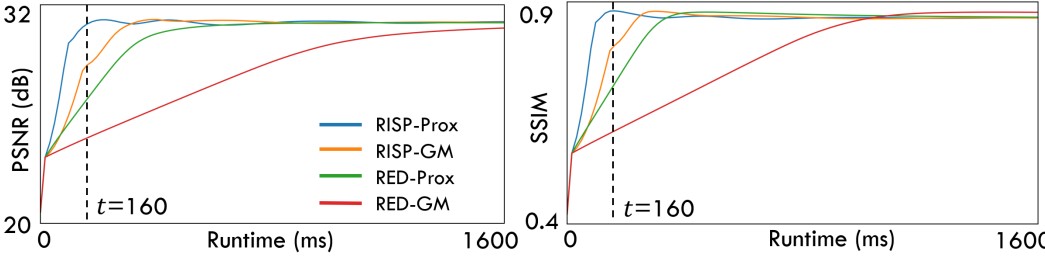

Figure 13: Averaged PSNR and SSIM curves for Rician noise removal with noise level $25.5/255$ on CBSD10 by RISP methods and baselines. The $x$-axis denotes the running time in milliseconds (ms). RISP-Prox converges within a 160 ms time budget.

**Experimental set-up.** We use the same denoiser as in linear inverse problems, and the same test data set (10 images of CBSD68) as in the deblurring experiment. The Rician noise level is set to $\sigma_y = 25.5/255$. The average PSNR and SSIM curves are reported in Figure 13. We observe a clear gain of restarted inertia for acceleration.

**Results.** Figure 13 visualizes the PSNR and SSIM curves against running time in milliseconds for Rician noise removal by RISP and baselines. Note that RISP-Prox takes 160 milliseconds to converge, RISP-GM needs 320 milliseconds, while RED-GM needs 1600 milliseconds. The final PSNR values by RISP aligns closely with RED baselines. This indicates that RISP can accelerate the convergence without compromising the performance.

## G.4 LARGE-SCALE IMAGING

**Forward model.** The linear inverse scattering task aims to reconstruct the permittivity contrast distribution of an object from measurements of its scattered field captured by an array of receivers. In our setup, we consider the first Born approximation (Wolf, 1969) which is commonly adopted in (Zheng et al., 2025; Gao et al., 2025; Sun et al., 2019b).

$$\boldsymbol{y} = \boldsymbol{H}(\boldsymbol{u} \odot \boldsymbol{x}) + \boldsymbol{n}, \tag{71}$$

where $\boldsymbol{x}$ denotes the permittivity contrast distribution, $\boldsymbol{y}$ is the scattered field measured by receivers, $\boldsymbol{H}$ is the discretized Green's function that models the responses of the optical system, $\boldsymbol{u}$ is the light field, $\odot$ denotes for the Hadamard product and $\boldsymbol{n}$ is Gaussian noise.

**Fidelity term.** The corresponding fidelity term is given by

$$f(\boldsymbol{x}) = \frac{\lambda}{2}\|\boldsymbol{H}(\boldsymbol{u} \odot \boldsymbol{x}) - \boldsymbol{y}\|^2. \tag{72}$$

For this fidelity term $f$, Assumptions 2–4 are satisfied.

**Experiment setup.** In this problem, we set 360 receivers and 240 transmitters. The Gaussian noise level is 0.0001, as in (Zheng et al., 2025). The dataset of ODT images was created using the CytoPacq Web Service (Wiesner et al., 2019)[1].

For the denoiser, we use the standard DRUNet architecture as proposed by (Zhang et al., 2021). The training set is generated by the CytoPacq Web Service, which contains 500 cell images with size $384 \times 384$. The denoiser is trained using the Adam optimizer, where we set the learning rate to be $1 \times 10^{-4}$, patch size to be 128, batch size to be 32, denoising strength distributed uniformly in $\sigma \in [0, 30]/255$. We train the denoiser 600 epochs, and it takes about 70 minutes to complete the training process.

**Calculation of** $\mathrm{Prox}_{\eta \mathrm{f}}$. For the computation of $\mathrm{prox}_{\eta f}$, there is no efficient solution. Since the image size is large, the matrix $\mathrm{Diag}(\boldsymbol{u})\boldsymbol{H}^{\mathrm{T}}\boldsymbol{H}\mathrm{Diag}(\boldsymbol{u})$ has size $1024^2 \times 1024^2$. Thus, computing the inverse of the matrix $\lambda \mathrm{Diag}(\boldsymbol{u})\boldsymbol{H}^{\mathrm{T}}\boldsymbol{H}\mathrm{Diag}(\boldsymbol{u}) + I_{1024^2}$ is not practical. In order to approximate $\mathrm{prox}_{\eta f}$ efficiently, we use the AutoGrad toolbox from PyTorch (Paszke et al., 2017) to calculate the

---

[1]https://cbia.fi.muni.cz/simulator

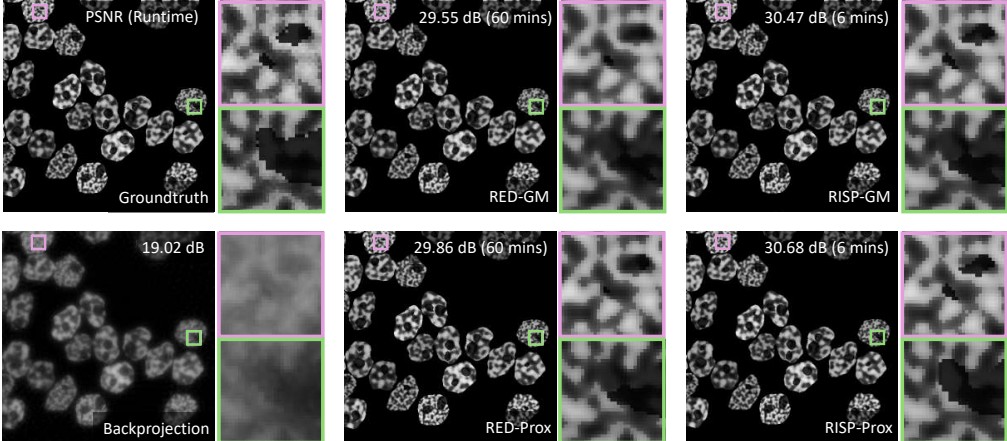

Figure 14: Visual reconstruction comparisons by different methods after convergence or maximum running time reached on linear inverse scattering with 360 receivers, 240 transmitters, and corrupted with 0.0003 Gaussian noises, on the test Cell image of size $512 \times 512$ by RISP methods and baselines.

gradient, and apply a gradient descent inner loop to solve the proximal sub-problem. The objective to compute $\text{prox}_{\eta f}(\boldsymbol{z})$, with $\boldsymbol{z}$ as input, is

$$L(\boldsymbol{x}; \boldsymbol{y}, \boldsymbol{z}) = \frac{\lambda}{2}\|\boldsymbol{H}(\boldsymbol{u} \odot \boldsymbol{x}) - \boldsymbol{y}\|^2 + \frac{1}{2}\|\boldsymbol{x} - \boldsymbol{z}\|^2. \tag{73}$$

---

**Algorithm 6** Inner Loop for $\text{prox}_{\eta f}(\boldsymbol{z})$

---

**Require:** $\boldsymbol{x}^0 = \boldsymbol{z}, t = 0$.
1: **while** $t < T$ and $\|\nabla_x L(\boldsymbol{x}^t; \boldsymbol{y}, \boldsymbol{z})\|^2 \leq \varepsilon$ **do**
2:      $\boldsymbol{x}^{t+1} = \boldsymbol{x}^t - \gamma \nabla_x L(\boldsymbol{x}^t; \boldsymbol{y}, \boldsymbol{z})$
3:      $t = t + 1$
4: **end while**
5: **return** $\boldsymbol{x}^t$

---

We use the gradient descent to update $\boldsymbol{x}$, until convergence, see Algorithm 6 for details. In Algorithm 6, $\boldsymbol{x}^0 = \boldsymbol{z}$ is initialized as the input. The stepsize $\gamma$ is set as $\gamma = 400/(\lambda\eta)$. The maximum iteration number $T = 100$, and $\varepsilon = 2 \times 10^{-3}$.

**Per-iteration runtime for** $1024 \times 1024$ **reconstruction.** For RISP-GM and RED-GM, score evaluation takes approximately 0.43 seconds, while the gradient of the data-fidelity term requires about 1.01 seconds per iteration. For RISP-Prox and RED-Prox, score evaluation again takes roughly 0.43 seconds, whereas the proximal evaluation of the data-fidelity term takes about 2.45 seconds per iteration. Overall, the per-iteration cost is dominated by the data-fidelity computation rather than by the score evaluation.

**Additional results on** $512 \times 512$ **reconstruction.** In Figure 14, we visualize the scattering reconstruction results by RISP and baselines after convergence or the maximum iteration is reached. We can clearly see that after 60 minutes, the zoomed-in parts by RED-GM and RED-Prox are still a bit blurry. While the proposed RISP methods can better reconstruct the fine details in the cells within 6 minutes.

Figure 15 shows the PSNR and SSIM curves by RISP and baselines on inverse scattering. It can be seen that, in general RISP methods exhibit a stable and faster convergence in PSNR and SSIM. Overall, RISP methods have a $10\times$ acceleration against RED-GM and RED-Prox. This verifies the potential of the proposed methods on large-scale problems.

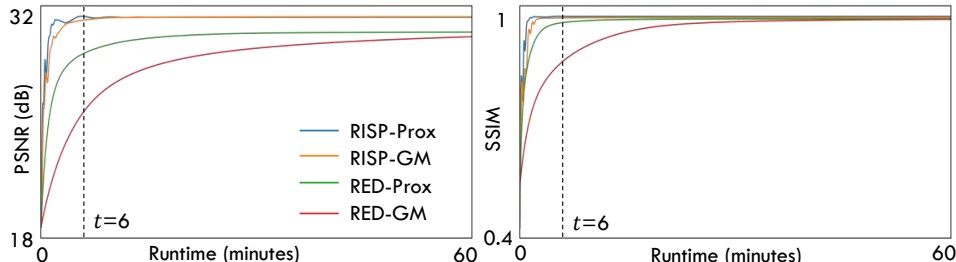

Figure 15: Convergence results on linear inverse scattering with 360 receivers, 240 transmitters, and corrupted with $0.0003$ Gaussian noises, on the test Cell image of size $512 \times 512$ by RISP methods and baselines. $x$-axis denotes the running time in minutes, $y$-axis denotes the PSNR value in dB, or SSIM value.

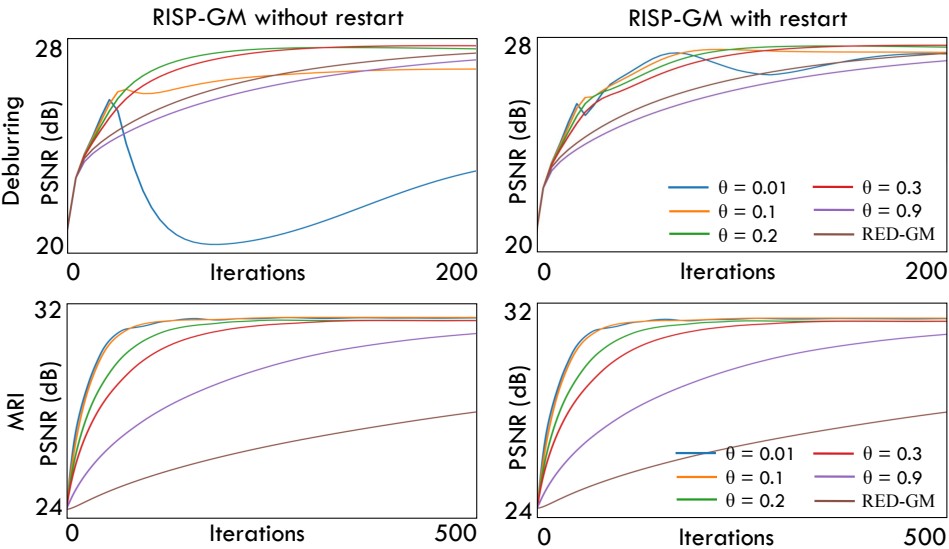

Figure 16: Influence of inertial strength $\theta$ in RISP-GM for deblurring (top row) and MRI (bottom row) tasks. We adopt the same settings and parameters as in Appendix G.2, with a restarting threshold of $B = 5,000$. Note how the restarting mechanism clearly stabilizes the convergence. We also note that the restart criterion may not be always triggered; see MRI results in the second row.

## G.5 HYPERPARAMETER ROBUSTNESS STUDIES

In this section, we examine the influence of the hyperparameters $\theta$ and $B$ on the performance and stability of RISP.

### G.5.1 INFLUENCE OF THE INERTIAL STRENGTH $\theta$.

In RISP, the inertial strength $\theta > 0$ controls the strength of inertia. The choice $\theta = 1$ removes inertia and reduces the RISP algorithm to its RED counterpart, whereas smaller values of $\theta$ increase the inertial effect. Figure 16 and 17 respectively visualize the convergence of RISP-GM and RISP-Prox for $\theta \in \{0.01, 0.1, 0.2, 0.3, 0.9\}$ in linear inverse imaging problems. All RISP algorithms are run with the corresponding settings and parameters as in Appendix G.2, with a fixed restarting threshold of $B = 5,000$. For small values of $\theta$, RISP algorithms *without* restart may become unstable and exhibits oscillatory behavior, as observed in the deblurring or inpainting experiments. In contrast, the restart mechanism effectively stabilizes the convergence. We note that the restart mechanism makes RISP less sensitive to the choice of $\theta$. In particular, values of $\theta$ in the range $[0.1, 0.3]$ yield comparable performance, suggesting this interval as a practical tuning range.

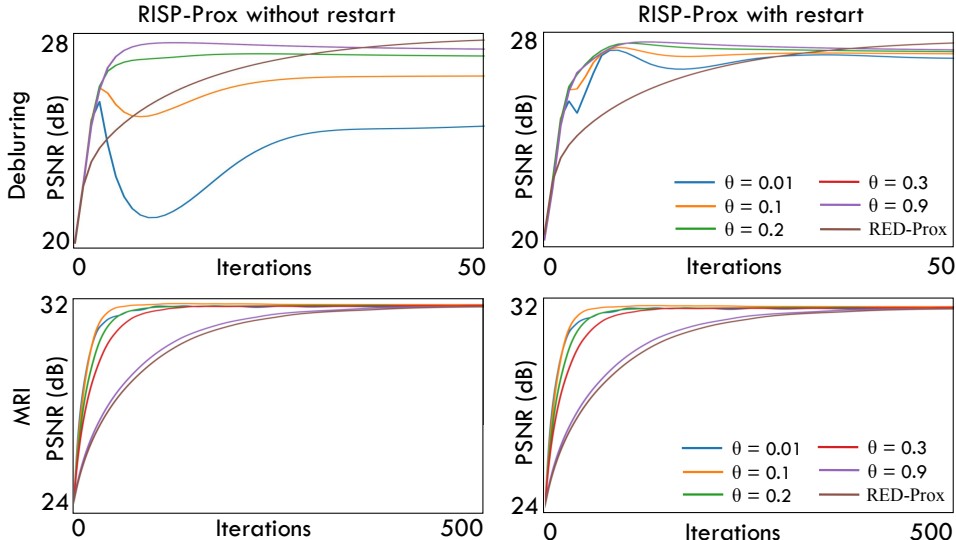

Figure 17: Influence of inertial strength $\theta$ in RISP-Prox for deblurring (first row) and MRI (second row). We adopt the same settings and parameters as in Appendix G.2, with a restarting threshold of $B = 5,000$. Note how the restarting mechanism stabilizes the convergence. We also note that the restart criterion may not be always triggered; see MRI results in the second row.

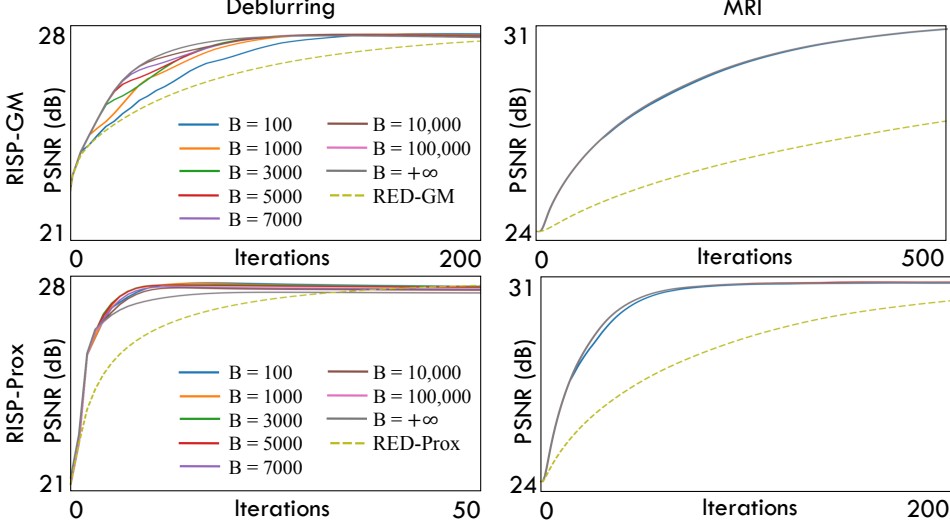

Figure 18: Influence of restarting threshold $B$ in RISP-GM (top row) and RISP-Prox (bottom row) for deblurring (left column) and MRI (right column) tasks. We adopte the corresponding settings and parameters as in Appendix G.2, with two inertial strengths $\theta = 0.2$. As shown in the deblurring results, a small value of $B$ clears inertia more frequently, leading to more stable stable convergence. In contrast, the effect of $B$ is less pronounced in settings such as MRI, where the restart criterion is triggered less often. Note that the behavior of RISP is similar on a large range of $B \in [10^2, 10^6]$.

### G.5.2 INFLUENCE OF THE RESTARTING THRESHOLD $B$.

In RISP, the restarting threshold $B$ determines how frequently inertia is cleared to prevent overshooting. A lower threshold triggers restarts more often, while a higher threshold results in fewer restarts. Figure 18 illustrates the influence of the restarting threshold $B \in \{100, 1000, 3000, 5000, 7000, 10000, 100000, +\infty\}$ in the tasks of image deblurring and MRI, with a fixed inertia strength $\theta = 0.1$. Here, $B = +\infty$ means that the restart mechanism is never triggered. When $B$ is small, restarts occur frequently, leading to more stable behavior. When $B$ is large, the

Table 1: Numerical values obtained by RISP with diffusion-based score prior on image deblurring with $\sigma = 5\%$, using the first blur kernel in Levin's blur kernel dataset. Running time in second and iteration number are reported at the peak PSNR for each method. Note the faster convergence enabled by RISP even if the score function is not strictly a gradient field.

| Methods | PSNR | LPIPS | Running time | Iteration |
|---|---|---|---|---|
| RISP-GM-Diffusion | 27.41 dB | 0.2188 | 5s | 35 |
| RISP-Prox-Diffusion | 27.30 dB | 0.2192 | 3s | 22 |
| RED-GM-Diffusion | 27.46 dB | 0.2267 | 27s | 200 |
| RED-Prox-Diffusion | 27.46 dB | 0.2190 | 5s | 35 |
| DPS | 26.18 dB | 0.1791 | 70s | 1000 |

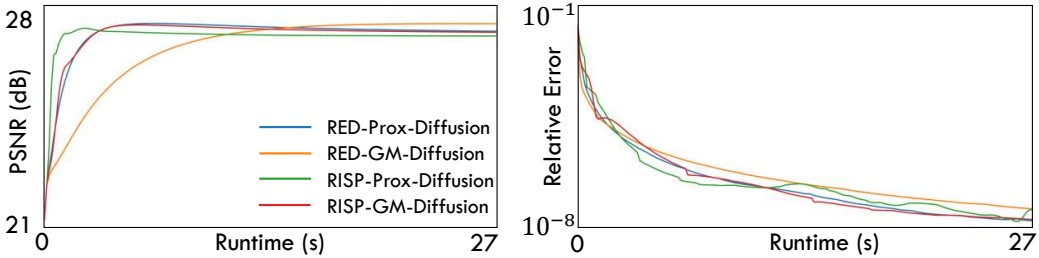

Figure 19: Convergence of RISP with diffusion-based score prior for the deblurring task, with a noise level of $12.5/255$ averaged on CBSD10, and a blur kernel from Levin's dataset (Levin et al., 2009). We observe a behavior of acceleration using RISP, even if using diffusion-based score priors.

restart mechanism is become less active, producing faster trajectories. Overall, the results show that RISP can achieve a good balance between stability and acceleration for a broad range of $B$ values.

### G.6 RISP WITH DIFFUSION-BASED SCORE PRIORS

We further illustrate the benefit of RISP when incorporating diffusion-based score priors. We consider the image deblurring task with $\sigma = 5\%$, using the first blur kernel in Levin's blur kernel dataset. We use the pre-trained score network from the unconditional $256 \times 256$ model publicly released by (Dhariwal & Nichol, 2021)[2].

Table 1 summarizes the numerical results. Running time in second and iteration number are reported at the peak PSNR for each method. As shown, RISP algorithms reach their peak PSNR in fewer iterations and less computational time than the corresponding RED variants. Figure 19 further corroborates the effectiveness by visually comparing the convergence of RISP and RED. These results clearly show that RISP continues to deliver acceleration even when the score function is not strictly a gradient field.

We additionally compare RISP against DPS (Chung et al., 2022a) in terms of performance. To evaluate the perceptual quality, we employ *Learned Perceptual Image Patch Similarity (LPIPS)*. For DPS, we use the recommended default settings and finetune the key hyperparameters. As shown in Table 1, DPS attains lower PSNR but better LPIPS due to its generative sampling nature, whereas RISP offers a more balanced trade-off between reconstruction and perceptual quality.

### G.7 COMPARISON BETWEEN RISP AND FISTA

We compare RISP with another inertia-based method, FISTA-PnP (Kamilov et al., 2018), as summarized in Algorithm 7. The evaluation is conducted on the image deblurring task with noise level $\sigma = 5\%$, using the first blur kernel from Levin's dataset. Both methods employ the same denoiser, GS-DRUNet with `SoftPlus` activations; FISTA-PnP treats the denoiser as a proximal operator on the prior term. Each method is finetuned to attain its best PSNR.

---

[2]https://github.com/openai/guided-diffusion.

---

**Algorithm 7** FISTA-PnP

---

**Require:** $\boldsymbol{y}^0 \in \mathbb{R}^d, n > 0$, and $\sigma, \tau > 0$
1: $k = 0, t_0 = 1$
2: **while** $k < n$ **do**
3:     $\boldsymbol{x}^{k+1} = \mathsf{D}_\sigma(\boldsymbol{y}^k - \tau \nabla f(\boldsymbol{y}^k))$
4:     $t_{k+1} = \left(1 + \sqrt{1 + 4t_k^2}\right)/2$
5:     $\boldsymbol{y}^{k+1} = \boldsymbol{x}^{k+1} + \frac{t_k - 1}{t_{k+1}}(\boldsymbol{x}^{k+1} - \boldsymbol{x}^k)$
6:     $k = k + 1$
7: **end while**
8: **return** $\boldsymbol{x}^n$

---

Table 2: Results by RISP and FISTA-PnP on deblurring with $\sigma = 5\%$ and the first blur kernel in Levin's blur kernel dataset. Running time in second and iteration number are reported at the peak PSNR for each method.

| Methods | PSNR | Running time | Iteration | Convergence? |
|---|---|---|---|---|
| FISTA-PnP | 27.60 dB | 0.5s | 30 | ✗ |
| RISP-GM | 27.66 dB | 0.5s | 29 | ✓ |
| RISP-Prox | 27.55 dB | 0.2s | 12 | ✓ |
| RED-GM | 28.03 dB | 2.5s | 143 | ✓ |
| RED-Rrox | 27.68 dB | 0.5s | 29 | ✓ |

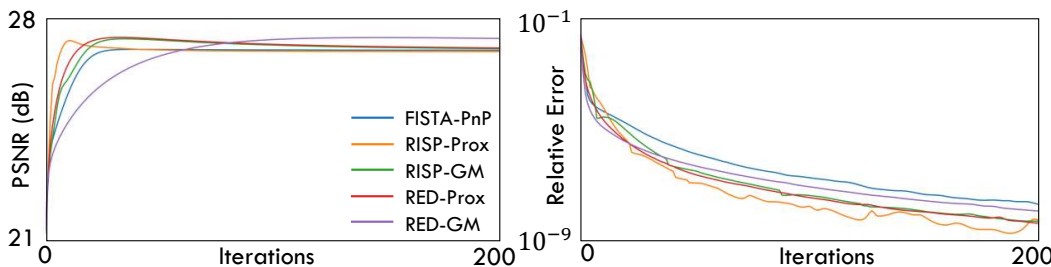

Figure 20: Convergence comparison between RISP and FISTA-PnP.

Table 2 and Figure 20 summerize the results. As shown, RISP-Prox and RISP-GM attains comparable (sometimes even faster) convergence speed with FISTA-PnP. For instance, RISP-Prox takes only 12 iterations to achieve its peak PSNR while FISTA-PnP requires 30 iterations. Note that FISTA-PnP uses the adaptive momentum update and lacks theoretical guarantees.

### G.8 PARAMETER SETTINGS FOR EXPERIMENTS

In Table 3, we give the hyperparameters setting for different inverse problems.

| Inverse problem | $\lambda$ | $\sigma$ | Method | $\eta$ | $\theta$ | $B$ |
|---|---|---|---|---|---|---|
| Deblurring | 15.0 | 0.1 | RED-GM | 0.1 | - | - |
| | | | RED-Prox | 2.0 | - | - |
| | | | RISP-GM | 0.07 | 0.2 | 5000 |
| | | | RISP-Prox | 5.0 | 0.2 | 5000 |
| Inpainting | 5.0 | 0.08 | RED-GM | 0.1 | - | - |
| | | | RED-Prox | 5.0 | - | - |
| | | | RISP-GM | 0.1 | 0.2 | 5000 |
| | | | RISP-Prox | 5.0 | 0.2 | 5000 |
| MRI ($\times 4$) | | 0.01 | RED-GM | 0.7 | - | - |
| | 1.0 | | RED-Prox | 1.0 | - | - |
| MRI ($\times 8$) | | 0.02 | RISP-GM | 0.4 | 0.2 | 5000 |
| | | | RISP-Prox | 1.0 | 0.2 | 5000 |
| SR | 10.0 | 0.03 | RED-GM | 0.4 | - | - |
| | | | RED-Prox | 10.0 | - | - |
| | | | RISP-GM | 0.4 | 0.2 | 5000 |
| | | | RISP-Prox | 10.0 | 0.2 | 5000 |
| Rician Noise Removal | $5 \times 10^{-3}$ | 0.05 | RED-GM | 0.03 | - | - |
| | | | RED-Prox | $5 \times 10^{-4}$ | - | - |
| | | | RISP-GM | 0.03 | 0.01 | 100 |
| | | | RISP-Prox | $5 \times 10^{-4}$ | 0.01 | 100 |
| Linear Inverse Scattering $1024 \times 1024$ | $1 \times 10^5$ | 0.03 | RED-GM | $1 \times 10^{-3}$ | - | - |
| | | | RED-Prox | $5 \times 10^{-3}$ | - | - |
| | | | RISP-GM | $1 \times 10^{-3}$ | 0.01 | $5 \times 10^5$ |
| | | | RISP-Prox | $5 \times 10^{-3}$ | 0.01 | $5 \times 10^5$ |
| Linear Inverse Scattering $512 \times 512$ | $2 \times 10^5$ | 0.03 | RED-GM | $2 \times 10^{-4}$ | - | - |
| | | | RED-Prox | $5 \times 10^{-3}$ | - | - |
| | | | RISP-GM | $2 \times 10^{-4}$ | 0.01 | 5000 |
| | | | RISP-Prox | $5 \times 10^{-3}$ | 0.01 | 5000 |

Table 3: Hyperparameter settings for the experiments.

