# OpenReview forum: "Provably Accelerated Imaging with Restarted Inertia and Score-based Image Priors"
_ICLR.cc/2026/Conference — ICLR 2026 Poster_

### Official Review · Reviewer_pe5V · 2025-10-30

**Soundness:** 3
**Presentation:** 3
**Contribution:** 2
**Rating:** 6
**Confidence:** 3

**Summary:**

This paper develops an acceleration strategy, namely RISP (Restarted Inertia with Score-based Priors), for the Regularization-by-Denoising (RED) framework in imaging inverse problems. Two variants are proposed: RISP-GM (gradient-style) and RISP-Prox (proximal-style), which incorporate an inertial/momentum step and a restart criterion into RED-like updates, and utilize a pretrained score network as the prior. Under assumptions (score is a gradient field, Lipschitz gradient, and Lipschitz Hessian of data fidelity and score), the authors prove accelerated stationary-point convergence rates of $O(1/n^{\frac{4}{7}})$ for both variants. They connect the discrete methods to a restarted heavy-ball ODE and validate empirically on linear and nonlinear inverse problems, demonstrating substantial iteration and runtime speedups while improving reconstruction quality.

**Strengths:**

- The paper identifies restarting inertia as a theoretically justifiable mechanism to accelerate RED-style algorithms while controlling instability, and presents concrete algorithmic variants.
- The derivation of $O(1/n^{\frac{4}{7}})$ convergence under nonconvex score priors is a meaningful advancement in the analysis of accelerated RED/PnP-style methods.
- The connection to a restarted heavy-ball ODE gives additional intuition and helps explain algorithmic behavior.
- The empirical section shows applications to linear and nonlinear inverse problems, showing both acceleration benefits and image quality improvement in reconstruction.
- The algorithms are simple to implement on top of existing RED/PnP solvers: the inertia and restart rules are lightweight and practical.

**Weaknesses:**

- The accelerated rates require Lipschitz continuity of Hessians (Assumption 3) and that the score be an exact gradient field (Assumption 1). While the paper discusses how these can be approximated in practice (e.g., gradient-step denoisers, Lipschitz activations), a more explicit discussion and empirical diagnostics showing how violation of these assumptions affects performance would strengthen the contribution.
- RISP introduces several new hyperparameters (e.g., the inertia parameter and restart threshold). The paper reports robustness in experiments, but lacks a systematic ablation or guidance for selecting these values in practice. An appendix table or a small experiment showing sensitivity would help practitioners.
- The restart trigger is based on accumulated relative change; it would be useful to compare this choice against other restart heuristics (e.g., function value increase, gradient norm increase, or adaptive restart strategies). A short ablation would clarify whether the theoretical restart choice is critical.
- The manuscript cites inertia/acceleration prior works, but could provide a more direct empirical and conceptual comparison to stronger recent baselines that incorporate momentum or acceleration (e.g., PnP-FISTA, PnP Quasi-Newton, and extrapolated three-operator splitting) beyond baseline RED-GM/RED-Prox.
1. PnP-FISTA: Kamilov et al., “Plug-and-Play FISTA for Solving Nonlinear Imaging Inverse Problems”.
2. PnP-QN: Tan et al., “Provably Convergent Plug-and-Play Quasi-Newton Methods”.
3. Wu et al., “Extrapolated Plug-and-Play Three-Operator Splitting Methods for Nonconvex Optimization with Applications to Image Restoration”.
-Some of the dramatic runtime gains in the large-scale experiment are attributed to the high per-iteration cost of RED variants; however, the per-iteration complexity and GPU/CPU implementation details (e.g., whether prox evaluation or score computation dominates) are not clearly reported.

**Questions:**

- Can you quantify how violations of Assumptions 1–3 affect the convergence rate empirically? For example, how does RISP perform when the score is not exactly a gradient field?
- Did you compare the chosen restart rule (accumulated relative change) to other restart schemes (e.g., O’Donoghue–Candes adaptive restart or gradient-based restart)? Can you add a discussion, if not a full-blown comparison with other known heuristics for restart?
- For large-scale inverse scattering, you use DRUNet and patch processing. How sensitive are results to the patching strategy, and does patch-based score evaluation affect theoretical assumptions (e.g., smoothness)?
- The paper states code will be released upon acceptance. Please ensure the repo includes (i) exact pretrained models used, (ii) scripts for reproducing key plots (gradient-norm decay, PSNR vs runtime), and (iii) detailed hyperparameter selection modules.

---

> ### Author Response · Authors · 2025-11-25
>
> **[Weakness 1]** We appreciate the reviewer for pointing out a potential confusion.
> -  First, Assumption 1 can be satisfied by using the gradient-step denoiser [1], which is given by $$g_\sigma(x) = \frac{1}{2\sigma^2}|| x - N_\sigma(x) ||^2 \quad\text{and}\quad S(x) = -\nabla g_\sigma(x) $$ where the regularizer $g_\sigma(x)$ is explicitly defined by a pre-trained denoiser, and the score function is its negative gradient. In practice, we used $\texttt{AutoDiff}$ to compute the gradient. To ensure Assumption 2 & 3 (Lipschitz gradient and Hessian), we use $\texttt{SoftPlus}$ activation functions in $N_\sigma(x)$. We mathematically show that this can guarantee these assumptions. We refer to Proposition 2 in Appendix G.1 for a detailed derivation.
>
>    We kindly note that our experiments also progressively show how RISP perform as fewer assumptions hold. In linear inverse problems (Section 5.1, except for MRI), all assumptions are satisfied. In nonlinear inverse problems (Section 5.2), Assumption 3 is violated by the data-fidelity term. In large-scale inverse scattering (Section 5.3), both Assumption 1 & 3 are violated. Across all settings, the results consistently show stable acceleration achieved by the RISP algorithms.
>
> - Prompted by this remark, we also conducted a new experiment to demonstrate the compatibility of RISP with pre-trained score functions used in diffusion models. The results are summarized in the following table.
>
>     **Table:**  Results on deblurring with $\sigma=5\%$ and the first blur kernel in Levin's blur kernel dataset. Running time in second and iteration number are reported at the peak PSNR for each method.
>     Methods | PSNR ($\uparrow$) | LPIPS ($\downarrow$) | Running time | Iteration
>     ---| ---|---|---|---
>     RISP-GM-Diffusion |	27.41 dB	|0.2188	|5s |	35
>     RISP-Prox-Diffusion	|27.30 dB	|0.2192	|3s	| 22
>     RED-GM-Diffusion	|27.46 dB	|0.2267	|27s	|200
>     RED-Prox-Diffusion	|27.46 dB	|0.2190	|5s	|35
>
>     The results clearly show that RISP continues to deliver acceleration even when the score function is not a gradient field. Additional convergence plots and further discussion are provided in Appendix G.6.
>
>
> **[Weakness 2]** In the revised Appendix G.5, we provided a detailed discussion on the robustness of both RISP algorithms to the restarting hyperparameters. We plot the convergence behavior across a range of momentum parameters $\theta \in [0.01, 1]$ as well as restart thresholds $B \in [100, 100000]$. Overall, the results show that RISP exhibits stable acceleration across a sufficiently wide range of $(\theta, B)$ combinations. We also note that smaller $\theta$ suggests stronger acceleration at the cost of increased oscillation, whereas smaller $B$ induces more frequent restarting, stablizing the convergence but tempering acceleration. In our experiments, we manually select appropriate values of $\theta$ and $B$ following this principle. Table 3 in the appendix summarizes the hyperparameter settings for each experiment presented in the main manuscript.
>
> **[Weakness 3]** We agree that a wide range of restart criteria exist, and we discuss several of them in Appendix A under “Restarting methods”, including speed-based restart rules [2], fixed-interval restarts [3], and hybrid strategies that interpolate between multiple criteria [4]. However, to the best of our knowledge, these alternative restart mechanisms do not come with theoretical guarantees in nonconvex settings [5,6]. While they may still function in practice, the scope of a direct comparison between RISP and approaches lacking theoretical guarantees remains limited.
>
> **[Weakness 4]** Prompted by the reviewer's remark, we conducted additional comparison between RISP and FISTA-PnP [7]. The table below summarizes the results for image deblurring.
>
> **Table**: Comparison between FISTA-PnP, RISP, and RED methods on deblurring with $\sigma=5\%$ and the first blur kernel in Levin's blur kernel dataset. Iteration number is reported at the peak PSNR value. The same learned denoiser GS-DRUNet is used in all methods.
> Methods | PSNR ($\uparrow$) | Running time | Iteration number | Convergence Guarantee
> ---|---|---|---|---
> FISTA-PnP | 27.60 dB | 0.5s | 30 | $\times$
> RISP-Prox | 27.55 dB | 0.2s | 12 | $\checkmark$
> RISP-GM | 27.66 dB | 0.5s | 29 | $\checkmark$
> RED-Prox | 27.68 dB | 143 | 29 | $\checkmark$
> RED-GM | 28.03 dB | 2.5s | 142 | $\checkmark$
>
> As shown, RISP-Prox and RISP-GM attains comparable (sometimes even faster) convergence speed with FISTA-PnP. Note that FISTA-PnP uses the adaptive momentum update and lacks theoretical guarantees. Convergence plot as well as detailed discussion is presented in Appendix G.6. We have also added citations for FISTA-PnP, PnP-Quasi-Newton, and PnP-DYS in Line 124 of the revised manuscript.

---

> > ### Author Response · Authors · 2025-11-25
> >
> > **[Weakness 5]** We report the per-iteration cost as follows:
> > - RISP-GM / RED-GM: score evaluation is about 0.43 second; gradient of the data-fidelity term is about 1.01 second per iteration.
> > - RISP-Prox / RED-Prox: score evaluation is about 0.43 second; proximal evaluation of the data-fidelity term is about 2.45 second per iteration.
> >
> > Note that all methods use the same DRUNet denoiser as score prior. Overall, the per-iteration cost is dominated by the data-fidelity computations rather than the score evaluations. We have included these details in the revised Appendix G.4.
> >
> > **[Question 3]** We agree that the DRUNet denoiser used in the inverse scattering experiment does not satisfy Assumptions 1–3. However, we observe that RISP continues to enable stable, fast convergence. This behavior is also corroborated by the additional results shown in Appendix G.4, where the problem scale is $512 \times 512$.
> >
> > Furthermore, as discussed in our response to Weakness 1 and in Appendix G.6, RISP also remains robust when equipped with a pre-trained diffusion-based score prior, which violates these assumptions. Together, these results demonstrate that RISP continues to converge reliably even when the theoretical assumptions are not strictly met.
> >
> > **[Question 4]** We appreciate the reviewer’s suggestions and will follow them when releasing our code.
> >
> >
> > **Reference**
> >
> > [1] Samuel Hurault, Arthur Leclaire, and Nicolas Papadakis. Gradient step denoiser for convergent
> > plug-and-play. In International Conference on Learning Representations (ICLR’22), 2022a.
> >
> > [2]  A Differential Equation for Modeling Nesterov’s Accelerated Gradient Method: Theory and Insights. Su. W. J., Boyd. S. P., Candès. E. J.
> >
> > [3] Efficient methods in convex programming, Nemirovski A.
> >
> > [4] A continuous-time model to interpolate between speed and function value restart in accelerated first order methods, J. J. Maulen, H. Guo, J. Peypouquet.
> >
> > [5] Accelerated Gradient Methods with Gradient Restart: Global Linear Convergence, C. Bao, L. Chen, J. Li, Z. Shen.
> >
> > [6] Parameter-Free FISTA by Adaptive Restart and Backtracking, J.F. Aujol, C. Dossal, H. Labarriere, A. Rondepierre.
> >
> > [7] A plug-and-play priors approach for solving nonlinear imaging inverse problems, U Kamilov, H Mansour, B E Wohlberg, IEEE Signal Processing Letters, 2018

---

### Official Review · Reviewer_UnZv · 2025-10-30

**Soundness:** 4
**Presentation:** 3
**Contribution:** 2
**Rating:** 4
**Confidence:** 2

**Summary:**

This paper proposes to accelerate the convergence of the regularization by denoising (RED) framework with score-based priors by incorporating momentum and restart strategies. The authors provide theoretical convergence rate guarantees and extend the analysis to a continuous-time dynamical system formulation. Both the theoretical convergence analysis and empirical experiments demonstrate faster convergence compared to the original RED framework, despite the nonlinearity of score-based priors.

**Strengths:**

1. The theoretical justification for accelerated convergence is rigorous and carefully presented. In particular, it is valuable that the analysis is conducted under non-convex score-based prior conditions, which are more realistic in modern generative models.
2. The method builds upon the well-studied RED framework and introduces modifications that are principled and consistent with its formulation.
3. The proposed changes require only a few additional lines of code, making the method highly practical. This simplicity also suggests that the proposed acceleration could potentially be applied to inverse problem solvers that use diffusion model sampling as a denoiser, not only traditional RED implementations.

**Weaknesses:**

1. The novelty of the proposed method feels somewhat modest. Momentum and restart strategies are widely used in optimization, so the main contribution here seems to lie in establishing acceleration guarantees under non-convex score-based priors. While the theoretical development is sound, the overall structure of the convergence analysis appears similar to existing frameworks for accelerated non-convex optimization. It is not entirely clear how substantially new the analytical perspective is.
2. While momentum can accelerate the early stages of optimization, it may also introduce oscillations or instability, particularly in nonlinear or ill-conditioned settings. In the experiments presented, there are cases where the final reconstruction quality is actually worse than the baseline without momentum. This suggests that careful hyperparameter tuning may be necessary, and that the practical benefits of acceleration may not always be realized at convergence.
3. Given that the original RED framework is rarely used in contemporary practice—having largely been superseded by more general PnP and score-based inverse problem formulations—the practical impact of the contribution may be somewhat limited, even if the analysis itself is well executed.

**Questions:**

Please refer to the Weaknesses section for the main points of clarification and concerns.

---

> ### Author Response · Authors · 2025-11-25
>
> **[Weakness 1]** We believe our work carries sufficient novelty from both technical and conceptual perspectives.
> - **Technical novelty in the analysis of RISP-Prox.** Our theoretical contribution extends the framework of [2] in two important directions. First, proximal variants of score-based methods are beneficial in practice, but extending convergence guarantees from the gradient-based setting to the proximal one is not straightforward. The proof in [2] relies on the eigenvalue decomposition, leading to a quadratic approximation that becomes separable after a change of basis. For RISP-Prox, however, the appropriate diagonalization differs, and full separability is lost. This requires substantial additional technical care to recover guarantees analogous to those in [2] (see Appendix E.4), and is not a routine extension.
> - **Novel continuous-time viewpoint.** We introduce a restarted Heavy Ball ODE as the continuous-time analog of the RISP algorithms. This perspective allows us to interpret RISP-GM and RISP-Prox as different discretizations of the underlying Heavy Ball dynamics, offerring additional insights into the algorithmic behavior. Moreover, we establish the convergence of the restarted Heavy Ball ODE in the non-convex setting, which, to the best of our knowledge, is novel.
> - **Contribution to RED and score-based priors.** While restarted inertia has been studied in non-convex optimization (see review in Appendix A), its integration with regularization by denoising (RED) to develop a provably accelerated algorithm compatible with learned score priors is new. Prior works attempting to establish acceleration under score-based priors rely on more restrictive assumptions (*e.g.*, linear denoisers [1]), whereas our results hold for practical, state-of-the-art learned priors.
>
> **[Weakness 2]** We agree that the momentum mechanism can introduce instabilities (see our discussion in Section 3). In fact, RISP incorporates the restarting mechanism to address the problem in both practice and theory. In the revised Appendix G.5, we conducted new experiments to demonstrate the robustness of both RISP algorithms to the restarting hyperparameters. We plot the convergence behavior across a range of momentum parameters $\theta \in [0.01, 1]$ as well as restart thresholds $B \in [100, 100000]$. Overall, the results show that RISP exhibits stable acceleration across a sufficiently wide range of $(\theta, B)$ combinations, without sacrificing performance. Table 3 in the appendix summarizes the hyperparameter settings for each experiment presented in the main manuscript.
>
> We acknowledge that RISP may converge to a different fixed point than RED. However, the goal of this work is not to show that RISP eventually reaches a “better" fixed point (e.g., in terms of PSNR), but to establish its provably faster convergence compatible with complex score-based image priors. As shown in the experiments, this advantage is particularly beneficial when the time budget is limited or when the problem is computationally intensive (see Section 5.2 & 5.3).
>
> **[Weakness 3]** First, RED is a popular framework that has been applied to a wider range of imaging applications, including image restoration [4], computational microscopy [5,6], magnetic resonance imaging [7], and computed tomography [8]. We kindly note that the original RED paper [4] has already collected over one thousand citations, underscoring its enduring impact within the field. Second, the frameworks of RED and PnP are deeply connected; many prior works have studied their equivalence under different settings [9-12]. Hence, progress in one framework naturally informs and strengthens the other, rather than diminishing the relevance. As noted in the manuscript, the RISP methodology could also be extended to PnP settings.

---

> > ### Author Response · Authors · 2025-11-25
> >
> > **Reference**
> >
> > [1] FISTA Iterates Converge Linearly for Denoiser-Driven Regularization, A. Sinha, KN. Chaudhury, SIAM Journal on Imaging Sciences, 2025
> >
> > [2] Huan Li and Zhouchen Lin. Restarted non convex accelerated gradient descent: No more polylogarithmic factor in the o(epsilonˆ(-7/4)) complexity. Journal of Machine Learning Research, 24(157):1–37,2023.
> >
> > [3] A  second-order  gradient-like  dissipative  dynamical  system  with  hessian-driven  damping, F.  Alvarez,  H.  Attouch,  J.  Bolte,  and  P.  Redont. 2002
> >
> > [4] Romano, Yaniv, Michael Elad, and Peyman Milanfar. "The little engine that could: Regularization by denoising (RED)." SIAM journal on imaging sciences 10, no. 4 (2017): 1804-1844.
> >
> > [5] Metzler, C. A. et al. prDeep: robust phase retrieval with a flexible deep network. in Proceedings of the 35th International Conference on Machine Learning 3498–3507 (PMLR, 2018).
> >
> > [6] Kimanius, D., Jamali, K., Wilkinson, M. E., Lövestam, S., Velazhahan, V., Nakane, T., & Scheres, S. H. (2024). Data-driven regularization lowers the size barrier of cryo-EM structure determination. Nature Methods, 21(7), 1216-1221.
> >
> > [7] Online deep equilibrium learning for regularization by denoising, J Liu, X Xu, W Gan, U Kamilov, Neurips 2022
> >
> > [8] Perelli, Alessandro, and Martin S. Andersen. "Regularization by denoising sub-sampled Newton method for spectral CT multi-material decomposition." Philosophical Transactions of the Royal Society A 379, no. 2200 (2021): 20200191.
> >
> > [9] E. T. Reehorst and P. Schniter, “Regularization by denoising: Clarifications and new interpretations,” IEEE Trans. Comput. Imag., vol. 5, no. 1, pp. 52–67, Mar. 2019.
> >
> > [10] Bayesian imaging using plug \& play priors: when langevin meets tweedie, R Laumont, VD Bortoli, A Almansa, J Delon, Alain Durmus, Marcelo Peyrera, SIAM Journal on Imaging, 2022 - SIAM
> >
> > [11] Sun, Yu, Zihui Wu, Yifan Chen, Berthy T. Feng, and Katherine L. Bouman. "Provable probabilistic imaging using score-based generative priors." IEEE Transactions on Computational Imaging (2024).
> >
> > [12] Liu, Jiaming, Salman Asif, Brendt Wohlberg, and Ulugbek Kamilov. "Recovery analysis for plug-and-play priors using the restricted eigenvalue condition." Advances in Neural Information Processing Systems 34 (2021): 5921-5933.

---

> > > ### Comment · Reviewer_UnZv · 2025-11-27
> > >
> > > After considering the authors’ rebuttal, I have revisited the significance of the paper, and I now find the contribution sufficiently compelling for acceptance. In particular, the extensive results from various large-scale experiments, along with the additional analyses provided in the rebuttal, convincingly support the practical importance of the proposed approach. Therefore, I am increasing my score.

---

### Official Review · Reviewer_2V7N · 2025-10-31

**Soundness:** 3
**Presentation:** 3
**Contribution:** 3
**Rating:** 6
**Confidence:** 4

**Summary:**

This paper proposes a method called Restarted Inertia with Score-based Priors (RISP), which extends the popular RED framework by incorporating momentum and a restart mechanism to accelerate convergence in imaging inverse problems. The authors provide non-convex convergence guarantees for two variants—RISP-GM and RISP-Prox—and connect the discrete algorithms to a continuous-time heavy-ball ODE. Experiments on both linear and nonlinear inverse problems demonstrate improved convergence speed and competitive reconstruction quality compared to RED baselines.

**Strengths:**

- The paper provides a rigorous non-convex convergence analysis for RISP, establishing an improved rate under Lipschitz-continuous Hessian assumptions. This is a meaningful theoretical advance in the analysis of accelerated methods for imaging inverse problems.
- The combination of inertia and restarting is well-motivated and effectively addresses overshooting and instability in non-convex settings. The connection to the heavy-ball ODE offers valuable continuous-time intuition.
- The authors thoroughly evaluate RISP on a range of imaging tasks, including linear and nonlinear inverse problems. The results consistently show faster convergence without sacrificing reconstruction quality, and the method scales well to large images

**Weaknesses:**

- The paper does not adequately discuss the recent surge of PnP-like methods that use pre-trained diffusion models as score-based priors, such as [1]–[7]. In particular, [7] (A Variational Perspective on Solving Inverse Problems with Diffusion Models) also combines diffusion models with a RED-like framework. A comparison or discussion of how RISP could integrate such powerful priors is missing.

- Another related work in PnP literature is missing [8], which also shows the extension to the RED framework.

- While the experiments are comprehensive, they are limited to traditional denoisers (DRUNet). It would be valuable to see how RISP performs when combined with state-of-the-art diffusion-based score functions, especially since such models have shown superior performance in many inverse problems.

- The method introduces new hyperparameters (theta and B), and while the authors claim robustness, no systematic ablation study is provided to support this claim across different tasks.

### References
[1] DIFFUSION POSTERIOR SAMPLING FOR GENERAL NOISY INVERSE PROBLEMS, ICLR 2023
[2] Improving Diffusion Models for Inverse Problems using Manifold Constraints, Neurips 2022
[3] Denoising diffusion models for plug-and-play image restoration, CVPRW 2023
[4] zero-shot image restoration using denoising diffusion null-space model, ICLR 2023
[5] Diffusion Posterior Sampling for Linear Inverse Problem Solving: A Filtering Perspective, ICLR 2024
[6] Pseudoinverse-Guided Diffusion Models for Inverse Problems, ICLR 2023
[7] A Variational Perspective on Solving Inverse Problems with Diffusion Models, ICLR 2024
[8] TFPnP: Tuning-free Plug-and-Play Proximal Algorithms with Applications to Inverse Imaging Problems, JMLR 2022

**Questions:**

See [Weaknesses]

---

> ### Author Response · Authors · 2025-11-25
>
> **[Weakness 1]** We agree with the reviewer that using pre-trained diffusion models as image priors is an important research direction. We note that RISP and [1–7] are fundamentally different in methodology: RISP is built upon the RED formulation and performs the maximum a posteriori (MAP) estimation, whereas [1–7] focus on posterior sampling. The former results in an optimization-based scheme, while the latter derives sampling-based algorithms. Prompted by the reviewer’s remark, we have added a new paragraph in Section 2 of the revised manuscript to review these references.
>
> Additionally, we conducted a new experiment to demonstrate the compatibility of RISP with the score network used in pre-trained diffusion models. The following table summarizes our results on image deblurring.
>
> **Table:** Results on deblurring with $\sigma=5\%$ and the first blur kernel in Levin's blur kernel dataset. Running time in second and iteration number are reported at the peak PSNR for each method.
> Methods | PSNR ($\uparrow$) | LPIPS ($\downarrow$) | Running time | Iteration
> ---| ---|---|---|---
> RISP-GM-Diffusion |	27.41 dB	|0.2188	|5s |	35
> RISP-Prox-Diffusion	|27.30 dB	|0.2192	|3s	| 22
> RED-GM-Diffusion	|27.46 dB	|0.2267	|27s	|200
> RED-Prox-Diffusion	|27.46 dB	|0.2190	|5s	|35
>
> When equipped with the diffusion-based score prior, RISP still reaches its best performance in fewer iterations and less computational time than the corresponding RED variants. This is further visualized in Appendix G.6, where we plot the convergence curves by RISP and RED with diffusion-based prior. Our results highlight the compatibility of RISP with pre-trained score networks.
>
> **[Weakness 2]** We thank the reviewer for bringing [8] to our attention. Parameter tuning is indeed an important practical challenge, and we have added this reference into our PnP introduction (line 102).
>
> **[Weakness 3]** We compared the performance of RISP equipped with diffusion-based score and GS-DRUNet. The following table summarizes the results on image deblurring.
>
> **Table:** Comparison of GS-DRUNet and diffusion-based score prior on deblurring with $\sigma=5%$ using the first kernel from Levin’s dataset. Running time in second and iteration number are reported at the peak PSNR for each method.
> Methods | PSNR ($\uparrow$) | LPIPS ($\downarrow$) | Running time | Iteration
> ---| ---|---|---|---
> RISP-GM-Diffusion |	27.41 dB	|0.2188	|5s |	35
> RISP-Prox-Diffusion	|27.30 dB	|0.2192	|3s	| 22
> RISP-GM-GSDRUNet | 27.66 dB | 0.2660 | 0.5s | 29
> RISP-Prox-GSDRUNet | 27.55 dB | 0.2627 | 0.2s | 12
> DPS	|26.18 dB |	0.1791	|70s	|1000
>
> We observe that both priors yield comparable PSNR values, with GS-DRUNet achieving slightly higher PSNR values. However, the diffusion-based prior offers substantially better perceptual quality, as reflected by its much lower LPIPS scores. We additionally include DPS [1] as a baseline with its hyperparameters finetuned. Note that DPS attains lower PSNR but better LPIPS due to its generative sampling nature, whereas RISP offers a more balanced trade-off between reconstruction and perceptual quality.
>
> **[Weakness 4]** In the revised Appendix G.5, we provided a systematic study on the sensitivity of both RISP algorithms to the restarting hyperparameters. We plot the convergence behavior across a range of momentum parameters $\theta \in [0.01, 1]$ as well as restart thresholds $B \in [100, 100000]$. Overall, the results show that RISP exhibits stable acceleration across a sufficiently wide range of $(\theta, B)$ combinations. Table 3 in the appendix summarizes the hyperparameter settings for each experiment presented in the main manuscript.
>
> **References**
>
> [1] DIFFUSION POSTERIOR SAMPLING FOR GENERAL NOISY INVERSE PROBLEMS, ICLR 2023
>
> [2] Improving Diffusion Models for Inverse Problems using Manifold Constraints, Neurips 2022
>
> [3] Denoising diffusion models for plug-and-play image restoration, CVPRW 2023
>
> [4] zero-shot image restoration using denoising diffusion null-space model, ICLR 2023
>
> [5] Diffusion Posterior Sampling for Linear Inverse Problem Solving: A Filtering Perspective, ICLR 2024
>
> [6] Pseudoinverse-Guided Diffusion Models for Inverse Problems, ICLR 2023
>
> [7] A Variational Perspective on Solving Inverse Problems with Diffusion Models, ICLR 2024
>
> [8] TFPnP: Tuning-free Plug-and-Play Proximal Algorithms with Applications to Inverse Imaging Problems, JMLR 2022

---

> > ### Comment · Reviewer_2V7N · 2025-11-26
> >
> > I'd love to thank authors for their detailed responses to my prior concerns. All my questions have been settled. I vote for acceptance for this paper.

---

### Official Review · Reviewer_wKJo · 2025-10-31

**Soundness:** 3
**Presentation:** 3
**Contribution:** 3
**Rating:** 8
**Confidence:** 4

**Summary:**

This paper proposes Restarted Inertia with Score-based Priors (RISP), an acceleration framework built upon the RED (Regularization by Denoising) methodology for solving imaging inverse problems. The key contribution  is the integration of a restarted heavy-ball-type inertia mechanism with score-based priors derived from deep denoisers, enabling provably faster convergence to stationary points compared to  RED. The authors provide both discrete algorithmic instantiations (RISP-GM and RISP-Prox) and a continuous-time dynamical systems interpretation linked to the heavy-ball ODE.

**Strengths:**

* Rigorous convergence analyses for both gradient-based (RISP-GM) and proximal (RISP-Prox) variants under realistic assumptions (e.g., Lipschitz Hessian).
* Demonstation of significant empirical speedups compared to RED on large scale inverse problems (debluring, super-resolution, inpainting and MRI)
* A continuous-time dynamical systems interpretation linked to the heavy-ball ODE is offered, enriching theoretical understanding.
* The performance of the proposed acceleration strategy is validated across diverse tasks, including linear and nonlinear inverse problems
  showing consistent acceleration and reconstruction quality

**Weaknesses:**

* The restart condition used in both algorithms (line 5 in both cases) uses a cumulative sum over all past iterations since last restart. This needs to be clarified as a running window accumulator to avoid confusion about global vs. local accumulation.
* The core idea, ombining restart + inertia + nonconvex smooth optimization is not new. The novelty mainly lies in the application domain (imaging with learned priors) and the compatibility with RED’s score interpretation.
* An ablation study on the sensitivity to hyperparameters is missing.
* Comparison to other acceleration strategies in PnP/RED is limited. For example Anderson acceleration (Hong et al., 2019) has been applied to RED and shows empirical speedups. Also,  FISTA-style momentum with adaptive restart (Alamo et al., 2019; Fercoq & Qu, 2019) could be a relevant baseline.

--

* References
Olivier Fercoq and Zheng Qu. "Adaptive restart of accelerated gradient methods under local quadratic growth condition". IMA Journal of Numerical Analysis, 39(4):2069–2095, 2019.

**Questions:**

While restart improves stability, the method introduces tunable hyperparameters (e.g., restart threshold B , inertia weight ε ). Although the authors claim robustness and provide some analysis in the appendix , more guidance or adaptive tuning strategies would make this work more practical.

---

> ### Author Response · Authors · 2025-11-25
>
> **[Weakness 1]** We appreciate the reviewer for catching this potential confusion. We have clarified this by rephrasing line 167 to include *"accumulated relative error since the last restart".*
>
> **[Weakness 2]** We thank the reviewer for recognizing the significance of our work. We want to take this opportunity to clarify our technical contributions.
>
> The technique of restarted inertia for nonconvex optimization has been studied in several prior works (see [7] and additional references in Appendix A). However, our work bridges this mechanism with regularization by denoising (RED), developing RISP as a provable accelerated extension that is compatible with learned score priors. We also address the technical challenges arising in this adaption. As noted by the reviewer in the *Strengths* section, we derive convergence analyses for both the gradient-based (RISP‑GM) and proximal (RISP‑Prox) variants. The proximal variant, in particular, requires special care compared with the gradient-based variant. For example, the quadratic approximation of the objective function, which is separable after a change of basis in RISP-GM, is no longer fully separable, and resolving this issue demands a different and more delicate treatment.
>
> To offer additional insight, we introduce a restarted ODE as the continuous-time analog of RISP and establish its convergence. Our continuous-time analysis uses different techniques than the discrete setting and avoids the quadratic approximation needed in discrete algorithms.
>
> **[Weakness 3]** In the revised Appendix G.5, we provided a detailed discussion on the sensitivity of both RISP algorithms to the restarting hyperparameters. We plot the convergence behavior across a range of momentum parameters $\theta \in [0.01, 1]$ as well as restart thresholds $B \in [100, 100000]$. Overall, the results show that RISP exhibits stable acceleration across a sufficiently wide range of $(\theta, B)$ combinations. We believe the new experiments has reinforced our prior discussion on hyperparameter sensitivity.
>
> **[Weakness 4]** Prompted by the remark, we conducted additional comparison between RISP and FISTA-PnP [4], which implements the Nesterov acceleration. The table below summarizes the results for image deblurring.
>
> **Table**: Comparison between FISTA-PnP, RISP, and RED methods on deblurring with $\sigma=5\%$ and the first blur kernel in Levin's blur kernel dataset. Iteration number is reported at the peak PSNR value. The same learned denoiser GS-DRUNet is used in all methods.
> Methods | PSNR ($\uparrow$) | Running time | Iteration number | Convergence Guarantee
> ---|---|---|---|---
> FISTA-PnP | 27.60 dB | 0.5s | 30 | $\times$
> RISP-Prox | 27.55 dB | 0.2s | 12 | $\checkmark$
> RISP-GM | 27.66 dB | 0.5s | 29 | $\checkmark$
> RED-Prox | 27.68 dB | 0.5s | 29 | $\checkmark$
> RED-GM | 28.03 dB | 2.5s | 143 | $\checkmark$
>
> As shown, RISP-Prox and RISP-GM attains comparable (sometimes even faster) convergence speed with FISTA-PnP. Note that FISTA-PnP uses the adaptive momentum update and lacks theoretical convergence guarantees. Convergence plots as well as detailed discussion is presented in Appendix G.7.
>
> **[Question]** Please refer to our response to [Weakness 3] regarding the sensitivity to hyperparameters. Note that smaller $\theta$ suggests stronger acceleration at the cost of increased oscillation, whereas smaller $B$ induces more frequent restarting, stablizing the convergence but tempering acceleration. In our experiments, we manually select appropriate values of $\theta$ and $B$ following this principle. Table 3 in the appendix summarizes the hyperparameter settings for each experiment presented in the main manuscript.
>
> **References**
>
> [1] Restarting accelerated gradient methods with a rough strong convexity estimate, O Fercoq, Z Qu, 2016
>
> [2] Restart FISTA with global linear convergence, T Alamo, D Limon, P Krupa, 18th European Control Conference (ECC), 2019
>
> [3] Acceleration of RED via vector extrapolation, T Hong, Y Romano, M Elad, JMIV, 2019
>
> [4] A plug-and-play priors approach for solving nonlinear imaging inverse problems, U Kamilov, H Mansour, B E Wohlberg, IEEE Signal Processing Letters, 2018
>
> [5] FISTA Iterates Converge Linearly for Denoiser-Driven Regularization, A Sinha, KN Chaudhury, SIAM Journal on Imaging Sciences, 2025
>
> [6] TFPnP: Tuning-free Plug-and-Play Proximal Algorithms with Applications to Inverse Imaging Problems, K Wei, A Aviles-Rivero, J Liang, Y Fu, H Huang, CB Schönlieb, JMLR 2022
>
> [7]  Restarted non convex accelerated gradient descent: No more polylogarithmic factor in the o(epsilonˆ(-7/4)) complexity. H and Z Lin. JMLR, 2023.

---

> > ### Comment · Reviewer_wKJo · 2025-11-26
> >
> > I appreciate the effort of the authors to perform the requested additional comparisons and for answering in detail to all my questions. All my comments have been adequately addressed and I will retain my positive score of this work.

---

### Meta-Review · Area_Chair_6qhW · 2026-01-06

**Summary:**

This paper proposes a simple and practical way to accelerate RED with score-based priors, by adding momentum and restart strategies. The main value is that the authors do not stop at an empirical improvement: they provide convergence-rate guarantees and also connect the method to a continuous-time dynamical system formulation. Both the analysis and the experiments suggest faster convergence than standard RED, even with nonlinear score-based priors.

Reviewer UnZv raises a fair concern that the algorithmic ingredients themselves are not highly novel, since momentum and restart are standard. In response, the authors clarify that the novelty lies in the setting and analysis: deriving acceleration guarantees for RED under non-convex score-based priors, and presenting RISP-prox from a continuous-time viewpoint, which is not clearly addressed in prior score-based RED work. While the conceptual novelty may be somewhat incremental, the technical development is solid and the results are consistent.

**Reviewer Concerns:**

.

**Reviewer Scores:**

.

---

### Decision · Program_Chairs · 2026-01-26

Accept (Poster)